# Normalized Space Alignment: A Versatile Metric for Representation Analysis

## Abstract

We introduce a manifold analysis technique for neural network representations. Normalized Space Alignment (NSA) compares pairwise distances between two point clouds derived from the same source and having the same size, while potentially possessing differing dimensionalities. NSA can act as both an analytical tool and a differentiable loss function, providing a robust means of comparing and aligning representations across different layers and models. It satisfies the criteria necessary for both a similarity metric and a neural network loss function. We showcase NSA's versatility by illustrating its utility as a representation space analysis metric, a structure-preserving loss function, and a robustness analysis tool. NSA is not only computationally efficient but it can also approximate the global structural discrepancy during mini-batching, facilitating its use in a wide variety of neural network training paradigms.

## 1 Introduction

Deep learning and representation learning have rapidly emerged as cornerstones of modern artificial intelligence, revolutionizing how machines interpret complex data. At the heart of this evolution is the ability of deep learning models to learn efficient representations from vast amounts of unstructured data, transforming it into a format where patterns become discernible and actionable. As these models delve deeper into learning intricate structures and relationships within data, the necessity for advanced metrics that can accurately assess and preserve the integrity of these learned representations becomes increasingly evident. The ability to preserve and align the representation structure of models would enhance the performance, interpretability and robustness (Finn et al., 2017) of these models.

Despite the significant advancements achieved through representation learning in understanding complex datasets, the challenge of precisely maintaining the structural integrity of representation spaces remains. Current methodologies excel at optimizing the relative positions of embeddings (Chen et al., 2020; Schroff et al., 2015), yet they often overlook the challenge of preserving exact structural relationships to improve the performance, interpretability, and robustness of neural networks. A similarity index that is computationally efficient and differentiable would not only empower practitioners to navigate high-dimensional spaces with ease but also facilitate the seamless integration of similarity-based techniques into gradient-based optimization pipelines.

We present a distance preserving structural similarity index [1] called "Normalized Space Alignment (NSA)." NSA measures both the global and the local discrepancy in structure between two spaces while ensuring computational efficiency. The compared representations can lie in different dimensional spaces provided there is a one to one mapping between the points. Globally, NSA measures the average discrepancy in pairwise distances; locally, it measures the average discrepancy in Local Intrinsic Dimensionality (Houle, 2017) of each point.

While manifold learning (Tenenbaum et al., 2000; Islam et al., 2023; Fefferman et al., 2016; Genovese et al., 2012) has become integral to deep learning, when applied to deep learning, they encounter three primary challenges (Psenka et al., 2023): (1) the lack of an explicit projection map between the original data and its corresponding low-dimensional representation, making adaptation to unseen data difficult; (2) the inability to scale to large datasets due to computational constraints; and (3) the

---

[1]Since NSA is a distance-based measure, it measures dissimilarity but we use the term 'similarity metric' colloquially throughout the paper to refer to metrics that quantify relationships between representation spaces.

extensive parameter tuning required, along with the need for a predefined understanding of the data. NSA aims to address these challenges while offering versatility in deep learning applications.

In line with previous works that introduce and theoretically establish similarity indices (Kornblith et al., 2019; Barannikov et al., 2022; Chen et al., 2022), we adopt a breadth-first approach in this paper. Rather than focusing on a specific application, we present NSA's broad applicability across various use cases, establishing its viability in a wide range of scenarios. Future work can explore NSA's impact in greater depth within specialized fields. Our contributions are summarized below:

- NSA is proposed as a differentiable and efficiently computable index. NSA's quadratic computational complexity (in the number of points) is much better than some of the existing measures (e.g., cubic in the number of simplices for RTD (Barannikov et al., 2022)). NSA is shown to be a viable loss function and its computation over mini-batches of the data is proven to be representative of the global NSA value.

- NSA's viability as a similarity index is evaluated in Section 4.1 and as a loss function in Section 4.2. It can be used as a supplementary loss in autoencoders for dimensionality reduction. Results on several datasets and downstream task analyses show that NSA preserves structural characteristics and semantic relationships of the original data better compared to previous works.

- NSA can analyze perturbations that result from adversarial attacks on GNNs(Section 4.4). NSA shows a high correlation with misclassification rate across varying perturbation rates and can also provide insights on node vulnerability and the inner workings of popular defense methods. Simple tests with NSA can provide insights on par with other works that review and investigate the robustness of Neural Networks (Mujkanovic et al., 2022; Jin et al., 2020a).

## 2 RELATED WORK

Early proposed measures of Structural Similarity Indices were based on variants of "Canonical Correlation Analysis (CCA)" (Morcos et al., 2018; Raghu et al., 2017). "Centered Kernel Alignment (CKA)" (Kornblith et al., 2019) utilizes a Representational Similarity Matrix (RSM) with mean-centered representations and a linear kernel to measure structural similarity. But it was shown to lack sensitivity to certain representational changes (Ding et al., 2021) and not satisfy the triangle inequality (Williams et al., 2021). Barannikov et al. (2022) proposed "Representation Topology Divergence (RTD)," to measure the dissimilarity in multi-scale topology between two point clouds of equal size with a one-to-one point correspondence. RTD has better correlation with model prediction disagreements than CKA but its high computational complexity and limited practicality for larger sample sizes constrain its wider application. Recently, Chen et al. (2023) proposed a filter subspace based similarity measure which can compare neural network similarity by utilising the weights from the convolutional filter of CNNs. Although not strictly a feature representation based similarity measure, it is capable of evaluating neural network similarity more efficiently than CKA or CCA.

Besides RTD and CKA, multiple other similarity indices have been proposed. Alignment based measures (Ding et al., 2021; Williams et al., 2022; Li et al., 2016; Hamilton et al., 2018; Wang et al., 2018) aim to compare pairs of representation spaces directly. Representational Similarity Matrix based measures (Shahbazi et al., 2021; Kriegeskorte et al., 2008; Kornblith et al., 2019; Székely et al., 2007; Tang et al., 2020; May et al., 2019; Boix-Adsera et al., 2022; Chen et al., 2022) generate an RSM that describes the similarity between representations of each instance in order to avoid measuring representations directly. This helps overcome the dimensionality limitation, allowing RSM based measures to compute similarity regardless of what dimension the two spaces lie in. Neighborhood based measures (Schumacher et al., 2020; Hamilton et al., 2016) compare the nearest neighbors of instances in the representation space by looking at the consistency in the k-nearest neighbors of each instance in the representation space. Topology based measures (Khrulkov & Oseledets, 2018; Tsitsulin et al., 2020b; Barannikov et al., 2022) are motivated by the manifold hypothesis. These measures aim to approximate the manifold of the representation space in terms of discrete topological structures such as graphs of simplicial complexes (Goodfellow et al., 2016b). Previous works are either incapable of comparing spaces across dimensions, do not satisfy one or more of the invariance conditions required to be an effective similarity index, have high computational complexity, cannot satisfy the conditions necessary to be a pseudometric or are not differentiable.

## 3 NORMALIZED SPACE ALIGNMENT

NSA comprises of two distinct but complementary components: Local NSA (LNSA) and Global NSA (GNSA). LNSA focuses on preserving the local neighborhoods of points, leveraging Local Intrinsic Dimensionality to ensure nuanced neighborhood consistency. In contrast, GNSA extends this analysis to a broader scale, aiming to maintain relative discrepancies between points across the entire representation space. Together, these components provide a multifaceted understanding of representation spaces. As a similarity index (Kornblith et al., 2019), NSA does not distinguish between two point clouds that belong to the same $E(n)$ symmetry group.

### 3.1 LOCAL NORMALIZED SPACE ALIGNMENT (LNSA)

LNSA focuses on the alignment of local neighborhoods of individual points within a dataset. This approach is grounded in the extensive research on Local Intrinsic Dimensionality (LID) (Bailey et al., 2022; Houle, 2017; Tsitsulin et al., 2020a; Camastra & Staiano, 2016). Prior to defining LNSA, we provide a foundational overview of LID, setting the stage for a comprehensive understanding of how LNSA leverages these principles to achieve precise alignment in local representation spaces.

#### 3.1.1 LOCAL INTRINSIC DIMENSIONALITY

Given a point cloud $X = \{ \vec{x}_1, \dots, \vec{x}_N \} \subseteq \mathbb{R}^n$, we say $X$ has intrinsic dimensionality $m$ if all points of $X$ lie in an $m$-dimensional manifold (Fukunaga, 1982). LID is intuitively defined to find the lowest dimensional manifold containing a local neighbor of a point of importance. Houle (2017) formally defined local intrinsic dimensionality as follows:

**Definition 3.1.** Given a point cloud $X \subseteq \mathbb{R}^n$ and a point of interest $\vec{x} \in X$. Let $R_{\vec{x}}$ be the random variable denoting distances of all points from $\vec{x}$ and $F_{\vec{x}}(\cdot)$ be the cumulative distribution function of $R_{\vec{x}}$. Let $F_{\vec{x}}$ be continuous and differentiable, then the LID of $\vec{x}$ is

$$\mathsf{IntrDim}_X(\vec{x}) = \lim_{\substack{\varepsilon \to 0 \\ r \to 0}} \frac{F_{\vec{x}}((1+\varepsilon)r) - F_{\vec{x}}(r)}{\varepsilon F_{\vec{x}}(r)}. \tag{1}$$

In practice, we estimate the LID of a point using the distances to its $k$ nearest neighbours. Amsaleg et al. (2015) defined the Maximum Likelihood Estimator for LID and defined lid of $\vec{x}_i \in X$ as:

$$\mathsf{lid}(i, X) = -\left( \frac{1}{k} \sum_{\substack{i_j: \\ (i_1, \dots, i_k) \leftarrow kNN_X(i)}} \log \frac{d(\vec{x}_i, \vec{x}_{i_j})}{d(\vec{x}_i, \vec{x}_{i_k})} \right)^{-1}, \tag{2}$$

where $(i_1, \dots, i_k) \leftarrow kNN_X(i)$ denotes that the $k$ nearest neighbours of $\vec{x}_i$ in $X$ are $(\vec{x}_{i_1}, \dots, \vec{x}_{i_k})$. $d(\cdot, \cdot)$ is a differentiable distance measure between two vectors.

When aligning two points in different spaces to ensure they have similar local structures, the objective extends beyond merely matching their local dimensions. It also involves ensuring that both points share the same neighbors. To achieve this we have to identify the same $k$ points near some point $i$ in space $X$ and $Y$. Hence we define the notion of approximate dimensionality (AD) of a point $\vec{x}_i$ with respect to any $k$ tuple $(\vec{x}_{i_1}, \dots, \vec{x}_{i_k})$ as:

$$\mathsf{AD}_X(i, (i_1, \dots, i_k)) = -\left( \frac{1}{k} \sum_{j=1}^{k} \log \frac{d(\vec{x}_i, \vec{x}_{i_j})}{d(\vec{x}_i, \vec{x}_{i_k})} \right)^{-1}. \tag{3}$$

The measure lid is then defined as: $\mathsf{lid}(i, X) = \mathsf{AD}_X(i, kNN_X(i))$.

Levina & Bickel (2004) defined the intrinsic dimensionality of a space as the average LID of all its points. This was later corrected by MacKay & Ghahramani (2005) to be the reciprocal of the average reciprocal of LID of each point. Formally defined as:

$$\mathsf{IntrDim}(X) = \left( \frac{1}{|X|} \sum_{\vec{x}_i \in X} \mathsf{lid}(i, X)^{-1} \right)^{-1}. \tag{4}$$

MacKay & Ghahramani (2005) showed that performing aggregation operations on the reciprocals of the LIDs of each point yields much better approximations.

### 3.1.2 DEFINITION

To define Local NSA (LNSA) using LID, we start with the premise that closely structured point clouds should have similar dimensional manifolds in their local neighborhoods. Formally, for two point clouds $X$ and $Y$, and considering the $k$ nearest neighbours of the $i^{th}$ point in $X$ indexed as $i_1, \ldots, i_k$, we examine the manifold dimensions in $Y$ that contain $i, i_1, \ldots, i_k$. This leads to the concept of "Symmetric lid" defined as:

$$\overline{\mathsf{lid}}_X(i, Y) = \mathsf{AD}_Y(i, kNN_X(i)). \tag{5}$$

We notice that the mean discrepancy of the reciprocals of the LID is strongly correlated with the local structure being preserved, this is similar to MacKay & Ghahramani (2005) where they show that taking a mean of reciprocals of the LID is correlated to the intrinsic dimensionality of the whole space. Hence we define LNSA of $X$ with respect to $Y$ as:

$$\mathsf{LNSA}_Y(X) = N^{-1} \sum_{1 \le i \le N} \left( \mathsf{lid}(i, Y)^{-1} - \overline{\mathsf{lid}}_Y(i, X)^{-1} \right)^2. \tag{6}$$

Further, we can symmetricise this term to get the premetric $\mathsf{LNSA}^{\mathsf{metric}}$ as:

$$\mathsf{LNSA}^{\mathsf{metric}}(X, Y) = \mathsf{LNSA}_X(Y) + \mathsf{LNSA}_Y(X). \tag{7}$$

When using NSA-based loss functions, we use LNSA whereas when using NSA-based similarity indices, we use $\mathsf{LNSA}^{\mathsf{metric}}$.

### 3.1.3 ANALYTICAL PROPERTIES

We show that $\mathsf{LNSA}^{\mathsf{metric}}$ is a premetric by proving the necessary properties in Appendix D: • $\mathsf{LNSA}^{\mathsf{metric}}(X, X) = 0$, • Symmetry, • Non-negativity.

LNSA is shown to satisfy the necessary conditions of Invariance to Isotropic Scaling, Invariance to Orthogonal Transformation, and (Not) Invariance to Invertible Linear Transformation as proposed by Kornblith et al. (2019) for a similarity index. These condition are proved in Appendix E and ensure that the similarity index is unaffected by rotation or scaling of the representation space.

## 3.2 GLOBAL NORMALIZED SPACE ALIGNMENT (GNSA)

GlobalNSA is a similarity index based on Representational Similarity Matrices (RSMs) (Klabunde et al., 2023). Given a representation $\mathbf{R} : N \times D$, all RSM based measures generate a matrix of instance wise similarities $D \in \mathbf{R}^{N \times N}$ where $D_{i,j} := d(R_i, R_j)$. Given two representations $\mathbf{R}$ and $\mathbf{R}'$, two corresponding RSMs are generated. GlobalNSA is calculated by subtracting the pairwise similarities between the two spaces and averaging these discrepancies to produce a single value that reflects the global difference between the two representation spaces. We present the motivation behind GlobalNSA and how RSMs capture global discrepancy in Appendix C.3.

### 3.2.1 DEFINITION

The GNSA between two point clouds $X = \{\vec{x_1}, \ldots, \vec{x_N}\}$ and $Y = \{\vec{y_1}, \ldots, \vec{y_N}\}$ is computed as:

$$\mathsf{GNSA}(X, Y, i) = \frac{1}{N} \sum_{1 \le j \le N} \left| \frac{d(\vec{x_i}, \vec{x_j})}{\max\limits_{\vec{x} \in X} d(\vec{x}, \vec{0})} - \frac{d(\vec{y_i}, \vec{y_j})}{\max\limits_{\vec{y} \in Y} d(\vec{y}, \vec{0})} \right|. \tag{8}$$

$$\mathsf{GNSA}(X, Y) = \frac{1}{N} \sum_{1 \le i \le N} \mathsf{GNSA}(X, Y, i). \tag{9}$$

Throughout the paper, we use $d(\cdot, \cdot)$ to denote the euclidean distance (unless specified otherwise).

---

[2] $\vec{0}$ denotes a vector (of appropriate size) which is the origin of the space.

### 3.2.2 ANALYTICAL PROPERTIES

We show that GNSA is a pseudometric by proving the necessary properties in the Appendix A: •
$GNSA(X, X) = 0$, • Symmetry, • Non-negativity, • Triangle Inequality.
GNSA is proven to be a similarity index by satisfying the three established criteria in Appendix B.

### 3.3 PUTTING IT ALL TOGETHER

We utilize GNSA for our empirical analysis as it performs well when comparing entire representation
spaces. When using NSA as a structure-preserving loss function, we calculate the final NSA Loss
through a weighted sum of local and global NSA, allowing us to adjust the emphasis between local
and global perspectives. One could prioritize local neighborhood preservation, such as in classifiers
with tightly clustered data, or emphasize global structure, as needed in link prediction models for
densely interconnected graphs. Hence, NSA is defined as:

$$NSA = l * LNSA + g * GNSA. \tag{10}$$

Only GNSA is used during empirical analysis while both GNSA and LNSA are used during dimen-
sionality reduction. We present ablation studies on how the two components affect performance,
quantifying the difference and visualizing it in Appendix Q.

### 3.3.1 COMPLEXITY ANALYSIS

LNSA's computational complexity is $O(N^2D + kND)$ [3] and GNSA's complexity is $O(N^2D)$ where
$N$ is the number of data points, $k$ is the number of nearest neighbours considered in LNSA and $D$ is
$\max(D(R), D(R'))$ with $R$ and $R'$ being the two representation spaces and $D(\cdot)$ the dimensionality
of the space. Hence, overall NSA has complexity of $O(N^2D + kND)$. Empirically, we find that
computing NSA on a GPU scales quadratically up to a very high batch size (Appendix H). CKA has
similar complexity to NSA while RTD has a complexity of $O(N^2D + R^3)$, where $R$ is the number
of simplices (in practice $R >> N$). Running times of NSA-AE are given in Appendix M. NSA-AE
is several times faster than RTD-AE and it can run on much larger batch sizes.

## 4 APPLICATIONS

NSA serves a dual purpose, acting not only as a similarity index to compare structural discrepancies
in representation spaces, but also as a structure preserving loss function. This enables a wide range of
applications where NSA could potentially be useful. We first validate NSA's viability as a similarity
index using the tests proposed by (Ding et al., 2021). We then present experiments in two broad
domains; dimensionality reduction and adversarial attack analysis. Additional experiments on cross-
layer analysis (Appendix V and Appendix J), convergence prediction (Appendix W) and manifold
approximation (Appendix R) are presented in the appendix.

### 4.1 ANALYZING REPRESENTATION SPACES

We evaluate the performance of our structural similarity index using statistical tests proposed by Korn-
blith et al. (2019) and further refined by Ding et al. (2021). These tests are based on two fundamental
intuitions: (1) a good structural similarity index should show the highest relative similarity between
corresponding layers of architecturally identical neural networks with different weight initializations,
and (2) it should be sensitive to the removal of high variance principal components from the data.
These initial tests are essential for validating the effectiveness of a similarity index.

### 4.1.1 SPECIFICITY TEST

When analyzing layerwise dissimilarity using structural similarity indices, a good metric is expected
to assign the lowest dissimilarity to corresponding layers of neural networks trained with different

---

[3]The time complexity of computing the $k$-nearest neighbours of each point is $O(N^2D + N^2 \log k)$ since it
takes $O(N^2D)$ time to find all pairwise distances and finding $k$ minimum elements of $N$ lists of size $N$ can
be done in $O(N^2 \log k)$ using min-heaps of size $k$. Since $D >> \log k$ in most cases, we write the total time
complexity as $O(N^2D)$.

initialization seeds while ensuring that the dissimilarity between non-corresponding layers remains higher. This expectation stems from the intuition that neural networks with the same architecture, trained on the same data but initialized differently, should exhibit structurally similar intermediate representations for corresponding layers.

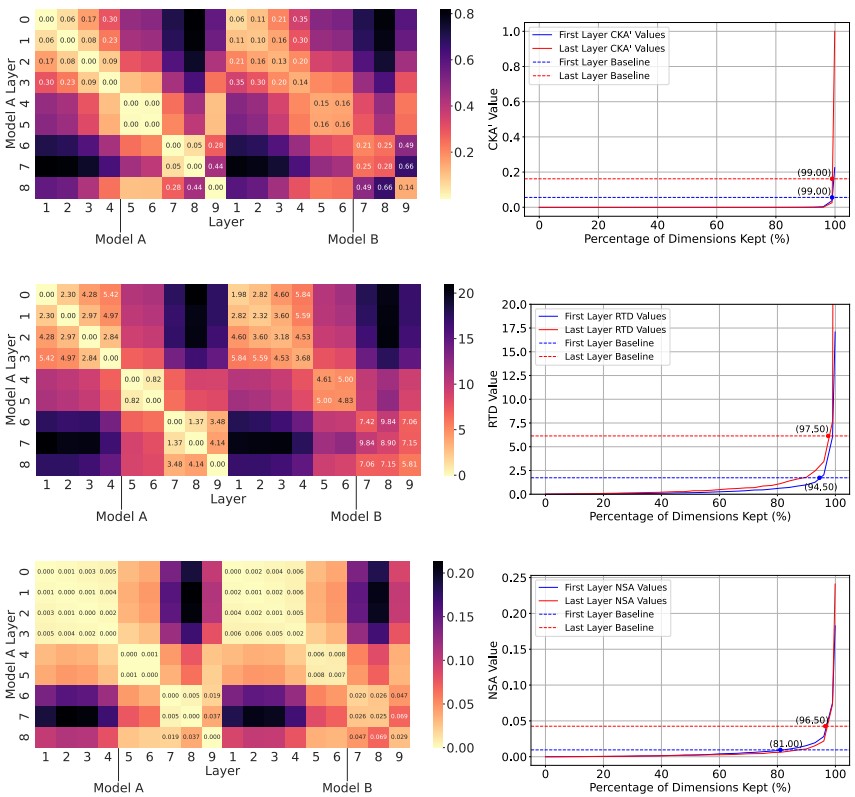

Figure 1: Specificity (Left) and Sensitivity (Right) Tests. Left (from top to bottom): CKA', RTD and NSA pairwise distances between each layer of two differently initialized networks. Right: Dissimilarities between a layer's representation and its low-rank approximation. Principal components are deleted in order of least variance.

In the first experiment, we evaluate whether the dissimilarity assigned to corresponding layers (e.g., layer 7 in model A vs. layer 7 in model B where A and B are the same architecture with different initializations) is lower than the dissimilarity assigned to non-corresponding layers (e.g., layer 7 in model A vs. layer 6 in model B). We trained ResNet-18 models on CIFAR-10 with different initialization seeds and computed layerwise dissimilarity on the intermediate embeddings[4] using NSA, CKA, and RTD. Our results in Figure 1 demonstrate that NSA assigns the lowest dissimilarity to corresponding layers across different initializations. However, both CKA (layers 6 and 8) and RTD (layers 7 and 8) fail to maintain this behavior in certain layers. We also extend this analysis to include comparisons between layers of the same model trained with the same seed and present our results and experimental setup in Appendix U.

### 4.1.2 SENSITIVITY TEST

An effective structural similarity index should be robust to the removal of low variance principal components and sensitive to the removal of high variance principal components. Previous studies have shown that CKA is insensitive to the removal of principal components until the most significant ones are removed. We conducted a sensitivity test with our ResNet-18 models trained on CIFAR-10, following a similar experimental setup to that used by Ding et al. (2021).

---

[4]Since RTD and NSA measure dissimilarity while CKA measures similarity, we present the results of 1-CKA for consistency in all our experiments.

For each index, we set the dissimilarity detectable threshold to be the score between representations from different seed initializations. The percentage of principal components that need to be deleted for the dissimilarity score to rise beyond the detectable threshold determines the sensitivity of the metric. The results on the first and last layer, shown in Figure 1, demonstrate that NSA is more sensitive to the removal of high variance principal components compared to RTD and CKA, by crossing the detection threshold when removing an average of 90.25% of principal components, compared to 96% for RTD and 99% for CKA.

## 4.2 NSA as a Loss Function

The viability of NSA as a loss function is established by demonstrating its non-negativity, nullity and continuity, and by developing a differentiation scheme. Proofs for all four properties are presented in Appendix C and F. We also demonstrate that the expectation of NSA over a minibatch is equal to the NSA of the whole dataset, cementing its feasibility as a loss function in Appendix G.

### 4.2.1 Convergence of Global NSA in mini batching

Neural networks are often trained on mini batches, which means that at any step a global structure preserving loss will only have access to a subset of the data. It is vital that a structure preserving loss function like GNSA converge to its global measure when used as a loss function. Similarity indices usually have a high computation cost, therefore most similarity indices work with a sample of the data and show that the value computed on the sample is a good approximation of the global value. Additionally, for a loss function, the expectation is that the mean of the losses from the mini-batches converges to the true loss over the entire dataset. This principle underlies stochastic gradient descent (SGD) and its variants, which are foundational to training neural networks. SGD assumes that the expected value of the gradients calculated on the mini-batches converges to the true gradient over the entire dataset, and a similar concept applies to the expectation of the loss. In Lemma G.1 we prove theoretically that in expectation, GNSA over a minibatch is equal to GNSA of the whole dataset.

Figure 2 shows that subset GNSA averaged over increasing trials approximates the GNSA of the entire dataset while RTD fails to do so. Here, we use the output embeddings of two GCN models with the exact same architecture but different initializations trained on node classification onthe Cora dataset. Using subset sizes of 200 and 500, we randomly sample from the original dataset and compute the metrics, averaging their results and comparing it to the value of the entire dataset.

While LNSA does not exhibit formal convergence under mini-batching, it intuitively demonstrates favorable behavior in such settings. LNSA's design, which relies on nearest-neighbor computations to assess structural preservation within local neighborhoods, ensures that these neighborhoods are typically preserved in expectation. This is because the sampling process in mini-batching generally retains a sufficient number of neighboring points in each batch. By appropriately adjusting the number of nearest neighbors $k$ considered in each mini-batch, LNSA is able to approximate the structural consistency of local representations, ensuring that the loss function remains representative of the global structure on a local scale. We present empirical convergence results on LNSA in Appendix Q where we discuss the effects of varying $k$ and observed best practices.

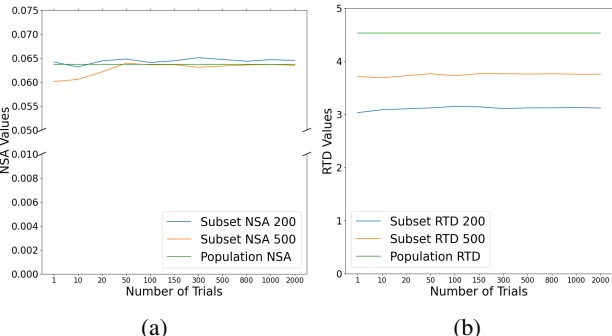

(a)                                        (b)

Figure 2: Expectation of subset metrics over a large number of trials. (a) Mean subset GlobalNSA variation over increasing trials. (b) Mean subset RTD variation over increasing trials.

### 4.2.2 DEFINING NSA-AE: AUTOENCODER USING NSA

To evaluate the efficacy of NSA as a structural discrepancy minimization metric, we take inspiration from TopoAE Moor et al. (2020) and RTD-AE Trofimov et al. (2023), and use NSA as loss function in autoencoders for dimensionality reduction. A normal autoencoder minimizes the MSE loss between the original $X$ and reconstructed embedding $\hat{X}$. We add NSA-Loss as an additional loss term that aims to minimize the discrepancy in representation structure between the original embedding space $X$ and the latent embedding space $Z$. The autoencoder is built as a compression autoencoder where the encoder attempts to reduce the original data to a latent dimension and the decoder attempts to reconstruct the original embedding. Since NSA can be used in autoencoders with mini-batch training, NSA-AE runs almost as fast as a regular autoencoder. We compare the performance of NSA-AE against PCA, UMAP, a regular autoencoder, TopoAE and RTD-AE on four real world datasets.

The performance of NSA-AE is evaluated on the structural and topological similarity between the input data $X$ and the latent data $Z$. We use the following evaluation metrics: (1) linear correlation of pairwise distances, (2) triplet distance ranking accuracy (Wang et al., 2021), (3) RTD (Barannikov et al., 2022), (4) GNSA, (5) LNSA and (6) kNN-Consistency. As seen in Table 1, NSA-AE has better structural similarity between the original data and the latent data compared to the other models. NSA-AE achieves MSE and running times similar to a normal autoencoder while previous works are several times slower, as presented in Appendix M. We also present the hyperparameters used along with dataset statistics in Appendices L and P.

Table 1: Autoencoder results. L.C = Linear Correlation, T.A. = Triplet distance ranking accuracy, RTD = Representation Topology Divergence, GNSA = GlobalNSA, LNSA@100 = LocalNSA on 100 nearest neighbors, kNN C @ 100 = 100 nearest neighbors consistency. NSA-AE outperforms or almost matches all other approaches on all the evaluation metrics. RTD-AE, which explicitly minimizes on RTD occasionally has lower LID and kNN-C.

| Dataset | Method | Quality measure | | | | | |
| | | L. C. | T. A. | RTD | GNSA | LNSA @ 100 | kNN-C @ 100 |
|---|---|---|---|---|---|---|---|
| MNIST | PCA | 0.910 | $0.871 \pm 0.008$ | $6.69 \pm 0.21$ | $0.0817 \pm 0.0025$ | $0.0267 \pm 0.00$ | 0.584 |
| | UMAP | 0.424 | $0.620 \pm 0.013$ | $18.06 \pm 0.48$ | $0.2305 \pm 0.0031$ | $1.0353 \pm 0.02$ | 0.353 |
| | AE | 0.801 | $0.778 \pm 0.007$ | $7.47 \pm 0.20$ | $0.0571 \pm 0.0011$ | $0.0529 \pm 0.00$ | 0.619 |
| | TopoAE | 0.765 | $0.771 \pm 0.010$ | $6.16 \pm 0.23$ | $0.0477 \pm 0.0011$ | $0.0335 \pm 0.00$ | 0.631 |
| | RTD-AE | 0.837 | $0.811 \pm 0.004$ | $4.26 \pm 0.14$ | $0.1694 \pm 0.0024$ | $0.0133 \pm 0.00$ | **0.709** |
| | NSA-AE | **0.947** | $\mathbf{0.886 \pm 0.005}$ | $\mathbf{4.29 \pm 0.11}$ | $\mathbf{0.0222 \pm 0.00}$ | $\mathbf{0.0123 \pm 0.00}$ | 0.634 |
| F-MNIST | PCA | 0.978 | $\mathbf{0.951 \pm 0.006}$ | $5.91 \pm 0.12$ | $0.1722 \pm 0.0038$ | $0.0566 \pm 0.00$ | 0.513 |
| | UMAP | 0.592 | $0.734 \pm 0.012$ | $12.16 \pm 0.39$ | $0.1420 \pm 0.0011$ | $0.6738 \pm 0.01$ | 0.365 |
| | AE | 0.872 | $0.850 \pm 0.008$ | $5.60 \pm 0.21$ | $0.0527 \pm 0.0028$ | $0.0515 \pm 0.00$ | 0.614 |
| | TopoAE | 0.875 | $0.854 \pm 0.009$ | $4.27 \pm 0.15$ | $0.111 \pm 0.0027$ | $0.0291 \pm 0.00$ | 0.601 |
| | RTD-AE | 0.949 | $0.902 \pm 0.004$ | $3.05 \pm 0.12$ | $0.0349 \pm 0.0015$ | $\mathbf{0.0104 \pm 0.00}$ | **0.661** |
| | NSA-AE | **0.985** | $0.939 \pm 0.003$ | $\mathbf{2.78 \pm 0.09}$ | $\mathbf{0.0114 \pm 0.00}$ | $0.0133 \pm 0.00$ | 0.633 |
| CIFAR-10 | PCA | 0.972 | $0.926 \pm 0.009$ | $4.99 \pm 0.16$ | $0.1809 \pm 0.0046$ | $0.0175 \pm 0.00$ | 0.432 |
| | UMAP | 0.756 | $0.786 \pm 0.010$ | $12.21 \pm 0.22$ | $0.1316 \pm 0.0026$ | $0.2880 \pm 0.00$ | 0.119 |
| | AE | 0.834 | $0.836 \pm 0.006$ | $4.07 \pm 0.28$ | $0.0616 \pm 0.0019$ | $0.0224 \pm 0.00$ | 0.349 |
| | TopoAE | 0.889 | $0.854 \pm 0.007$ | $3.89 \pm 0.11$ | $0.0625 \pm 0.0014$ | $0.0277 \pm 0.00$ | 0.360 |
| | RTD-AE | 0.971 | $0.922 \pm 0.002$ | $\mathbf{2.95 \pm 0.08}$ | $0.0113 \pm 0.0003$ | $0.0124 \pm 0.00$ | 0.416 |
| | NSA-AE | **0.985** | $\mathbf{0.941 \pm 0.003}$ | $3.04 \pm 0.16$ | $\mathbf{0.0086 \pm 0.00}$ | $\mathbf{0.0103 \pm 0.00}$ | **0.437** |
| COIL-20 | PCA | 0.966 | $\mathbf{0.932 \pm 0.005}$ | $6.49 \pm 0.23$ | $0.2204 \pm 0.0$ | $0.2000 \pm 0.00$ | 0.801 |
| | UMAP | 0.274 | $0.567 \pm 0.016$ | $15.50 \pm 0.67$ | $0.1104 \pm 0.0$ | $3.3023 \pm 0.00$ | 0.569 |
| | AE | 0.850 | $0.836 \pm 0.008$ | $9.57 \pm 0.27$ | $0.0758 \pm 0.0$ | $0.5678 \pm 0.00$ | 0.683 |
| | TopoAE | 0.804 | $0.805 \pm 0.011$ | $7.33 \pm 0.21$ | $0.0676 \pm 0.0$ | $0.1044 \pm 0.00$ | 0.654 |
| | RTD-AE | 0.908 | $0.871 \pm 0.005$ | $5.89 \pm 0.10$ | $0.0523 \pm 0.0$ | $0.0496 \pm 0.00$ | 0.761 |
| | NSA-AE | **0.976** | $0.919 \pm 0.005$ | $\mathbf{3.55 \pm 0.18}$ | $\mathbf{0.0139 \pm 0.00}$ | $\mathbf{0.0085 \pm 0.00}$ | **0.809** |

### 4.3 DOWNSTREAM TASK ANALYSIS ON LINK PREDICTION

We demonstrate that NSA-AE effectively preserves the structural integrity between the original data and the latent space. However, it is imperative to note that such preservation does not inherently ensure the utility of the resultant latent embeddings. Metrics like NSA, which focus on exact structure preservation, excel in scenarios where the preservation of semantic relationships between data points holds paramount importance.

Link prediction is an ideal task to demonstrate NSA's structure preserving ability. Successful link prediction mandates a global consistency within the embedding space, necessitating that nodes with likely connections are proximate while dissimilar nodes are distant. Any form of clustering or localized alterations to the representation structure can disrupt node interrelationships. In our study,

we employ a Graph Convolutional Network (GCN) to train link prediction models across four distinct graph datasets. The original GCN embedding space encompasses 256 dimensions. This space is subsequently processed through NSA-AE, resulting in a reduced-dimensional representation. The latent embeddings are then tested on their ability to predict the existence of links between nodes. Notably, the latent embeddings produced by NSA-AE exhibit superior performance when compared to both a normal AE and RTD-AE as shown in Table 2. Additionally, we demonstrate that NSA can work at scale by working with the Word2Vec dataset. We present results showcasing the effectiveness of semantic textual similarity matching using Word2Vec embeddings in the Appendix K.

Table 2: Downstream task analysis with Link Prediction. The outputs from a GCN trained on link prediction are used as input. Latent embeddings are obtained at various dimensions and ROC-AUC scores are calculated on the reduced dimension embeddings using the original ground truth labels. NSA-AE outperforms both a regular autoencoder and RTD-AE across almost all datasets and all latent dimensions. The first row presents the results of training a normal GCN at different dimensionalities for reference. NSA-AE achieves competitive performance with GCN models trained from scratch.

| Method | Latent Dim | Dataset | | | |
|---|---|---|---|---|---|
| | | Amazon Comp | Cora | Citeseer | Pubmed |
| GCN | 128 | 95.59 | 98.9 | 98.71 | 97.39 |
| AE | 128 | 50.478 | 50.0 | 50.0 | 51.17 |
| RTD-AE | 128 | 94.63 | 86.52 | 90.54 | 91.80 |
| NSA-AE | 128 | **95.75** | **97.83** | **97.91** | **97.37** |
| GCN | 64 | 96.00 | 99.00 | 98.76 | 97.38 |
| AE | 64 | 50.56 | 50.0 | 50.0 | 50.03 |
| RTD-AE | 64 | 91.63 | 82.38 | 89.23 | **97.26** |
| NSA-AE | 64 | **94.60** | **97.89** | **98.38** | 96.97 |
| GCN | 32 | 96.48 | 98.83 | 98.55 | 97.32 |
| AE | 32 | 59.63 | 49.84 | 49.80 | 52.78 |
| RTD-AE | 32 | 88.34 | 83.13 | 57.18 | 87.05 |
| NSA-AE | 32 | **95.76** | **97.68** | **96.11** | **97.66** |

## 4.4 ANALYZING ADVERSARIAL ATTACKS

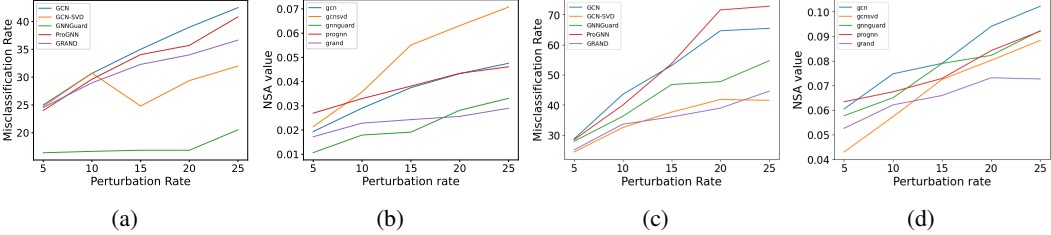

(a)  (b)  (c)  (d)

Figure 3: Robustness tests on GNN architectures with NSA. (a) Misclassification Rate against Data Perturbation Rate under global evasion attack. (b) NSA against perturbation rate under global evasion attack. (c) Misclassification Rate against Data Perturbation Rate under global poisoning attack. (d) NSA against perturbation rate under global poisoning attack

Adversarial attacks have become a critical area of concern across various domains in machine learning, from computer vision to natural language processing and graph-based data. These attacks exploit vulnerabilities in model architectures, to introduce subtle perturbations in the model's representation space to drastically alter its predictions. A structural discrepancy analysis term like NSA can help analyze and identify vulnerabilities in these networks. By comparing the underlying structure of data representations before and after an attack, NSA allows us to quantitatively assess how adversarial perturbations impact the model's predictions, regardless of the architecture being employed. For our study, we applied NSA in the context of GNNs, but the method can be equally effective in analyzing the robustness of other architectures, such as CNNs or transformers, under adversarial conditions.

In this experimental study, we subjected five distinct GNN architectures to both poisoning and evasion adversarial attacks using projected gradient descent Metattack(Xu et al., 2019; Mujkanovic et al., 2022) on the Cora Dataset. We use a GCN along with four robust GNN variants; SVD-GCN (Entezari et al., 2020), GNNGuard (Zhang & Zitnik, 2020), GRAND (Chamberlain et al., 2021) and ProGNN (Jin et al., 2020b). To assess the vulnerability of these architectures, we perturbed the initial adjacency

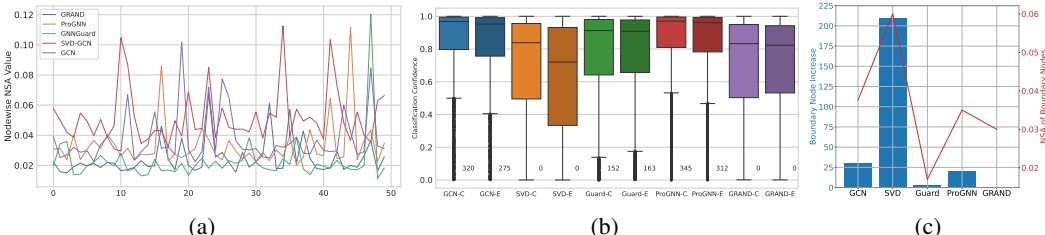

(a)                (b)                (c)

Figure 4: Analyzing node vulnerability with NSA. (a) Nodewise NSA of the 50 nodes with the greatest decline in Classification Confidence. SVD-GCN has the highest nodewise NSA variations. (b) Distribution of Classification Confidence before and after an evasion attack on various Graph Neural Network architectures. A suffix of 'C' after the architecture name refers to original dataset results and a suffix of 'E' refers to confidence on the poisoned dataset (c) Increase in number of boundary nodes for each model post attack and its correlation with the NSA of the boundary nodes.

matrices by introducing edge additions ranging from 5% to 25% of the total edges. The objective was to gauge the impact of these perturbations on the misclassification rates of the GNN models and to see if NSA shows a strong correlation to the misclassification rates over different perturbation rates.

NSA was computed at each stage by comparing the entire graph's output representations (Cora has 2708 nodes) with those from the perturbed graph. Our experiments revealed that NSA's variation over different perturbation rates mirrors the misclassification trends of various GNN architectures during poisoning attacks Figure 3 and NSA's ranking of architecture vulnerability is in line with previous works (Jin et al., 2020a; Mujkanovic et al., 2022).

However, in evasion attacks, SVD-GCN stood out with its NSA scores significantly deviating from its misclassification rates. To further explore such anomalies, NSA can also be used to measure node-level discrepancies. Figure 4 examines the boundary nodes of different GNN architectures. Post-attack, SVD-GCN showed a substantial increase in boundary nodes—defined by low classification confidence and proximity to the decision boundary—accompanied by a general decline in classification confidence. This surge, coupled with SVD-GCN's highest pointwise NSA values among all tested architectures, potentially accounts for its high NSA scores and heightened vulnerability despite appearing to be robust when evaluated only using accuracy based metrics. It is also this vulnerability in structure, that is exploited by Mujkanovic et al. (2022) with adaptive adversarial attacks, to cause a catastrophic failure in SVD-GCN. This demonstrates that NSA can be used to interpret and identify the source of discrepancies and can unveil concealed weaknesses in defense methods not immediately apparent from conventional metrics.

## 5 CONCLUSION AND LIMITATIONS

We have demonstrated that the proposed measure of NSA is simple, efficient, and useful across many aspects of analysis and synthesis of representation spaces: robustness across initializations, convergence across epochs, autoencoders, adversarial attacks, and transfer learning. Its computational efficiency and the ability to converge to its global value in mini batches allow it to be useful in many applications and generalizations where scalability is desired.

While NSA demonstrates substantial benefits in various applications, certain limitations must be acknowledged to provide a comprehensive understanding of its scope and potential areas for future improvements. The choice of Euclidean distance as the primary metric for NSA may not be optimal in high-dimensional spaces due to the curse of dimensionality (Bellman, 1961). Despite this, Euclidean distance remains prevalent in various domains, including contrastive (Chen et al., 2020) or triplet losses (Schroff et al., 2015), style transfer (Johnson et al., 2016) and similarity indices. NSA's versatility across different applications is a significant strength, yet it is not a universal solution suitable for all scenarios without modification. Finally, NSA is primarily a structural similarity metric, which means that it focuses on preserving the geometrical configuration of data points rather than their functional equivalency. This focus can lead to scenarios where NSA indicates significant differences between datasets that are functionally similar but structurally distinct.

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

# A APPENDIX

## A PROOFS FOR GNSA AS A PSEUDOMETRIC

**Lemma A.1** (Identity). *Let $X$ be a point cloud over some space, then $\mathsf{GNSA}(X,X) = 0$.*

*Proof.* Let $X = \{x_1, \ldots, x_N\}$. Let $\max_{x \in X}(d(x,0)) = D$. Then

$$\mathsf{GNSA}(X,X) = \frac{1}{N^2} \sum_{1 \le i,j \le N} \left| \frac{d(x_i, x_j)}{D} - \frac{d(x_i, x_j)}{D} \right|.$$

Simplifying, we get

$$\mathsf{GNSA}(X,X) = \frac{1}{N^2 D} \sum_{1 \le i,j \le N} |d(x_i, x_j) - d(x_i, x_j)| = 0.$$

$\square$

**Lemma A.2** (Symmetry). *Let $X, Y$ be two point clouds of the same size, then $\mathsf{GNSA}(X,Y) = \mathsf{GNSA}(Y,X)$.*

*Proof.* Let $X = \{x_1, \ldots, x_N\}$ and $Y = \{y_1, \ldots, y_N\}$. Let $\max_{x \in X}(d(x,0)) = D_X$ and $\max_{y \in Y}(d(y,0)) = D_Y$. Then

$$\mathsf{GNSA}(X,Y) = \frac{1}{N^2} \sum_{1 \le i,j \le N} \left| \frac{d(x_i, x_j)}{D_X} - \frac{d(y_i, y_j)}{D_Y} \right|.$$

Since $|a - b| = |b - a|$,

$$\mathsf{GNSA}(X,Y) = \frac{1}{N^2} \sum_{1 \le i,j \le N} \left| \frac{d(y_i, y_j)}{D_Y} - \frac{d(x_i, x_j)}{D_X} \right|.$$

Hence, $\mathsf{GNSA}(X,Y) = \mathsf{GNSA}(Y,X)$.

$\square$

**Lemma A.3** (Non-negativity). *Let $X, Y$ be two point clouds of the same size, then $\mathsf{GNSA}(X,Y) \ge 0$.*

*Proof.* Let $X = \{x_1, \ldots, x_N\}$ and $Y = \{y_1, \ldots, y_N\}$. Let $\max_{x \in X}(d(x,0)) = D_X$ and $\max_{y \in Y}(d(y,0)) = D_Y$. Then

$$\mathsf{GNSA}(X,Y) = \frac{1}{N^2} \sum_{1 \le i,j \le N} \left| \frac{d(x_i, x_j)}{D_X} - \frac{d(y_i, y_j)}{D_Y} \right|.$$

Since, for all $i, j$,

$$\left| \frac{d(x_i, x_j)}{D_X} - \frac{d(y_i, y_j)}{D_Y} \right| \ge 0,$$

we get $\mathsf{GNSA}(X,Y) \ge 0$.

$\square$

**Lemma A.4** (Triangle inequality). *Let $X, Y$ and $Z$ be three point clouds of the same size, then $\mathsf{GNSA}(X,Z) \le \mathsf{GNSA}(X,Y) + \mathsf{GNSA}(Y,Z)$.*

*Proof.* Let $X = \{x_1, \ldots, x_N\}$, $Y = \{y_1, \ldots, y_N\}$ and $Z = \{z_1, \ldots, z_N\}$. Let $\max_{x \in X}(d(x,0)) = D_X$, $\max_{y \in Y}(d(y,0)) = D_Y$ and $\max_{z \in Z}(d(z,0)) = D_Z$. Then

$$\mathsf{GNSA}(X,Z) = \frac{1}{N^2} \sum_{1 \le i,j \le N} \left| \frac{d(x_i, x_j)}{D_X} - \frac{d(z_i, z_j)}{D_Z} \right|.$$

For each $i, j$, add and subtract $\frac{d(y_i, y_j)}{D_Y}$, then

$$\mathsf{GNSA}(X,Z) = \frac{1}{N^2} \sum_{1 \le i,j \le N} \left| \left( \frac{d(x_i, x_j)}{D_X} - \frac{d(y_i, y_j)}{D_Y} \right) + \left( \frac{d(y_i, y_j)}{D_Y} - \frac{d(z_i, z_j)}{D_Z} \right) \right|.$$

Since $|a + b| \leq |a| + |b|$,

$$\mathsf{GNSA}(X,Z) \leq \frac{1}{N^2} \sum_{1 \leq i,j \leq N} \left( \left| \frac{d(x_i,x_j)}{D_X} - \frac{d(y_i,y_j)}{D_Y} \right| + \left| \frac{d(y_i,y_j)}{D_Y} - \frac{d(z_i,z_j)}{D_Z} \right| \right).$$

Hence,

$$\mathsf{GNSA}(X,Z) \leq \frac{1}{N^2} \sum_{1 \leq i,j \leq N} \left| \frac{d(x_i,x_j)}{D_X} - \frac{d(y_i,y_j)}{D_Y} \right| + \frac{1}{N^2} \sum_{1 \leq i,j \leq N} \left| \frac{d(y_i,y_j)}{D_Y} - \frac{d(z_i,z_j)}{D_Z} \right|,$$

or $\mathsf{GNSA}(X,Z) \leq \mathsf{GNSA}(X,Y) + \mathsf{GNSA}(Y,Z)$. $\qquad\square$

From Lemma A.1, Lemma A.2, Lemma A.3 and Lemma A.4, we see that GNSA is a psuedometric over the space of point clouds.

## B  PROOFS FOR GNSA AS A SIMILARITY METRIC

### B.1  INVARIANCE TO ISOTROPIC SCALING

**Lemma B.1** (Invariance to Isotropic scaling in the first coordinate). *Let $X$ and $Y$ be two point clouds of the same size. Let $c \in \mathbf{R}$ and $c \neq 0$, $X_c$ be the point cloud with each point in $X$ scaled by a factor of $c$, then $\mathsf{GNSA}(X,Y) = \mathsf{GNSA}(X_c,Y)$.*

*Proof.* Let $X = \{x_1, \ldots, x_N\}$ and $Y = \{y_1, \ldots, y_N\}$, then $X_c = \{cx_1, \ldots, cx_N\}$. Let $\max_{x \in X}(d(x,0)) = D_X$, $\max_{y \in Y}(d(y,0)) = D_Y$ and $\max_{x \in X_c}(d(x,0)) = D_{X_c}$. Since, each point in $X_c$ is the $c$ times each point in $X$. Then, we can write $D_{X_c} = \max_{x \in X}(d(cx,0))$. Since, for any two points, $x_1$ and $x_2$, $d(cx_1, cx_2) = |c|d(x_1,x_2)$, then $D_{X_c} = |c| \max_{x \in X}(d(x,0))$. Hence, $D_{x_c} = |c|D_X$. From the definition of GNSA,

$$\mathsf{GNSA}(X_c,Y) = \frac{1}{N^2} \sum_{1 \leq i,j \leq N} \left| \left( \frac{d(cx_i,cx_j)}{D_{X_c}} - \frac{d(y_i,y_j)}{D_Y} \right) \right|.$$

Again, using $d(cx_i,cx_j) = |c|d(x_i,x_j)$ and $D_{X_c} = |c|D_X$,

$$\mathsf{GNSA}(X_c,Y) = \frac{1}{N^2} \sum_{1 \leq i,j \leq N} \left| \left( \frac{|c|d(x_i,x_j)}{|c|D_X} - \frac{d(y_i,y_j)}{D_Y} \right) \right|.$$

Simplifying,

$$\mathsf{GNSA}(X_c,Y) = \frac{1}{N^2} \sum_{1 \leq i,j \leq N} \left| \left( \frac{d(x_i,x_j)}{D_X} - \frac{d(y_i,y_j)}{D_Y} \right) \right|.$$

Hence, $\mathsf{GNSA}(X_c,Y) = \mathsf{GNSA}(X,Y)$. $\qquad\square$

By Lemma A.2, we can see that this gives us invariance to isotropic scaling in the second coordinate as well.

**Lemma B.2** (Invariance to Isotropic scaling in the second coordinate). *Let $X$ and $Y$ be two point clouds of the same size. Let $c \in \mathbf{R}$ and $c \neq 0$, $Y_c$ be the point cloud with each point in $Y$ scaled by a factor of $c$, then $\mathsf{GNSA}(X,Y) = \mathsf{GNSA}(X,Y_c)$.*

Combining Lemma B.1 and *Lemma B*.2, we get the required lemma.

**Lemma B.3** (Invariance to Isotropic scaling). *Let $X$ and $Y$ be two point clouds of the same size. Let $c_1, c_2 \in \mathbf{R}$, $c_1 \neq 0$ and $c_2 \neq 0$, $X_{c_1}$ be the point cloud with each point in $X$ scaled by a factor of $c_1$ and $Y_{c_2}$ be the point cloud with each point in $Y$ scaled by a factor of $c_2$, then $\mathsf{GNSA}(X,Y) = \mathsf{GNSA}(X_{c_1}, Y_{c_2})$.*

## B.2 Invariance to Orthogonal transformation

**Lemma B.4** (Invariance to Orthogonal transformation in the first coordinate). *Let $X$ and $Y$ be two point clouds of the same size. Let $U$ be an orthogonal transformation on the point space of $X$, $X_U$ be the point cloud with each point in $X$ transformed using $U$, then $\mathsf{GNSA}(X, Y) = \mathsf{GNSA}(X_U, Y)$.*

*Proof.* Let $X = \{x_1, \ldots, x_N\}$ and $Y = \{y_1, \ldots, y_N\}$, then $X_U = \{Ux_1, \ldots, Ux_N\}$. Let $\max_{x \in X}(d(x, 0)) = D_X$, $\max_{y \in Y}(d(y, 0)) = D_Y$ and $\max_{x \in X_U}(d(x, 0)) = D_{X_U}$. Since, each point in $X_U$ is the $U$ times each point in $X$. Then, we can write $D_{X_U} = \max_{x \in X}(d(Ux, 0))$. Since, for any two points, $x_1$ and $x_2$, $d(Ux_1, x_2) = d(x_1, U^T x_2)$, and $U^T 0 = 0$, then $D_{X_c} = \max_{x \in X}(d(x, 0))$. Hence, $D_{x_c} = D_X$. From the definition of $\mathsf{GNSA}$,

$$\mathsf{GNSA}(X_U, Y) = \frac{1}{N^2} \sum_{1 \leq i, j \leq N} \left| \left( \frac{d(Ux_i, Ux_j)}{D_{X_c}} - \frac{d(y_i, y_j)}{D_Y} \right) \right|.$$

Again, using $d(Ux_i, Ux_j) = d(x_i, U^T U x_j)$, and $U^T U = I$,

$$\mathsf{GNSA}(X_U, Y) = \frac{1}{N^2} \sum_{1 \leq i, j \leq N} \left| \left( \frac{d(x_i, x_j)}{D_X} - \frac{d(y_i, y_j)}{D_Y} \right) \right|.$$

Hence, $\mathsf{GNSA}(X_U, Y) = \mathsf{GNSA}(X, Y)$. $\qquad\square$

By Lemma A.2, we get invariance to orthogonal transformation in the second coordinate too and combining, we get the required lemma.

**Lemma B.5** (Invariance to Orthogonal transformation). *Let $X$ and $Y$ be two point clouds of the same size. Let $U_1$ be an orthogonal transformation on the point space of $X$, $X_{U_1}$ be the point cloud with each point in $X$ transformed using $U_1$. Similarly, let $U_2$ be an orthogonal transformation on the point space of $Y$, $Y_{U_2}$ be the point cloud with each point in $Y$ transformed using $U_2$, then $\mathsf{GNSA}(X, Y) = \mathsf{GNSA}(X_{U_1}, Y_{U_2})$.*

## B.3 Not Invariant under Invertible Linear Transformation (ILT)

We can easily see this with a counter example. Let $X = \{(1, 0, 0), (0, 1, 0), (0, 0, 1)\}$. Let $Y = \{(1, 1, 0), (1, 0, 1), (0, 1, 1)\}$. Define $A$ to be an invertible linear map such that $A(1, 0, 0) = (2, 0, 0)$, $A(0, 1, 0) = (0, 1, 0)$ and $A(0, 0, 1) = (0, 0, 1)$. Let $X_A$ be the point cloud with each point in $X$ transformed using $A$. Then from some calculation, we can see that $\mathsf{GNSA}(X, Y) = 0$ but $\mathsf{GNSA}(X_A, Y) = \frac{1}{9}$. Hence, we can see that GNSA is not invariant under Invertible Linear Transformations.

## C Proofs for GNSA as a Loss Function

### C.1 Differentiability of GNSA

To derive the sub-gradient $\frac{\partial \mathsf{GNSA}(X, Y)}{\partial x_i}$, we first define the following notation. Let $\mathbb{I} : \{T, F\} \to \{0, 1\}$ be a function that takes as input some condition and outputs 0 or 1 such that $\mathbb{I}(T) = 1$ and $\mathbb{I}(F) = 0$. Let $D_X = \max_{x \in X}(d(x, 0))$. Then it is easy to see that

$$\frac{\partial D_X}{\partial x_i} = \frac{x_i}{d(x_i, 0)} \mathbb{I}\{\arg\max_{x \in X}(d(x, 0)) = i\}.$$

Also, notice that $\frac{\partial d(x_i, x_j)}{x_i} = \frac{x_i - x_j}{d(x_i, x_j)}$, and that for $j \neq i$ and $k \neq i$, $\frac{\partial d(x_j, x_k)}{x_i} = 0$.

Let $m_X^{i,j} = \frac{d(x_i, x_j)}{D_X}$ and $m_Y^{i,j} = \frac{d(y_i, y_j)}{D_Y}$. Hence, combining, we get

$$\frac{\partial m_X^{j,k}}{\partial x_i} = \frac{\frac{\partial d(x_j, x_k)}{\partial x_i}}{D_X} - \frac{d(x_j, x_k) \frac{\partial D_X}{\partial x_i}}{(D_X)^2}, \text{ where these partial differentials are derived above.}$$

Lastly, notice that $\frac{\partial |m_X^{j,k} - m_Y^{j,k}|}{\partial m_X^{j,k}} = (-1)^{\mathbb{I}(m_Y^{j,k} > m_X^{j,k})}$.

Combining all the above, we get

$$\frac{\partial \mathsf{GNSA}(X,Y)}{\partial x_i} = \frac{1}{N^2} \sum_{1 \leq j,k \leq N} \frac{\partial \left| m_X^{j,k} - m_Y^{j,k} \right|}{\partial m_X^{j,k}} \frac{\partial m_X^{j,k}}{\partial x_i} \quad = \frac{1}{N^2} \sum_{1 \leq j,k \leq N} (-1)^{\mathbb{I}(m_Y^{j,k} > m_X^{j,k})} \frac{\partial m_X^{j,k}}{\partial x_i}.$$

Sub-gradients $\frac{\partial \mathsf{GNSA}(X,Y)}{\partial y_i}$ can be similarly derived.

## C.2 CONTINUITY

We prove the continuity of GNSA by showing that GNSA is a composition of continuous functions (Carothers, 2000, Theorem 5.10). Let

$$f_{i,j}(X,Y) = \left| \frac{d(x_i, x_j)}{\max_{x \in X}(d(x,0))} - \frac{d(y_i, y_j)}{\max_{y \in Y}(d(y,0))} \right|.$$

It's easy to see that $\qquad \mathsf{GNSA}(X,Y) = \frac{1}{N^2} \sum_{1 \leq i,j \leq N} f_{i,j}(X,Y).$

Since a sum of continuous functions is continuous, all we need to show is that $f_{i,j}(X,Y)$ is continuous for all $1 \leq i, j \leq N$. This is easy to see because $f_{i,j}(X,Y)$ can be seen as a composition of $d(\cdot, \cdot)$ and $\max(\cdot)$, along with the algebraic operations of subtraction, division and taking the absolute value. All the above operations are continuous everywhere[5], we get that their composition is also continuous everywhere. Hence, $\mathsf{GNSA}(\cdot, \cdot)$ is continuous.

## C.3 MOTIVATION FOR GNSA

The global part of NSA is inspired by Representational Similarity Matrix (RSM)-Based Measures (Klabunde et al., 2023). RSMs describe the similarity of the representation of each instance to all other instances within a given representation. The elements of the RSM $S_{i,j}$ can be defined using a valid distance or similarity function, formally: $S_{i,j} := s(R_i, R_j)$.

The motivation behind RSM-based measures is to capture the relational structure in the data. By comparing the pairwise similarities of instances within two different representations, RSM-based measures provide insights into how the internal structure of representations changes. RSMs are invariant to orthogonal transformations, which means they retain their structure under rotations and reflections. When using euclidean distance as a similarity function, the RSM is invariant under euclidean group $E(n)$, assuming translations.

One might think that the most intuitive approach to compare RSMs is to use a matrix norm. This method works well when the data is similarly scaled, as it is zero when the data is identical and larger when the data is dissimilar. However, it is not invariant to scaling, which can cause issues if the data is not similarly scaled. To ensure the data is on a similar scale, we first normalize the data using the normalization function:

$$R_{norm} = \frac{R}{2 * max(||x_i||)}$$

Where $x_i$ is the point farthest from the origin in the space. One of the primary reasons for our metric's effectiveness is this normalization scheme. Although other normalizations are possible, NSA performs best when we normalize the space $X$ by twice the distance between the origin of the space and the farthest point in the space. This ensures that all the points are within a unit sphere in $N$ dimensions where $N$ is the dimensionality of the space. The origin of the space is the point in $\mathbb{R}^N$ around which all points belonging to $X$ are centered.

Most popular similarity measures, such as CKA(Kornblith et al., 2019) and RTD(Barannikov et al., 2022) also generate a similarity matrix. GNSA is simple in its implementation, but our choice of distance function and unique normalization scheme make GNSA a pseudometric, a similarity index and a valid, differentiable loss function.

---

[5]Note that division is not continuous when the denominator goes to zero. Since, the denominators are $\max_{x \in X}(d(x,0))$ and $\max_{y \in Y}(d(y,0))$, as long as neither of the representations is all zero, the denominator does not go to zero.

## C.4 IMPROVING ROBUSTNESS OF NSA

Formally, NSA is normalized by the point that is furthest away from the origin for the representation space. In practice, NSA yields more robust results and exhibits reduced susceptibility to outliers when distances are normalized relative to a quantile of the distances from the origin (e.g., 0.98 quantile). This quantile-based approach ensures that the distances are adjusted proportionally, taking into account the spread of values among the points with respect to their distances from the origin. It also ensures that any outliers do not affect the rescaling of the point cloud to the unit space.

# D  PROOFS FOR LNSA AS A PREMETRIC

**Lemma D.1** (Identity). *Let $X$ be a point cloud over some space, then* $\mathsf{LNSA}^{\mathsf{metric}}(X, X) = 0$.

*Proof.* Let $X = \{\vec{x}_1, \ldots, \vec{x}_N\}$. Hence, we see that

$$\mathsf{LNSA}^{\mathsf{metric}}(X, X) = \mathsf{LNSA}_X(X) + \mathsf{LNSA}_X(X) = 2\mathsf{LNSA}_X(X).$$

We show that $\mathsf{LNSA}_X(X) = 0$. We see that

$$\mathsf{LNSA}_X(X) = \frac{1}{N} \sum_{1 \le i \le N} \left( \mathsf{lid}(i, X)^{-1} - \overline{\mathsf{lid}}_X(i, X)^{-1} \right)^2.$$

Note that

$$\overline{\mathsf{lid}}_X(i, X) = \mathsf{AD}_X(i, kNN_X(i)) = \mathsf{lid}(i, X).$$

Hence we get

$$\mathsf{LNSA}_X(X) = \frac{1}{N} \sum_{1 \le i \le N} \left( \mathsf{lid}(i, X)^{-1} - \mathsf{lid}(i, X)^{-1} \right)^2 = 0.$$

And hence we have

$$\mathsf{LNSA}^{\mathsf{metric}}(X, X) = 2\mathsf{LNSA}_X(X) = 0.$$

$\square$

**Lemma D.2** (Symmetry). *Let $X, Y$ be two point clouds of the same size, then* $\mathsf{LNSA}^{\mathsf{metric}}(X, Y) = \mathsf{LNSA}^{\mathsf{metric}}(Y, X)$.

*Proof.* Let $X = \{x_1, \ldots, x_N\}$ and $Y = \{y_1, \ldots, y_N\}$. Then

$$\mathsf{LNSA}^{\mathsf{metric}}(X, Y) = \mathsf{LNSA}_X(Y) + \mathsf{LNSA}_Y(X) = \mathsf{LNSA}^{\mathsf{metric}}(Y, X).$$

Hence,

$$\mathsf{LNSA}^{\mathsf{metric}}(X, Y) = \mathsf{LNSA}^{\mathsf{metric}}(Y, X).$$

$\square$

**Lemma D.3** (Non-negativity). *Let $X, Y$ be two point clouds of the same size, then* $\mathsf{LNSA}^{\mathsf{metric}}(X, Y) \ge 0$.

*Proof.* Let $X = \{x_1, \ldots, x_N\}$ and $Y = \{y_1, \ldots, y_N\}$. Note that

$$\mathsf{LNSA}^{\mathsf{metric}}(X, Y) = \mathsf{LNSA}_X(Y) + \mathsf{LNSA}_Y(X),$$

and

$$\mathsf{LNSA}_X(Y) = \frac{1}{N} \sum_{1 \le i \le N} \left( \mathsf{lid}(i, X)^{-1} - \overline{\mathsf{lid}}_X(i, Y)^{-1} \right)^2.$$

Since, for all $i$, $\left( \mathsf{lid}(i, X)^{-1} - \overline{\mathsf{lid}}_X(i, Y)^{-1} \right)^2 \ge 0$, hence we get $\mathsf{LNSA}_X(Y) \ge 0$. Similarly, $\mathsf{LNSA}_Y(X) \ge 0$ and hence, $\mathsf{LNSA}^{\mathsf{metric}}(X, Y) \ge 0$. $\square$

From Lemma D.1, Lemma D.2 and Lemma D.3, we see that $\mathsf{LNSA}^{\mathsf{metric}}$ is a premetric over the space of point clouds.

# E   PROOFS FOR LNSA AS A SIMILARITY METRIC

## E.1   INVARIANCE TO ISOTROPIC SCALING

**Lemma E.1** (Invariance to Isotropic scaling in the first coordinate). *Let $X$ and $Y$ be two point clouds of the same size. Let $c \in \mathbf{R}$ and $c \neq 0$, $X_c$ be the point cloud with each point in $X$ scaled by a factor of $c$, then $\mathsf{LNSA}^{\mathsf{metric}}(X, Y) = \mathsf{LNSA}^{\mathsf{metric}}(X_c, Y)$.*

*Proof.* Let $X = \{\vec{x}_1, \ldots, \vec{x}_N\}$ and $Y = \{\vec{y}_1, \ldots, \vec{y}_N\}$, then $X_c = \{c\vec{x}_1, \ldots, c\vec{x}_N\}$. We show that $\mathsf{AD}_{X_c}(i, (i_1, \ldots, i_k)) = \mathsf{AD}_X(i, (i_1, \ldots, i_k))$. Note that

$$\mathsf{AD}_{X_c}(i, (i_1, \ldots, i_k)) = -\left(\frac{1}{k}\sum_{j=1}^{k}\log\left(\frac{d(c\vec{x}_i, c\vec{x}_{i_j})}{d(c\vec{x}_i, c\vec{x}_{i_k})}\right)\right)^{-1}.$$

Since, $\frac{d(c\vec{x}_i, c\vec{x}_{i_j})}{d(c\vec{x}_i, c\vec{x}_{i_k})} = \frac{cd(\vec{x}_i, \vec{x}_{i_j})}{cd(\vec{x}_i, \vec{x}_{i_k})} = \frac{d(\vec{x}_i, \vec{x}_{i_j})}{d(\vec{x}_i, \vec{x}_{i_k})}$. Hence,

$$\mathsf{AD}_{X_c}(i, (i_1, \ldots, i_k)) = -\left(\frac{1}{k}\sum_{j=1}^{k}\log\left(\frac{d(\vec{x}_i, \vec{x}_{i_j})}{d(\vec{x}_i, \vec{x}_{i_k})}\right)\right)^{-1}.$$

Hence, $\mathsf{AD}_{X_c}(i, (i_1, \ldots, i_k)) = \mathsf{AD}_X(i, (i_1, \ldots, i_k))$. Hence, we get $\mathsf{lid}(\cdot, X_c) = \mathsf{lid}(\cdot, X)$. Also note that since $kNN_{X_c}(\cdot) = kNN_X(\cdot)$, we have that $\overline{\mathsf{lid}}_{X_c}(\cdot, Y) = \overline{\mathsf{lid}}_X(\cdot, Y)$. Hence, we get $\mathsf{LNSA}^{\mathsf{metric}}(X, Y) = \mathsf{LNSA}^{\mathsf{metric}}(X_c, Y)$. $\square$

By Lemma D.2, we can see that this gives us invariance to isotropic scaling in the second coordinate as well.

**Lemma E.2** (Invariance to Isotropic scaling in the second coordinate). *Let $X$ and $Y$ be two point clouds of the same size. Let $c \in \mathbf{R}$ and $c \neq 0$, $Y_c$ be the point cloud with each point in $Y$ scaled by a factor of $c$, then $\mathsf{LNSA}^{\mathsf{metric}}(X, Y) = \mathsf{LNSA}^{\mathsf{metric}}(X, Y_c)$.*

Combining Lemma E.1 and *Lemma E*.2, we get the required lemma.

**Lemma E.3** (Invariance to Isotropic scaling). *Let $X$ and $Y$ be two point clouds of the same size. Let $c_1, c_2 \in \mathbf{R}$, $c_1 \neq 0$ and $c_2 \neq 0$, $X_{c_1}$ be the point cloud with each point in $X$ scaled by a factor of $c_1$ and $Y_{c_2}$ be the point cloud with each point in $Y$ scaled by a factor of $c_2$, then $\mathsf{LNSA}^{\mathsf{metric}}(X, Y) = \mathsf{LNSA}^{\mathsf{metric}}(X_{c_1}, Y_{c_2})$.*

## E.2   INVARIANCE TO ORTHOGONAL TRANSFORMATION

**Lemma E.4** (Invariance to Orthogonal transformation in the first coordinate). *Let $X$ and $Y$ be two point clouds of the same size. Let $U$ be an orthogonal transformation on the point space of $X$, $X_U$ be the point cloud with each point in $X$ transformed using $U$, then $\mathsf{LNSA}^{\mathsf{metric}}(X, Y) = \mathsf{LNSA}^{\mathsf{metric}}(X_U, Y)$.*

*Proof.* Let $X = \{\vec{x}_1, \ldots, \vec{x}_N\}$ and $Y = \{\vec{y}_1, \ldots, \vec{y}_N\}$, then $X_U = \{U\vec{x}_1, \ldots, U\vec{x}_N\}$. We show that $\mathsf{AD}_{X_U}(i, (i_1, \ldots, i_k)) = \mathsf{AD}_X(i, (i_1, \ldots, i_k))$. Note that

$$\mathsf{AD}_{X_c}(i, (i_1, \ldots, i_k)) = -\left(\frac{1}{k}\sum_{j=1}^{k}\log\left(\frac{d(U\vec{x}_i, U\vec{x}_{i_j})}{d(U\vec{x}_i, U\vec{x}_{i_k})}\right)\right)^{-1}.$$

Since, $\frac{d(U\vec{x}_i, U\vec{x}_{i_j})}{d(U\vec{x}_i, U\vec{x}_{i_k})} = \frac{d(U^T U\vec{x}_i, \vec{x}_{i_j})}{d(U^T U\vec{x}_i, \vec{x}_{i_k})} = \frac{d(\vec{x}_i, \vec{x}_{i_j})}{d(\vec{x}_i, \vec{x}_{i_k})}$. Hence,

$$\mathsf{AD}_{X_c}(i, (i_1, \ldots, i_k)) = -\left(\frac{1}{k}\sum_{j=1}^{k}\log\left(\frac{d(\vec{x}_i, \vec{x}_{i_j})}{d(\vec{x}_i, \vec{x}_{i_k})}\right)\right)^{-1}.$$

Hence, $\mathsf{AD}_{X_U}(i, (i_1, \ldots, i_k)) = \mathsf{AD}_X(i, (i_1, \ldots, i_k))$. Hence, we get $\mathsf{lid}(\cdot, X_U) = \mathsf{lid}(\cdot, X)$. Also note that since $kNN_{X_U}(\cdot) = kNN_X(\cdot)$, we have that $\overline{\mathsf{lid}}_{X_U}(\cdot, Y) = \overline{\mathsf{lid}}_X(\cdot, Y)$. Hence, we get $\mathsf{LNSA}^{\mathsf{metric}}(X, Y) = \mathsf{LNSA}^{\mathsf{metric}}(X_U, Y)$. $\qquad\square$

By Lemma D.2, we get invariance to orthogonal transformation in the second coordinate too and combining, we get the required lemma.

**Lemma E.5** (Invariance to Orthogonal transformation). *Let $X$ and $Y$ be two point clouds of the same size. Let $U_1$ be an orthogonal transformation on the point space of $X$, $X_{U_1}$ be the point cloud with each point in $X$ transformed using $U_1$. Similarly, let $U_2$ be an orthogonal transformation on the point space of $Y$, $Y_{U_2}$ be the point cloud with each point in $Y$ transformed using $U_2$, then $\mathsf{LNSA}^{\mathsf{metric}}(X, Y) = \mathsf{LNSA}^{\mathsf{metric}}(X_{U_1}, Y_{U_2})$.*

### E.3 Not Invariant under Invertible Linear Transformation (ILT)

We can easily see this with a counter example. Let $X = \{(1, 0, 0), (0, 1, 0), (0, 0, 1)\}$. Let $Y = \{(1, 1, 0), (1, 0, 1), (0, 1, 1)\}$. Define $A$ as an invertible linear map such that $A(1, 0, 0) = (2, 0, 0)$, $A(0, 1, 0) = (0, 1, 0)$ and $A(0, 0, 1) = (0, 0, 1)$. Let $X_A$ be the point cloud with each point in $X$ transformed using $A$. Then from some calculations we can see that $\mathsf{LNSA}^{\mathsf{metric}}(X, Y) = 0$ but $\mathsf{LNSA}^{\mathsf{metric}}(X_A, Y) = \frac{1}{12} \left( \log(5) - \log(2) \right)^2$. Hence, we can see that GNSA is not invariant under Invertible Linear Transformations.

## F Proofs for LNSA as a Loss Function

### F.1 Differentiability of LNSA

To derive the sub-gradient $\frac{\partial \mathsf{LNSA}_Y(X)}{\partial x_i}$, we first define the following notation.

Let

$$f_{j,Y}(X) = \left( \mathsf{lid}(j, Y)^{-1} - \overline{\mathsf{lid}}_Y(j, X)^{-1} \right)^2.$$

$$\text{Using this,} \qquad \mathsf{LNSA}_Y(X) = \frac{1}{N} \sum_{1 \le j \le N} f_{j,Y}(X).$$

Hence,

$$\frac{\partial \mathsf{LNSA}_Y(X)}{\partial x_i} = \frac{1}{N} \sum_{1 \le j \le N} \frac{\partial f_{j,Y}(X)}{\partial \vec{x}_i}.$$

Note that $\mathsf{lid}(j, Y)^{-1}$ is constant with respect to $\vec{x}_i$, hence,

$$\frac{\partial f_{j,Y}(X)}{\partial \vec{x}_i} = -2 \left( \mathsf{lid}(j, Y)^{-1} - \overline{\mathsf{lid}}_Y(j, X)^{-1} \right) \frac{\partial \overline{\mathsf{lid}}_Y(j, X)^{-1}}{\partial \vec{x}_i}.$$

To compute $\frac{\partial \overline{\mathsf{lid}}_Y(j, X)^{-1}}{\partial \vec{x}_i}$, we divide this into the four cases below:

- $i = j$: Let $kNN_Y(i) = (i_1, \cdot, i_k)$, then $\overline{\mathsf{lid}}_Y(i, X)^{-1} = \sum_{l=1}^k \log \left( \frac{d(\vec{x}_i, \vec{x}_{i_l})}{d(\vec{x}_i, \vec{x}_{i_k})} \right)$. Hence,

$$\frac{\partial \overline{\mathsf{lid}}_Y(i, X)^{-1}}{\partial \vec{x}_i} = \sum_{l=1}^k \left( \frac{1}{d(\vec{x}_i, \vec{x}_{i_l})} \frac{\partial d(\vec{x}_i, \vec{x}_{i_l})}{\partial \vec{x}_i} - \frac{1}{d(\vec{x}_i, \vec{x}_{i_k})} \frac{\partial d(\vec{x}_i, \vec{x}_{i_k})}{\partial \vec{x}_i} \right).$$

- $j \in kNN_Y(i)$ **and** $j$ **is not the last element of** $kNN_Y(i)$: Let $kNN_Y(i) = (i_1, \cdot, i_k)$, then $\overline{\mathsf{lid}}_Y(i, X)^{-1} = \sum_{l=1}^k \log \left( \frac{d(\vec{x}_i, \vec{x}_{i_l})}{d(\vec{x}_i, \vec{x}_{i_k})} \right)$. Hence,

$$\frac{\partial \overline{\mathsf{lid}}_Y(i, X)^{-1}}{\partial \vec{x}_i} = \frac{1}{d(\vec{x}_i, \vec{x}_j)} \frac{\partial d(\vec{x}_i, \vec{x}_j)}{\partial \vec{x}_i}.$$

- $j \in kNN_Y(i)$ **and** $j$ **is the last element of** $kNN_Y(i)$: Let $kNN_Y(i) = (i_1, \cdot, i_k)$, then $\overline{\mathsf{lid}}_Y(i, X)^{-1} = \sum_{l=1}^{k} \log \left( \frac{d(\vec{x}_i, \vec{x}_{i_l})}{d(\vec{x}_i, \vec{x}_{i_k})} \right)$. Hence,

$$\frac{\partial \overline{\mathsf{lid}}_Y(i, X)^{-1}}{\partial \vec{x}_i} = (1 - k) \left( \frac{1}{d(\vec{x}_i, \vec{x}_j)} \frac{\partial d(\vec{x}_i, \vec{x}_j)}{\partial \vec{x}_i} \right).$$

- $j \notin kNN_Y(i)$ **and** $j \neq i$: Then

$$\frac{\partial \overline{\mathsf{lid}}_Y(i, X)^{-1}}{\partial \vec{x}_i} = 0.$$

Hence, combining the above, we have the sub-gradients of $\mathsf{LNSA}_Y(X)$.

## F.2 CONTINUITY

We prove the continuity of $\mathsf{LNSA}^{\mathsf{metric}}$ by showing that $\mathsf{LNSA}^{\mathsf{metric}}$ is a composition of continuous functions (Carothers, 2000, Theorem 5.10). Let

$$f_{i,Y}(X) = \left( \mathsf{lid}(i, Y)^{-1} - \overline{\mathsf{lid}}_Y(i, X)^{-1} \right)^2.$$

Using this, $\quad \mathsf{LNSA}_Y(X) = \frac{1}{N} \sum_{1 \leq i \leq N} f_{i,Y}(X).$

Since a sum of continuous functions is continuous, all we need to show is that $f_{i,Y}(X)$ is continuous for all $1 \leq i \leq N$. Let $kNN_Y(i) = (i_1, \cdot, i_k)$, then note that

$$f_{i,Y}(X) = \left( \sum_{l=1}^{k} \left( \log \left( \frac{d(\vec{x}_i, \vec{x}_{i_l})}{d(\vec{x}_i, \vec{x}_{i_k})} \right) - \log \left( \frac{d(\vec{y}_i, \vec{y}_{i_l})}{d(\vec{y}_i, \vec{y}_{i_k})} \right) \right) \right)^2,$$

hence can be seen as a composition of $d(\cdot, \cdot)$ and $\log(\cdot)$, along with the algebraic operations of addition, subtraction, multiplication and division. All the above operations are continuous everywhere[6], we get that their composition is also continuous everywhere. Hence, $\mathsf{LNSA}_Y(\cdot)$ is continuous.

## G CONVERGENCE OF SUBSET GNSA

Here, we prove that GNSA adapts well to minibatching and hence can be used as a loss term. In particular, Lemma G.1 proves theoretically that in expectation, GNSA over a minibatch is equal to GNSA of the whole dataset. This supplements the results presented in Figure 2. The experiment is run on the output embeddings of a GCN trained on node classification for the Cora Dataset. We compare the convergence with a batch size of 200 and 500.

**Lemma G.1** (Subset GNSA convergence). *Let $X, Y$ be two point clouds of the same size $N$ such that $X = \{x_1, \ldots, x_N\}$ and $Y = \{y_1, \ldots, y_N\}$. Let $\tilde{X}$ be some randomly sampled $s$ sized subsets of $X$ (for some $s \leq N$). Define $\tilde{Y} \subset Y$ as $y_i \in \tilde{Y}$ iff $x_i \in \tilde{X}$. Then the following holds true*

$$\mathbb{E}_{\tilde{X}, \tilde{Y}} \left[ \mathsf{GNSA}(\tilde{X}, \tilde{Y}) \right] = \mathsf{GNSA}(X, Y).$$

*Proof.* Let $\max_{x \in X}(d(x, 0)) = D_X$ and $\max_{y \in Y}(d(y, 0)) = D_Y$. Then

$$\mathsf{GNSA}(X, Y) = \frac{1}{N^2} \sum_{1 \leq i, j \leq N} \left| \frac{d(x_i, x_j)}{D_X} - \frac{d(y_i, y_j)}{D_Y} \right|.$$

Define $\mathbb{I}_{\tilde{X}}(x_i)$ as

$$\mathbb{I}_{\tilde{X}}(x_i) = \begin{cases} 1 \text{ if } x_i \in \tilde{X}, \\ 0 \text{ otherwise.} \end{cases}$$

---

[6]Note that division is not continuous when the denominator goes to zero. Since, the denominators are $(d(\vec{x}_i, \vec{x}_{i_k}))$, as long as all $k$ neighbours of a point are not zero distance away, the denominator does not go to zero.

Then we can see that

$$\mathsf{GNSA}(\tilde{X}, \tilde{Y}) = \frac{1}{s^2} \sum_{1 \le i,j \le N} \left| \frac{d(x_i, x_j)}{D_X} - \frac{d(y_i, y_j)}{D_Y} \right| \mathbb{I}_{\tilde{X}}(x_i) \, \mathbb{I}_{\tilde{X}}(x_j).$$

Hence, taking expectation over $\tilde{X}, \tilde{Y}$, we get

$$\mathop{\mathbb{E}}_{\tilde{X}, \tilde{Y}} \left[ \mathsf{GNSA}(\tilde{X}, \tilde{Y}) \right] = \frac{1}{s^2} \mathop{\mathbb{E}}_{\tilde{X}, \tilde{Y}} \left[ \sum_{1 \le i,j \le N} \left| \frac{d(x_i, x_j)}{D_X} - \frac{d(y_i, y_j)}{D_Y} \right| \mathbb{I}_{\tilde{X}}(x_i) \, \mathbb{I}_{\tilde{X}}(x_j) \right].$$

By linearity of expectation, we get

$$\mathop{\mathbb{E}}_{\tilde{X}, \tilde{Y}} \left[ \mathsf{GNSA}(\tilde{X}, \tilde{Y}) \right] = \frac{1}{s^2} \sum_{1 \le i,j \le N} \left| \frac{d(x_i, x_j)}{D_X} - \frac{d(y_i, y_j)}{D_Y} \right| \mathop{\mathbb{E}}_{\tilde{X}, \tilde{Y}} \left[ \mathbb{I}_{\tilde{X}}(x_i) \, \mathbb{I}_{\tilde{X}}(x_j) \right].$$

Assuming $s >> 1$, each $x_i$ is independently in $\tilde{X}$ with probability $s/N$, hence, we get

$$\mathop{\mathbb{E}}_{\tilde{X}, \tilde{Y}} \left[ \mathsf{GNSA}(\tilde{X}, \tilde{Y}) \right] = \frac{1}{s^2} \sum_{1 \le i,j \le N} \left| \frac{d(x_i, x_j)}{D_X} - \frac{d(y_i, y_j)}{D_Y} \right| \frac{s}{N} \frac{s}{N}.$$

Hence, we have

$$\mathop{\mathbb{E}}_{\tilde{X}, \tilde{Y}} \left[ \mathsf{GNSA}(\tilde{X}, \tilde{Y}) \right] = \mathsf{GNSA}(X, Y).$$

□

## H    NSA RUNNING TIME OVER INCREASING BATCH SIZES

Theoretically, NSA is quadratic in the number of points as we need to calculate pairwise distances. But since we use torch functions and the data is loaded on the GPU, empirical running times are significantly lower. Figure 5(a) presents the running time of GlobalNSA and Figure 5(a) presents the running time of NSA (Global and Local combined) on two embedding spaces of very high dimensionality ( 200,000). We notice that even when we get to a very high batch size (5000), the running time scaling is nearly linear. It is worth noting that these running times are higher as the dimensionality is very high, in most practical scenarios, your embeddings would lie in a lower dimension.

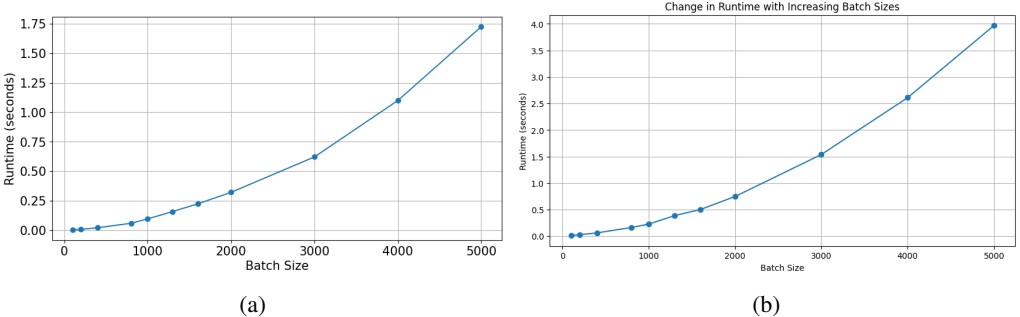

(a)                                        (b)

Figure 5: Running time of NSA over increasing batch sizes. (a) Running time of GlobalNSA over increasing batch size. (b) Running time of NSA over increasing batch size

## I    ADDITIONAL TESTS WITH CNNS

We perform additional layer-wise similarity tests and initial adversarial analysis on ResNet architectures and present the results here. Using ResNet-18 and ResNet-32 models pretrained on ImageNet, we extract intermediate embeddings from a subset of 100K images, focusing on post-residual layers

(end of blocks). This choice is supported by prior findings (Kornblith et al., 2019; Veit et al., 2016), which show that layers within blocks do not capture meaningful information correlating with other layers. Consistent with our other experiments, NSA is computed on a subset of 4,000 samples and averaged over 10 trials. The resulting similarity heatmap, shown in Figure 6, exhibits a clear layerwise pattern in both models. This reinforces NSA's ability to quantify dissimilarity across architectures, approximate large datasets, and scale efficiently.

In Section 4.4, we present results demonstrating NSA's correlation with traditional metrics in analyzing adversarial attacks. While our earlier experiments focus on GNN architectures, NSA is architecture-agnostic and can work with any model generating high-dimensional embeddings. It effectively quantifies the structural changes in representation spaces caused by adversarial perturbations.

To validate this, we perform a Projected Gradient Descent (PGD) evasion attack (Madry et al., 2019) on a ResNet-20 model pretrained on CIFAR-10, poisoning increasing percentages of the test data. NSA is computed between the embeddings of the original test set and the poisoned test set. Results, presented in Figure 7 (a), show a clean correlation between NSA and misclassification rate. We extend this analysis to a few more ResNet models, where we focus on interpretability. We present the NSA of the first 50 images computed individually in Figure 7 (b) . It is observed that any image that is perturbed demonstrates a large increase in NSA value upon being poisoned, indicating a notable shift in their positions within the embedding space. These findings demonstrate NSA's utility in analyzing, interpreting, and identifying the sources of perturbations in models, aligning with our earlier results on GNN architectures in Section 4.4.

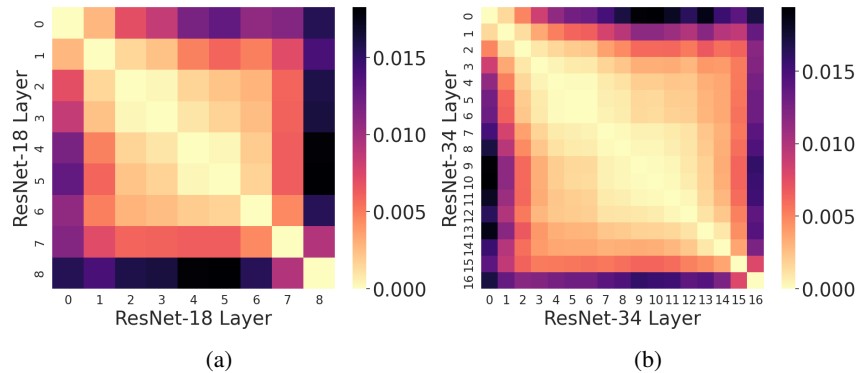

Figure 6: Layerwise Analysis of ResNet models trained on ImageNet with NSA

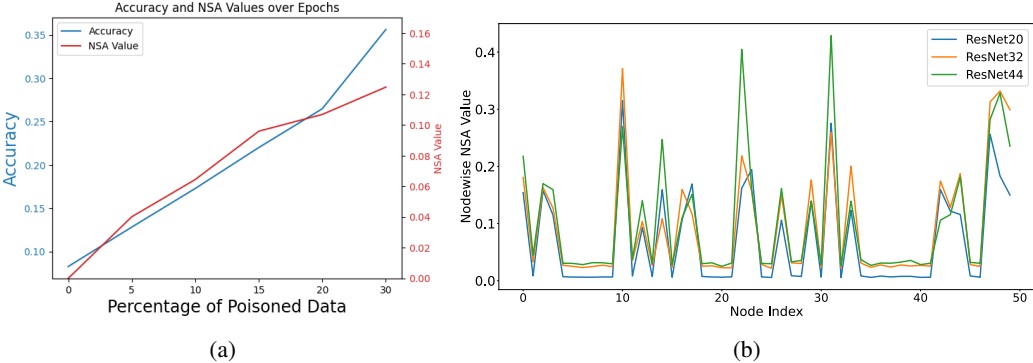

Figure 7: Adversarial analysis on ResNet models trained on CIFAR-10. (a) Correlation of NSA with misclassification rate under evasion attack at increasing perturbation rates. (b) Nodewise NSA on the first 50 images of the test set of CIFAR-10 on the perturbed dataset vs the clean dataset. Any perturbed image shows a significant spike in its NSA value post attack.

## J ARE GNNs TASK SPECIFIC LEARNERS?

We use NSA to investigate whether representations produced by GNNs are specific to the downstream task. For this, we explored the similarity of representations of different GNN architectures when trained on the same task versus their similarity when trained on different downstream tasks.

### J.1 CROSS ARCHITECTURE TESTS ON THE SAME DOWNSTREAM TASK

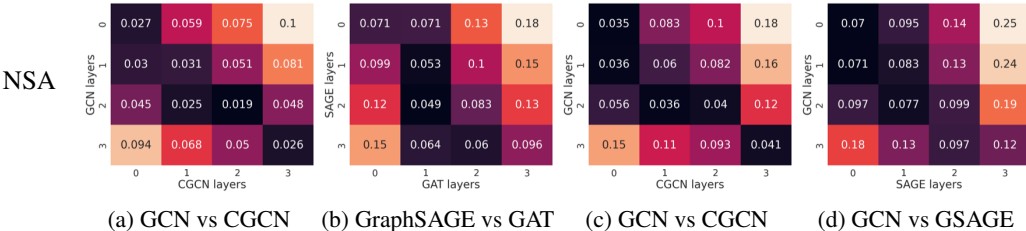

|  (a) GCN vs CGCN | (b) GraphSAGE vs GAT | (c) GCN vs CGCN | (d) GCN vs GSAGE |

Figure 8: Cross Architecture Tests using NSA on the Amazon Computers Dataset. (a) Layerwise NSA values between GCN and ClusterGCN on Node Classification (b) Layerwise NSA values between GraphSAGE and GAT on Node Classification (c) Layerwise NSA values between GCN and ClusterGCN on Link Prediction (d) Layerwise NSA values between GCN and GSAGE on Link Prediction. Similar architectures showcase a layerwise pattern when trained on the same task.

We tested layerwise representational similarity on the task of node classification (on the Amazon Computers Dataset) across different GNN architectures: GCN, ClusterGCN (CGCN), GraphSAGE, and GAT. Just like in the sanity tests, models that are similar in architecture or training paradigms have a higher layerwise similarity: this implies that the highest similarity is between GCN and ClusterGCN that differ only in their training paradigms. We also observe (a less pronounced) linear relationship between the other pairs of architectures. These results are shown in Figure 8.

### J.2 ARCHITECTURE TESTS ON DIFFERENT DOWNSTREAM TASKS

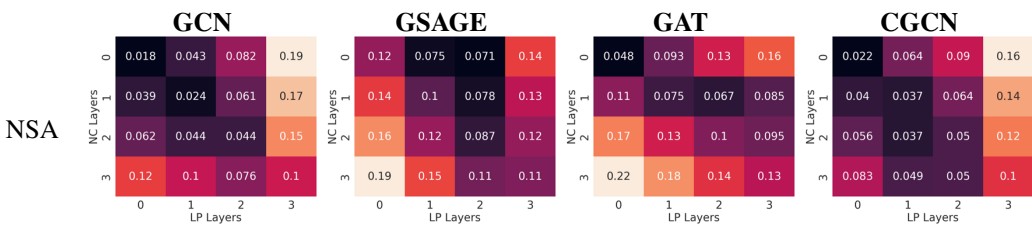

Figure 9: Effect of Downstream Task on Representations Achieved by the Same Architecture (Amazon Computers Dataset). The layerwise dissimilarity of four GNN architectures is compared on two different downstream tasks: Node Classification and Link Prediction. There is no observable correlation across layers suggesting that the GNNs generate different representation spaces for the same dataset on different downstream tasks.

We conducted experiments with different downstream tasks to assess layerwise similarity between two models, both of which shared structural identity except for disparities in their final layers. Specifically, we employed node classification and link prediction as the two downstream tasks for our investigations. We present our findings in Figure 9. They reveal that there is little relationship across layers for any of the four architectures.

The results presented in this section indicate that that different GNN architectures have similar representation spaces when trained on the same downstream task and conversely, similar architectures have different representation spaces when trained on different downstream tasks. Extending this idea further, it should be possible to train Graph Neural Networks to conform to a task specific representation template. This structural template will be agnostic of GNN architectures and provide a

high degree of functional similarity. If we train GNNs to minimize discrepancy loss with such a task specific template, we could train more directly and without adding downstream layers of tasks.

## K  DOWNSTREAM TASK ANALYSIS WITH THE STS DATASET

We assess the performance of embeddings derived from NSA-AE in the context of the Semantic Text Similarity (STS) Task. We employ Google's Word2Vec word vectors as the initial dataset, processing these 300-dimensional vectors through both a standard autoencoder and the NSA-Autoencoder to achieve reduced dimensions of 128 and 64. The efficacy of these dimensionality-reduced embeddings is then evaluated using the STS Multi EN dataset (Cer et al., 2017). This evaluation aims to determine if the word vectors, once dimensionally reduced, maintain their alignment, as evidenced by a strong positive correlation in similarity scores on the STS dataset. The comparative results, as detailed in Table 3, demonstrate that the NSA-Autoencoder outperforms the traditional autoencoder in preserving semantic accuracy at both 128 and 64 dimensions.

Table 3: Pearson Correlation between the similarities output from the original embeddings and the similarities output from the reduced embeddings. NSA-AE shows a much higher correlation with the similarity outputs obtained by evaluating the original word2vec embeddings on the STS multi EN dataset

| Latent Dimension | Basic AutoEncoder | NSA AutoEncoder |
|---|---|---|
| 128 | -0.518 | 0.7632 |
| 64 | 0.567 | 0.7619 |

## L  AUTOENCODER HYPERPARAMETERS

The hyperparameter setup for all the autoencoder architectures is detailed in Table 4. All autoencoder architectures are trained using pytorch-lightning and the input data is normalized using a MinMaxScaler.

Table 4: Autoencoder Hyperparameters. All four architectures used the same hyperparameters. To ensure similarity of testing conditions we replicate the hyperparameter setup from Trofimov et al. (2023)

| Dataset Name | Batch Size | LR | Hidden Dim | Layers | Epochs | Metric Start Epoch |
|---|---|---|---|---|---|---|
| MNIST | 256 | $10^{-4}$ | 512 | 3 | 250 | 60 |
| F-MNIST | 256 | $10^{-4}$ | 512 | 3 | 250 | 60 |
| CIFAR-10 | 256 | $10^{-4}$ | 512 | 3 | 250 | 60 |
| COIL-20 | 256 | $10^{-4}$ | 512 | 3 | 250 | 60 |

## M  RUNNING TIME AND RECONSTRUCTION LOSS FOR AUTOENCODER

We show that using NSA along with Mean Squared Error in an autoencoder does not significantly compromise the model's ability to reconstruct the original embedding. Additionally, NSA-AE is more efficient than RTD-AE and Topo-AE. The results are presented in Table 5

## N  GNN HYPERPARAMETERS

The hyperparameter setup for all the GNN architectures is detailed in Table 6

Table 5: Reconstruction Loss and Time Per Epoch for different Autoencoder architectures. All architectures were trained for 250 epochs with the auxiliary loss (TopoLoss, RTD, NSA) kicking in after 60 epochs.

| Dataset | Method | Metric | | |
| --- | --- | --- | --- | --- |
| | | Time per epoch | Train MSE | Test MSE |
| MNIST | AE | 2.344 | 7.64e-3 | 8.62e-3 |
| | TopoAE | 39.168 | 6.89e-3 | 8.16e-3 |
| | RTD-AE | 51.608 | 9.36e-3 | 1.07e-2 |
| | NSA-AE | 5.816 | 8.75e-3 | 1.00e-2 |
| F-MNIST | AE | 6.436 | 8.99e-03 | 9.79e-03 |
| | TopoAE | 37.26 | 8.94-03 | 9.83e-03 |
| | RTD-AE | 59.94 | 1.12e-02 | 1.24e-02 |
| | NSA-AE | 5.764 | 9.49e-03 | 1.04e-02 |
| CIFAR-10 | AE | 5.16 | 1.56e-02 | 1.65e-02 |
| | TopoAE | 58.664 | 1.54e-02 | 1.68e-02 |
| | RTD-AE | 56.172 | 1.60e-02 | 1.91e-02 |
| | NSA-AE | 6.996 | 1.58e-02 | 1.71e-02 |
| COIL-20 | AE | 2.012 | 1.67e-02 | - |
| | TopoAE | 4.876 | 1.09e-02 | - |
| | RTD-AE | 16.404 | 1.90e-02 | - |
| | NSA-AE | 8.716 | 1.80e-02 | - |

Table 6: GNN Architecture Information

| Architecture | Layers | Hidden Dim | LR | Epochs | Additional Info |
| --- | --- | --- | --- | --- | --- |
| GCN | 4 | 128 | 0.001 | 200 | - |
| GraphSAGE | 4 | 128 | 0.001 | 200 | Mean Aggregation |
| GAT | 4 | 128 | 0.001 | 200 | 8 heads with Hid Dim 8 each |
| CGCN | 4 | 128 | 0.001 | 200 | 8 subgraphs |

## O    GNN METRICS

The test accuracy for node classification and the ROC AUC value for link prediction is detailed in Table 7

Table 7: GNN Metric Data on Amazon Computer Dataset. Test Accuracy is for Node Classification and ROC AUC Score is for Link Prediction

| Architecture | Accuracy (NC) | ROC AUC (LP) |
| --- | --- | --- |
| GCN | 0.8257 | 0.8638 |
| GraphSAGE | 0.8800 | 0.8068 |
| GAT | 0.8273 | 0.7997 |
| CGCN | 0.8200 | 0.8716 |

## P DATASETS

We report the statistics of the datasets used for the empirical analysis of GNNs in Table 8. We report the exact statistics of the autoencoder datasets in Table 9.

### P.1 GRAPH DATASETS

The Amazon computers dataset is a subset of the Amazon co-purchase graph (Shchur et al., 2019). The nodes represent products on amazon and the edges indicate that two products are frequently bought together. The node features are the product reviews encoded in bag-of-words format. The class labels represent the product category. We also use the Flickr, Cora, Citeseer and Pubmed dataset in additional experiments.

Table 8: Dataset Statistics for Graph

| Dataset | Amazon Computer | Flickr | Cora | Citeseer | Pubmed |
|---|---|---|---|---|---|
| Number of Nodes | 13752 | 89250 | 2708 | 3327 | 19717 |
| Number of Edges | 491722 | 899756 | 10556 | 9104 | 88648 |
| Average Degree | 35.76 | 10.08 | 3.90 | 2.74 | 4.50 |
| Node Features | 767 | 500 | 1433 | 3703 | 500 |
| Labels | 10 | 7 | 7 | 6 | 3 |

### P.2 AUTOENCODER DATASETS

Four diverse real-world datasets were utilized for our experiments: MNIST (LeCun et al., 2010), Fashion-MNIST (F-MNIST) (Xiao et al., 2017), COIL-20 (Nene et al., 1996), and CIFAR-10 (Krizhevsky & Hinton, 2009), to comprehensively evaluate the performance of our autoencoder model. MNIST, comprising 28x28 grayscale images of handwritten digits, serves as a foundational benchmark for image classification and feature extraction tasks. Fashion-MNIST extends this by offering a similar format but with 10 classes of clothing items, making it an ideal choice for fashion-related image analysis. COIL-20 presents a unique challenge, with 20 object categories, where each category consists of 72 128x128 color images captured from varying viewpoints, offering a more complex 3D object recognition scenario.

Table 9: Dataset Statistics for AE

| Dataset | Classes | Train Size | Test Size | Image Size | Data Type |
|---|---|---|---|---|---|
| MNIST | 10 | 60,000 | 10,000 | 28x28 (784) | Grayscale |
| Fashion-MNIST (F-MNIST) | 10 | 60,000 | 10,000 | 28x28 (784) | Grayscale |
| COIL-20 | 20 | 1,440 | - | 128x128 (16384) | Color |
| CIFAR-10 | 10 | 60,000 | 10,000 | 32x32*3 (3072) | Color |

## Q ABLATION STUDY

NSA consists of two parts, GlobalNSA and LocalNSA. These two components work well together and are capable of working individually too. LocalNSA makes locally optimal decisions but it could run into a problem when the global structure needs to be preserved. Figure 10 shows a scenario where LocalNSA would not obtain a high k-NN consistency between two representation spaces. This could potentially explain the low kNN consistency when minimizing with just LocalNSA in Table 10. LocalNSA's inability to preserve global structure becomes especially prevalent in our LinkPrediction tests in Table 11. We intend on further improving LocalNSA and coming up with better preprocessing

Table 10: Autoencoder results. NSA-AE outperforms or almost matches all other approaches on all the evaluation metrics. RTD-AE, which explicitly minimizes on RTD has a slightly lower RTD value while PCA has marginally higher Triplet Ranking Accuracy on Cluster Centers.

| Dataset | Method | | | | Quality measure | | | |
|---|---|---|---|---|---|---|---|---|
| | | L. C. | T. A. | RTD | T. A. C.C | NSA | LID-NSA | kNN-C |
| MNIST | GNSA-AE | 0.942 | $0.885 \pm 0.006$ | $5.44 \pm 0.12$ | $0.944 \pm 0.230$ | $\mathbf{0.0198 \pm 0.0001}$ | $0.0342 \pm 0.00$ | 0.616 |
| | LNSA-AE | 0.819 | $0.794 \pm 0.007$ | $7.39 \pm 0.21$ | $0.832 \pm 0.374$ | $0.0548 \pm 0.00$ | $0.0373 \pm 0.00$ | 0.607 |
| | NSA-AE | **0.947** | $\mathbf{0.886 \pm 0.005}$ | $\mathbf{4.29 \pm 0.11}$ | $\mathbf{0.946 \pm 0.226}$ | $0.0222 \pm 0.00$ | $\mathbf{0.0123 \pm 0.00}$ | **0.634** |
| F-MNIST | GNSA-AE | **0.987** | $\mathbf{0.952 \pm 0.002}$ | $4.11 \pm 0.21$ | $0.992 \pm 0.089$ | $\mathbf{0.0091 \pm 0.0001}$ | $0.0350 \pm 0.00$ | 0.592 |
| | LNSA-AE | 0.887 | $0.859 \pm 0.006$ | $5.41 \pm 0.18$ | $0.952 \pm 0.214$ | $0.0488 \pm 0.00$ | $0.0470 \pm 0.00$ | 0.608 |
| | NSA-AE | 0.9854 | $0.939 \pm 0.003$ | $\mathbf{2.78 \pm 0.09}$ | $\mathbf{0.992 \pm 0.089}$ | $0.0114 \pm 0.00$ | $\mathbf{0.0133 \pm 0.00}$ | **0.633** |
| CIFAR-10 | GNSA-AE | **0.985** | $0.936 \pm 0.004$ | $3.07 \pm 0.11$ | $0.984 \pm 0.125$ | $\mathbf{0.0077 \pm 0.0001}$ | $0.0122 \pm 0.00$ | 0.453 |
| | LNSA-AE | 0.8616 | $0.849 \pm 0.006$ | $4.02 \pm 0.28$ | $0.936 \pm 0.245$ | $0.0624 \pm 0.00$ | $0.0220 \pm 0.00$ | 0.359 |
| | NSA-AE | 0.985 | $\mathbf{0.941 \pm 0.003}$ | $3.04 \pm 0.16$ | $0.984 \pm 0.125$ | $0.0086 \pm 0.00$ | $\mathbf{0.0103 \pm 0.00}$ | **0.437** |
| COIL-20 | GNSA-AE | 0.955 | $0.919 \pm 0.004$ | $7.46 \pm 0.23$ | $0.939 \pm 0.240$ | $0.0157 \pm 0.0$ | $0.3427 \pm 0.00$ | 0.721 |
| | LNSA-AE | 0.848 | $0.838 \pm 0.007$ | $9.74 \pm 0.26$ | $0.887 \pm 0.316$ | $0.0672 \pm 0.00$ | $0.5810 \pm 0.00$ | 0.703 |
| | NSA-AE | **0.976** | $\mathbf{0.919 \pm 0.005}$ | $\mathbf{3.55 \pm 0.18}$ | $\mathbf{0.962 \pm 0.191}$ | $\mathbf{0.0139 \pm 0.00}$ | $\mathbf{0.0085 \pm 0.00}$ | **0.809** |

schemes to optimize its performance. In practice, GNSA works well and performs competitively even when used without LNSA. LNSA provides a noticeable improvement but requires dataset specific fine tuning to find the optimal hyperparameter set up.

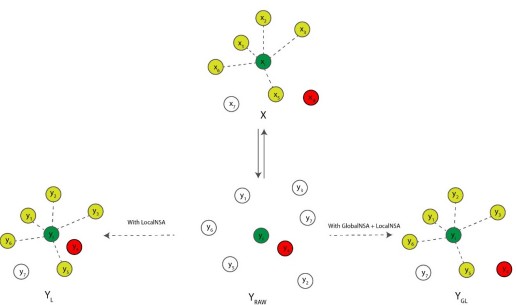

Figure 10: Illustration showing why LocalNSA might not maintain neighborhood consistency when used without GlobalNSA. The objective is to minimize the structural discrepancy between the reference space X (top) and generated space Y (mid-bottom). LocalNSA aims to minimize the difference in distances for the k-nearest neighbors of the $i_{th}$ point in X space (in yellow). If a random initialization causes an unknown point (in red) to move close to the $i_{th}$ point in Y space, LocalNSA will not move it out, instead focusing on bringing the 5 NNs of X close to i in Y causing a drop in neighborhood consistency $Y_L$ (left-bottom). Using GlobalNSA with LocalNSA alleviates this issue as GlobalNSA pushes point 4 away in the $Y_{GL}$ (right-bottom) .

Table 11: Downstream task analysis with Link Prediction. The output embeddings from a GCN trained on link prediction are passed through all 3 autoencoder architectures. Latent embeddings are obtained at various dimensions and ROC-AUC scores are calculated on the latent embeddings. Using both LNSA and GNSA gives the best performance.

| Method | Latent Dim | Dataset | | | |
|---|---|---|---|---|---|
| | | Amazon Comp | Cora | Citeseer | Pubmed |
| | 128 | 95.74 | 96.04 | 97.18 | 97.31 |
| G-NSA-AE | 64 | **95.90** | 95.54 | 96.85 | 96.40 |
| | 32 | **96.01** | 95.37 | 94.25 | 97.53 |
| | 128 | 76.82 | 50.01 | 50.13 | 93.03 |
| L-NSA-AE | 64 | 74.24 | 50.69 | 50.25 | 93.64 |
| | 32 | 80.67 | 55.44 | 51.43 | 95.69 |
| | 128 | **95.75** | 97.83 | 97.91 | 97.37 |
| NSA-AE | 64 | 94.60 | 97.89 | 98.38 | 96.97 |
| | 32 | 95.76 | 97.68 | 96.11 | 97.66 |

## Q.1 Visualizing The Effect with Spheres Dataset

We use the Spheres dataset (Moor et al., 2020), reducing the original 100 dimension data to 2 dimensions to visually observe the effect that different scaling factors for GNSA and LNSA have. The Spheres dataset consists of 10 spheres clustered within an 11th large sphere in 100 dimensions. A visualization of what the structure looks like in 3D is provided in Figure 15. The model is trained on a batch size of 200 and with no rescaling applied to the Spheres dataset. Additionally, NSA is the only loss applied here while in the autoencoder experiments, we also use MSE loss to contain the latent space. We observe that neither LNSA or GNSA do a perfect job when working individually. LNSA seems to work best with a scaling factor of 0.5 or 1 while GNSA starts exhibiting stable behavior when $g$ is atleast 1. Throughout all our experiments where both LNSA and GNSA are used, the values of both $l$ and $g$ are set to 1.

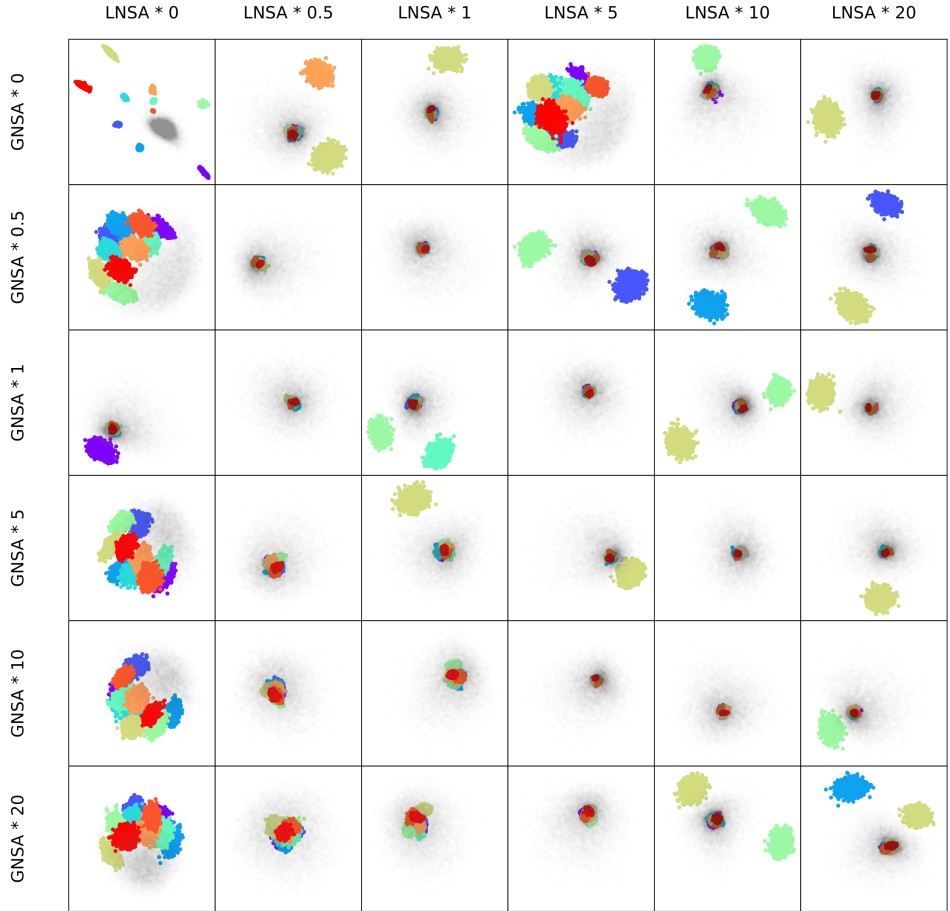

Figure 11: Illustrating the effect that different values of $l$ and $g$ have on convergence on the Spheres Dataset. A good reconstruction is defined by all the colored spheres being within the gray sphere.

## Q.2 Choosing the Right Value of $k$

While $l$ and $g$ typically work without scaling (setting both values to 1), the choice of $k$ is more nuanced. The parameter $k$ controls the number of nearest neighbors in the reference space that LNSA uses to compute local similarity. Intuitively, smaller values of $k$ provide a more localized view, while larger values approach a global perspective. Figure 13 confirms this intuition.

In this experiment, we attempt to reconstruct a 3D swiss roll using only the LocalNSA loss, where the neural network minimizes distances within the local neighborhood defined by $k$. As $k$ increases, the reconstructed swiss roll retains more of its geometric structure. At low $k$ values, where LocalNSA

focuses solely on local structures without a global perspective, the reconstruction results in a ribbon-like structure that lacks global coherence. However, as $k$ increases, the network progressively captures the broader geometric characteristics of the swiss roll.

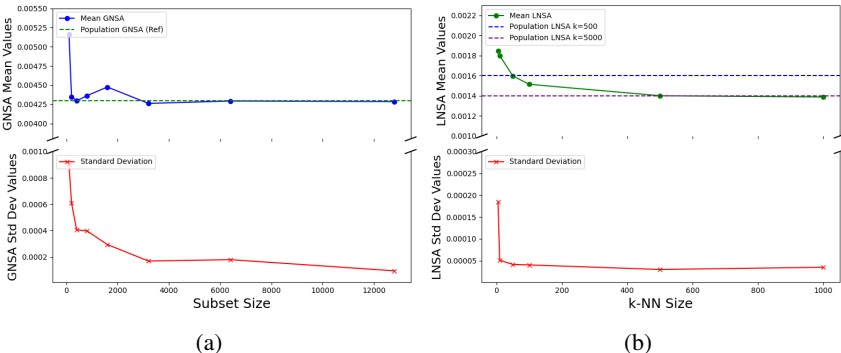

(a)                                         (b)

Figure 12: Variation of Mean and Standard Deviation of subsets for GNSA and LNSA. Values are computed between the outputs of a ResNet-18 and a ResNet-34 on 50,000 ImageNet images. (a) Mean and Standard Deviation of GlobalNSA at various mini batch sizes. (b) Convergence of LocalNSA to its Global value in mini batching. The subset values are averaged over 10 runs. The X-axis varies the k value of the minibatch. The minibatch size is set to 5000. We also present the standard deviations to demonstrate the stability of LocalNSA

A critical consideration is how to select the appropriate $k$ value during mini-batching. Intuitively, LocalNSA converges to its global value if the proportion of $k$ to the dataset size is preserved when applied to mini-batches. For example, if the dataset contains 20,000 points and the global $k$ value is set to 200 (1/100th of the dataset), then the mini-batch $k$ value should also be set to 1/100th of the mini-batch size to maintain consistency.

We illustrate this concept in Figure 12, showing how consistent $k$ scaling between the full dataset and mini-batches ensures that LocalNSA converges to the expected global value, maintaining both stability and interpretability in the loss function across different batch sizes.

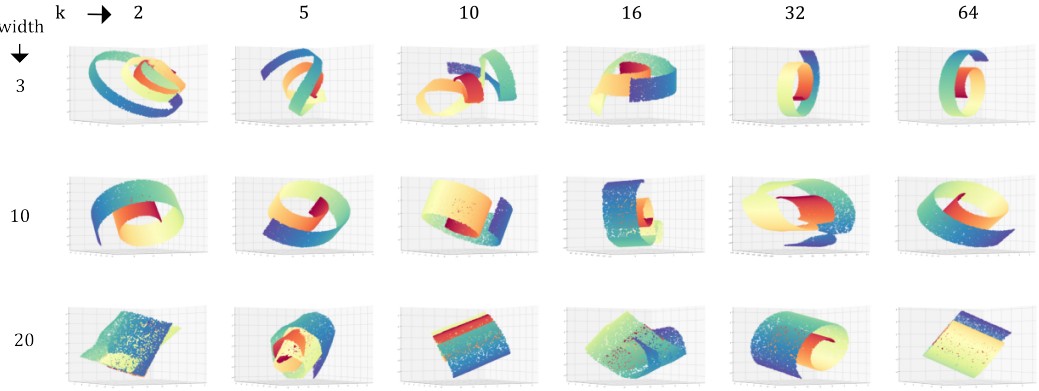

Figure 13: Illustrating the effect of different values of k on the reconstruction of a Swiss Roll. The width of the Swiss Roll is varied across rows to showcase how nearest neighbors approximation might falter when the width is too high and the k is too low. The batch size is set to 128 and the Swiss Roll consists of 20,000 points

# R  APPROXIMATING THE MANIFOLD OF A REPRESENTATION SPACE

Incorporating a structure-preserving metric with MSE, NSA-AE maintains the Euclidean distances between data points as it compresses the data into a lower-dimensional space. High-dimensional data

typically resides in a lower-dimensional manifold (Goodfellow et al., 2016a), where the Euclidean distance approximates the geodesic distance. Utilizing this approximation allows NSA-AE to unfold complex datasets to their manifold dimensions, providing a clearer interpretation of underlying data relationships that are often obscured in higher dimensional ambient spaces.

Our experiments with the Swiss Roll dataset illustrate the effectiveness of this approach. Figure 14(b) and 14(c) from our results display the outcomes of dimensionality reduction when prioritizing Euclidean distances versus geodesic distances. To compute geodesic distances for NSA, we employ the Floyd-Warshall algorithm to approximate the all pairs shortest path distances on the original space. Preserving the geodesic distances lets us construct a more accurate representation of the Swiss Roll's original manifold structure. Conversely, in cases where the manifold dimension of the input data remains unknown, the latent dimension at which NSA minimization becomes optimal inherently corresponds to the manifold dimension of the data. This observation underscores the practical significance of understanding the manifold dimension when leveraging NSA for dimensionality reduction within an autoencoder framework.

We also present results on dimensionality reduction for the Spheres Dataset from Moor et al. (2020). We show that you can use NSA-AE to reduce the 100 dimension dataset to 3 or even 2 dimensions and still preserve the structure of the representation space in Figure 15. While manifold learning techniques such as ISOMAP (Tenenbaum et al., 2000) and MDS (Kruskal, 1964) are also capable of unfolding complex datasets and preserving geodesic distances, they require access to the entire dataset for computation. This necessitates significant computational resources, particularly for large datasets. In contrast, NSA is able to achieve the same result while operating on mini-batches, making it substantially more computationally efficient and scalable.

Finally, we present reconstruction results on the Mammoth dataset. The original dataset is in 3 Dimensions and NSA-AE successfully preserves the structure of the dataset in 2 Dimensions. We present this result in Figure 16.

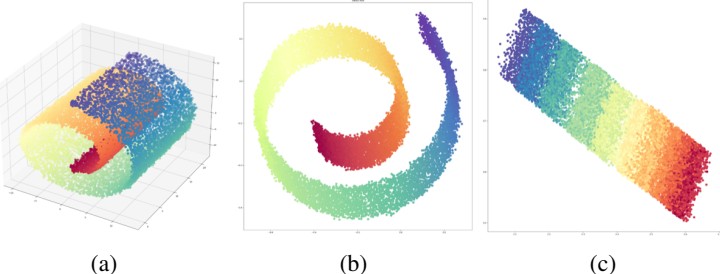

(a)            (b)            (c)

Figure 14: Results of NSA-AE on the Swiss Roll Dataset. (a) The original Swiss Roll Dataset in 3D. (b) Result of dimensionality reduction to 2D using NSA-AE when minimizing euclidean distance. (c) Result of dimensionality reduction to 2D using NSA-AE when minimizing geodesic distance

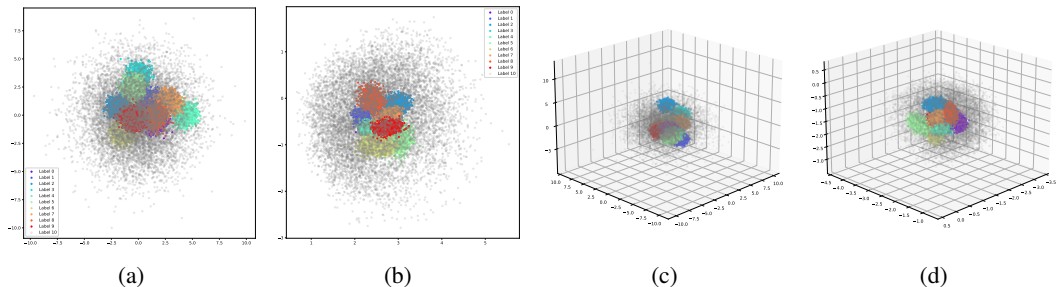

(a)        (b)        (c)        (d)

Figure 15: Dimensionality reduction on the Spheres dataset with NSA-AE. (a) 2 randomly chosen dimensions from the 100 dimension dataset (b) 2D-Latent Space obtained from NSA-AE (c) 3 randomly chosen dimensions from the 101 dimension dataset (d) 3D-Latent Space obtained from NSA-AE

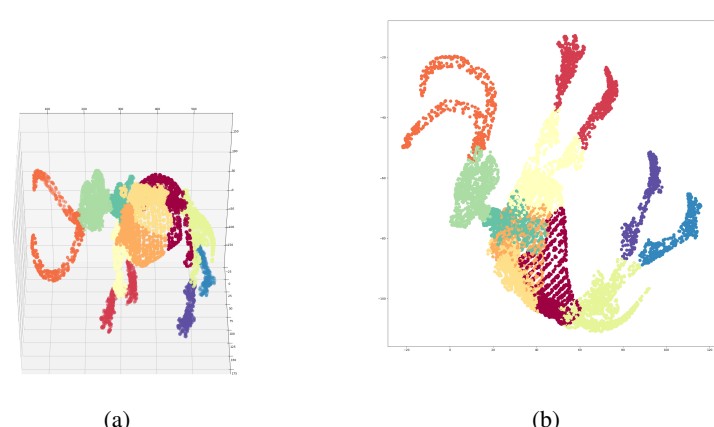

(a)  (b)

Figure 16: Results of NSA-AE on the Mammoth Dataset. (a) The original Mammoth Dataset in 3D. (b) Result of dimensionality reduction to 2D using NSA-AE.

## S VISUALIZING THE LATENT EMBEDDINGS FROM AUTOENCODERS

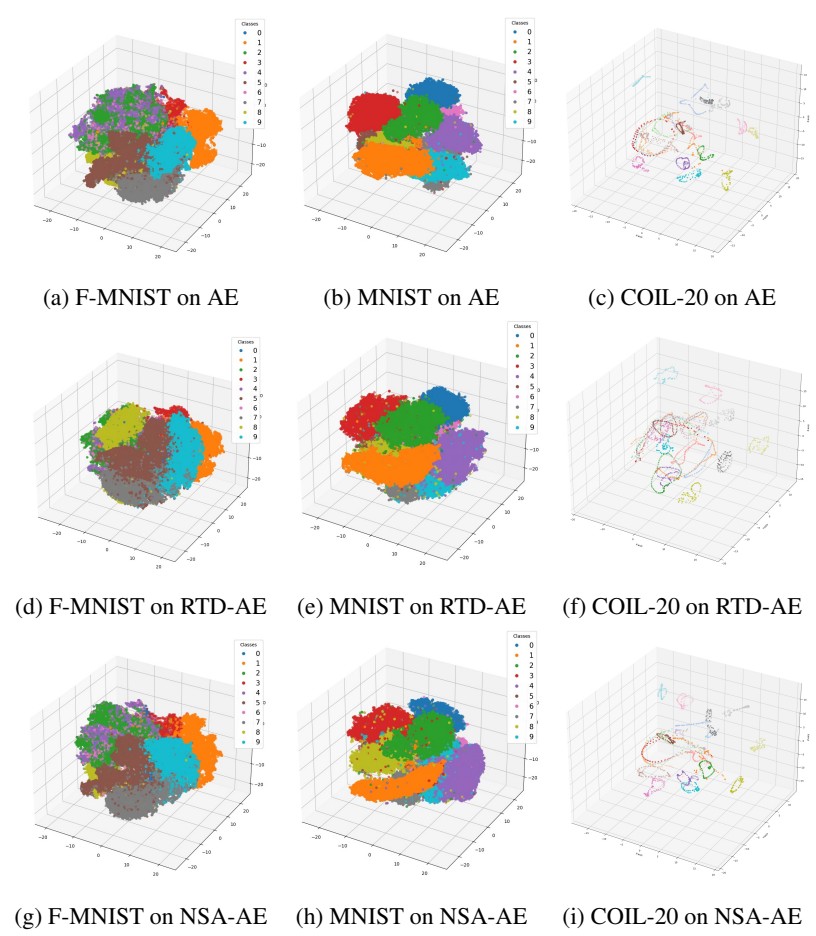

(a) F-MNIST on AE  (b) MNIST on AE  (c) COIL-20 on AE

(d) F-MNIST on RTD-AE  (e) MNIST on RTD-AE  (f) COIL-20 on RTD-AE

(g) F-MNIST on NSA-AE  (h) MNIST on NSA-AE  (i) COIL-20 on NSA-AE

Figure 17: Visualizing the latent representations of the autoencoders with t-SNE

We use t-SNE(van der Maaten & Hinton, 2008) to reduce the latent embeddings obtained from all 4 datasets used in Table 1 to reduce their dimension from 16 to 3. The results are presented in Figure 17. We observe that t-SNE generates similar clusters across all three architectures; the basic Autoencoder, RTD-Autoencoder and NSA-Autoencoder.

It is important to note that the observation of similar clustering patterns between AE and NSA-AE embeddings in t-SNE plots does not necessarily indicate identical representation structures in the original high-dimensional latent spaces. This is because t-SNE's emphasis on local similarities can result in visually analogous clusters even when the global structures or the exact nature of the representations differ significantly between the two methods. NSA-AE's latent space captures the exact structure of the input representation space significantly better than a normal AE, preserving the data's geometric and topological properties. This is demonstrated by NSA-AE's latent embeddings having superior accuracy in downstream tests where exact structure preservation is vital (Section 4.3 and Appendix K).

## T    REPRODUCIBILITY STATEMENT

Our code is anonymously available at https://anonymous.4open.science/r/NSA-B1EF. The code includes notebooks with instructions to reproduce all the experiments presented in this paper. Hyperparameter setups are available in Appendices L, N and P. Detailed instructions are given in the README and in the notebooks. All the code was run on a conda environment with a single NVIDIA V100 GPU.

## U    LAYERWISE SPECIFICITY TESTS

In the second experiment, we compare the layerwise dissimilarity of differing initializations to layers of the same model trained with the same seed. In this setup, a robust metric should not only assign the lowest dissimilarity to corresponding layers across different seeds but also ensure that the dissimilarity between corresponding layers is lower than that between non-corresponding layers within the same model (e.g., layer 7 vs. other layers of the same network). This analysis is crucial for quantifying a metric's depth insensitivity, which we define as the average number of non corresponding layers within the same network that exhibit a lower dissimilarity than corresponding layers across different seeds. A high depth insensitivity indicates poor specificity, as it implies that the metric fails to differentiate layers in different networks from non-corresponding layers within the same network.

We observe that NSA exhibits the lowest depth insensitivity, with an average of 1.11 layers being incorrectly prioritized over corresponding layers in different networks. In contrast, CKA and RTD show average depth insensitivity values of 2.33 and 1.88, respectively. These results are in line with the experiments proposed by Ding et al. (2021) to quantify the performance of similarity indices.

Both the specificity experiments use the same experimental setup, with six models trained independently on CIFAR-10 using different initialization seeds. Similarity computations for all metrics are averaged over 10 trials, with subset sizes of 4000 for NSA and 400 for RTD, following their respective recommendations. We also average results across three sets of pairwise comparisons between the models to ensure reliability and present the standard deviations across the sets. Standard deviations across trials are orders of magnitude smaller than the mean and do not affect the reported trends and are therefore not reported. Figure 18 showcases the results of our layerwise experiments.

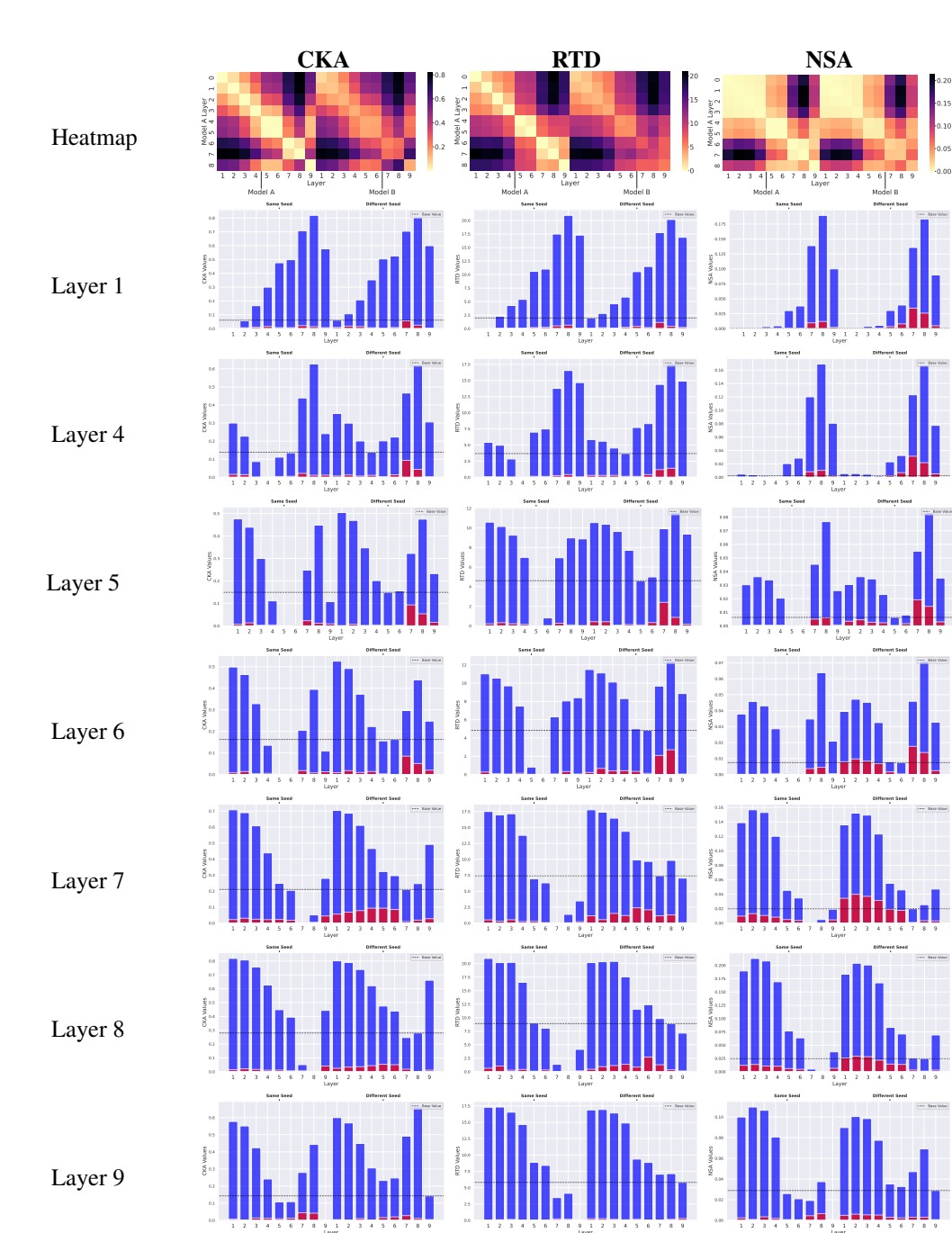

Figure 18: Layer wise Specificity Tests. We showcase the variation of all 3 metrics when trained on different seeds. Column 1 shows the layerwise dissimilarity of CKA, Column 2 shows the layerwise dissimilarity of RTD, and Column 3 shows the layerwise dissimilarity of NSA. The blue columns are the mean values across all 3 pairs of experiments and the red columns are the standard deviations. The dotted line marks the dissimilarity value of corresponding layers across initializations. The left set of columns in each subfigure represent the results on the same initialization while the right set of columns represent the results on different initializations.

## V    EMPIRICAL ANALYSIS ON ADDITIONAL DATASETS

Empirical analysis of intermediate representations with similarity measures can help uncover the intricate workings of various neural network architectures. When two networks that are structurally identical but trained from different initial conditions are compared, we expect that, for each layer's intermediate representation, the most similar representation in the other model should be the corresponding layer that matches in structure. We can also perform cross-layer analysis and cross-downstream task analysis of the intermediate representations of a neural network to help uncover learning patterns of a network. We use 4 different GNN architectures; GCN (Kipf & Welling, 2017), GraphSAGE (Hamilton et al., 2017), ClusterGCN (Chiang et al., 2019) and Graph Attention Network (GAT) (Veličković et al., 2018). We evaluate with Sanity Tests, Cross Architecture Tests, Downstream Task tests and Convergence Tests. We utilize the Amazon Computers Dataset (Shchur et al., 2019), Cora (McCallum et al., 2000), Citeseer (Giles et al., 1998), Pubmed (Sen et al., 2008) and Flickr (Zeng et al., 2020) Datasets for our experiments.

### V.1    INITIALIZATION SANITY TESTS

The sanity test compares the intermediate representations obtained from each layer of the Graph Neural Network against all the other layers of another GNN with the exact same architecture but trained with a different initialization. we expect that, for each layer's intermediate representation, the most similar intermediate representation in the other model should be the corresponding layer that matches in structure. We only demonstrate the performance of NSA for both Node Classification and Link Prediction.

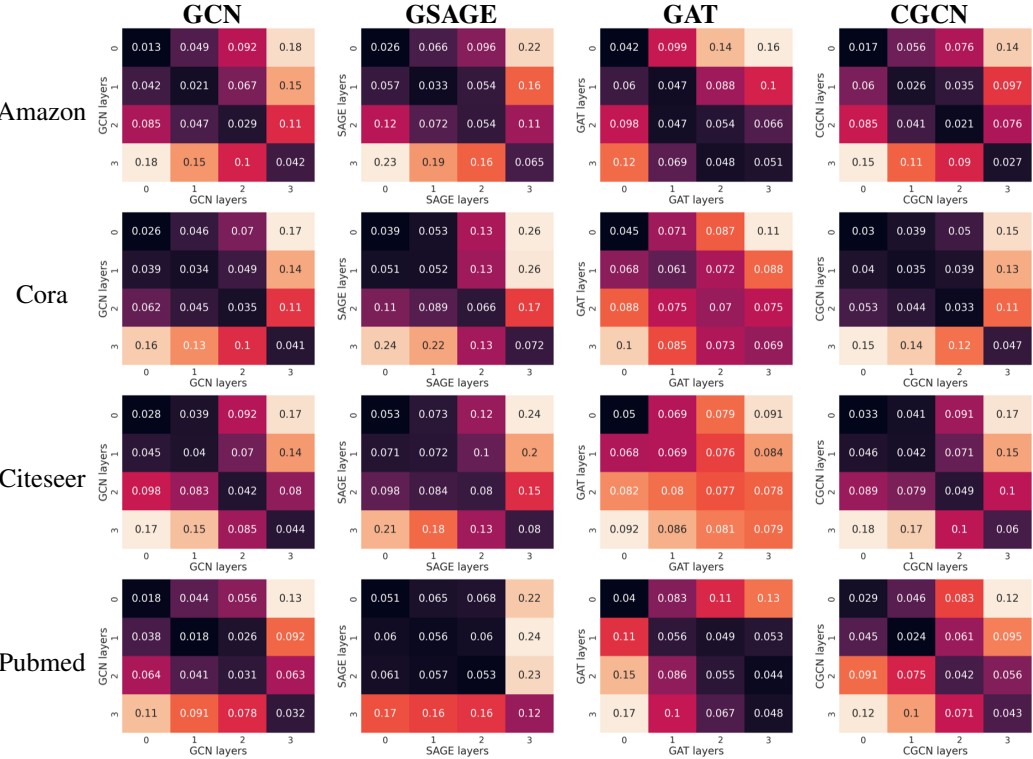

Figure 19: Sanity Tests for Link Prediction on 3 datasets. The heatmaps show layer-wise dissimilarity values for two different initialization of the same dataset on four different GNN architectures.

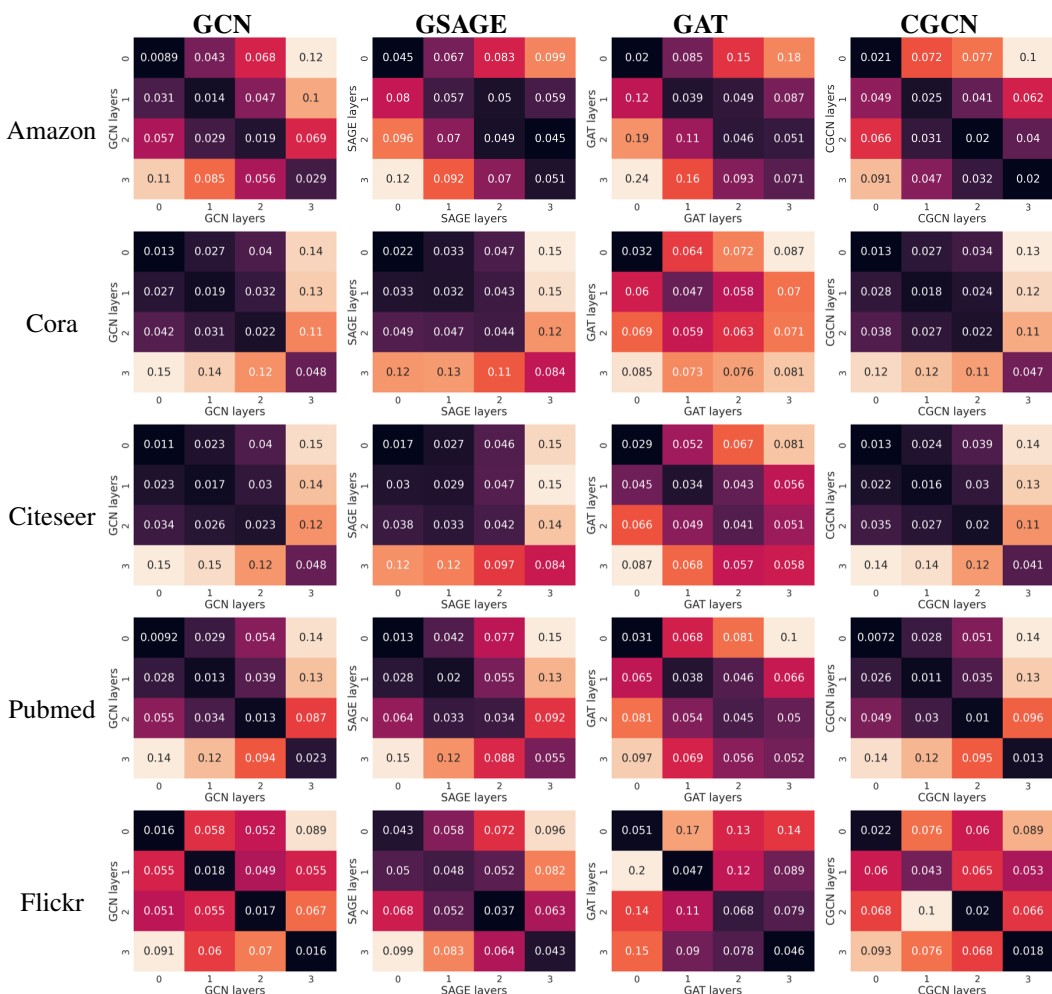

Figure 20: Sanity Tests for Node Classification on all 4 datasets. The heatmaps show layer-wise dissimilarity values for two different initialization of the same dataset on four GNN architectures.

## V.2 CROSS ARCHITECTURE TESTS ON NODE CLASSIFICATION

In this section we evaluate the similarity in representations across architectures for the Amazon Computers dataset and the three Planetoid datasets; Cora, Citeseer and Pubmed on Node Classification.

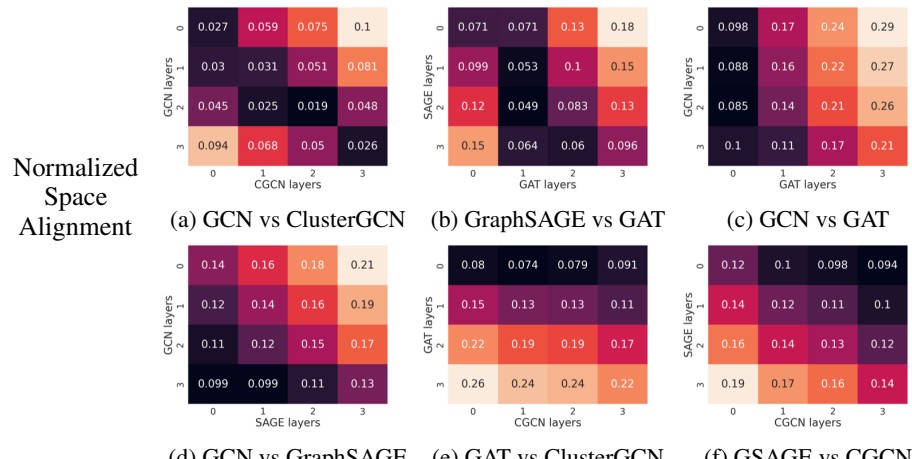

Figure 21: Cross Architecture Tests using NSA for Node Classification on the Amazon Dataset

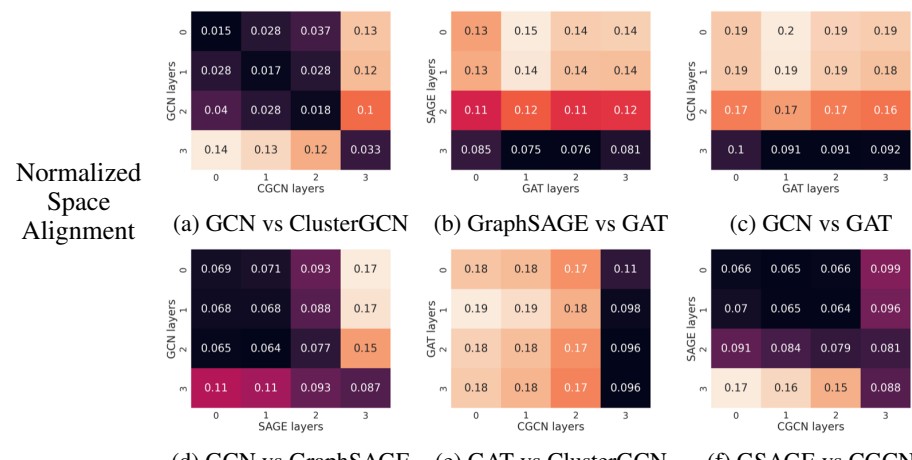

Figure 22: Cross Architecture Tests using NSA for Node Classification on the Cora Dataset

Normalized Space Alignment

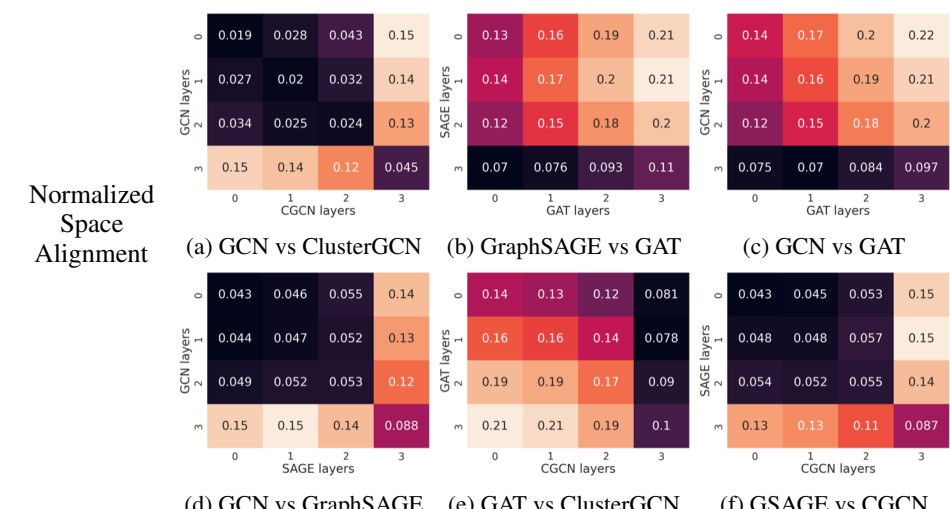

(a) GCN vs ClusterGCN    (b) GraphSAGE vs GAT    (c) GCN vs GAT

(d) GCN vs GraphSAGE    (e) GAT vs ClusterGCN    (f) GSAGE vs CGCN

Figure 23: Cross Architecture Tests using NSA for Node Classification on the Citeseer Dataset

Normalized Space Alignment

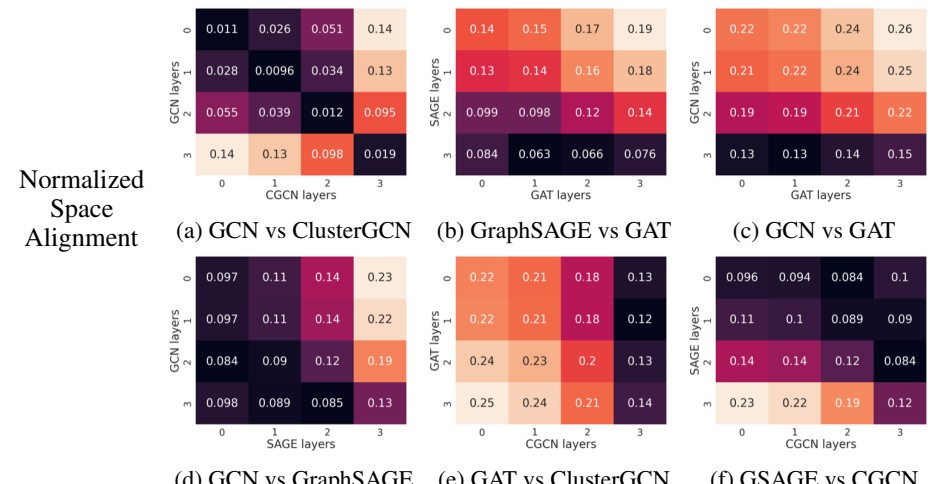

(a) GCN vs ClusterGCN    (b) GraphSAGE vs GAT    (c) GCN vs GAT

(d) GCN vs GraphSAGE    (e) GAT vs ClusterGCN    (f) GSAGE vs CGCN

Figure 24: Cross Architecture Tests using NSA for Node Classification on the Pubmed Dataset

### V.3 CROSS ARCHITECTURE TESTS ON LINK PREDICTION

In this section we evaluate the similarity in representations across architectures for the Amazon Computers dataset and the three Planetoid datasets; Cora, Citeseer and Pubmed on Link Prediction.

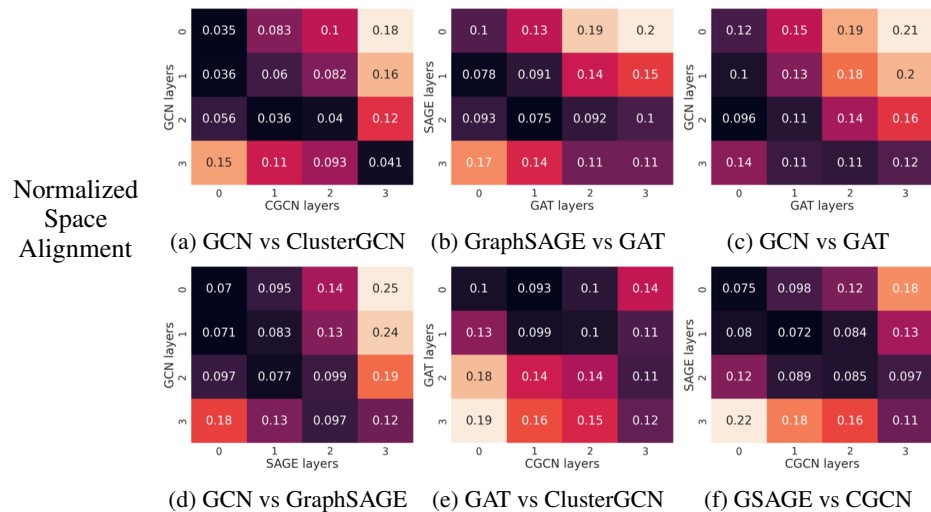

Figure 25: Cross Architecture Tests using NSA for Link Prediction on the Amazon Dataset

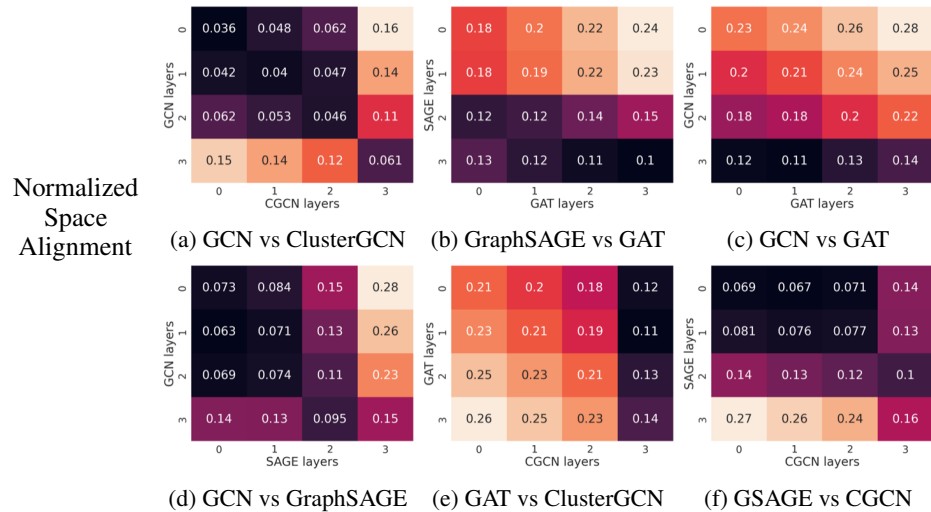

Figure 26: Cross Architecture Tests using NSA for Link Prediction on the Cora Dataset

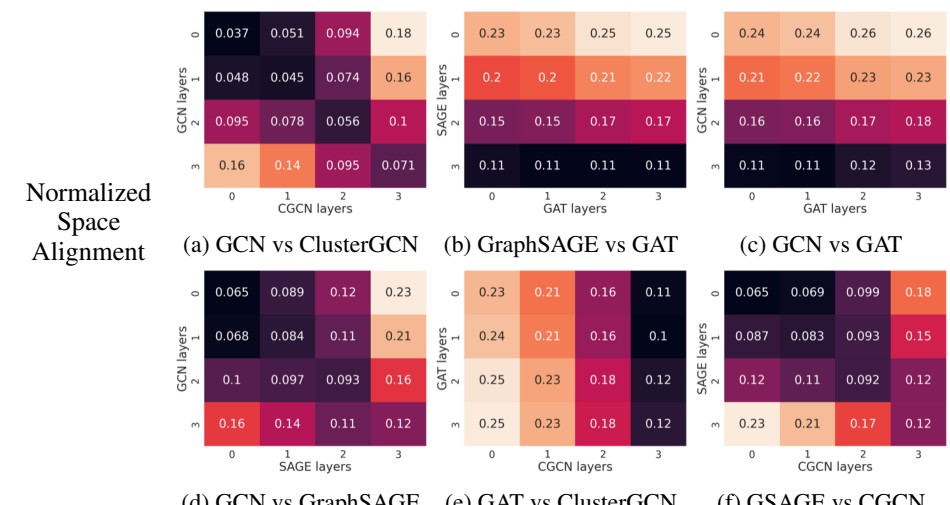

Normalized Space Alignment

(a) GCN vs ClusterGCN    (b) GraphSAGE vs GAT    (c) GCN vs GAT

(d) GCN vs GraphSAGE    (e) GAT vs ClusterGCN    (f) GSAGE vs CGCN

Figure 27: Cross Architecture Tests using NSA for Link Prediction on the Citeseer Dataset

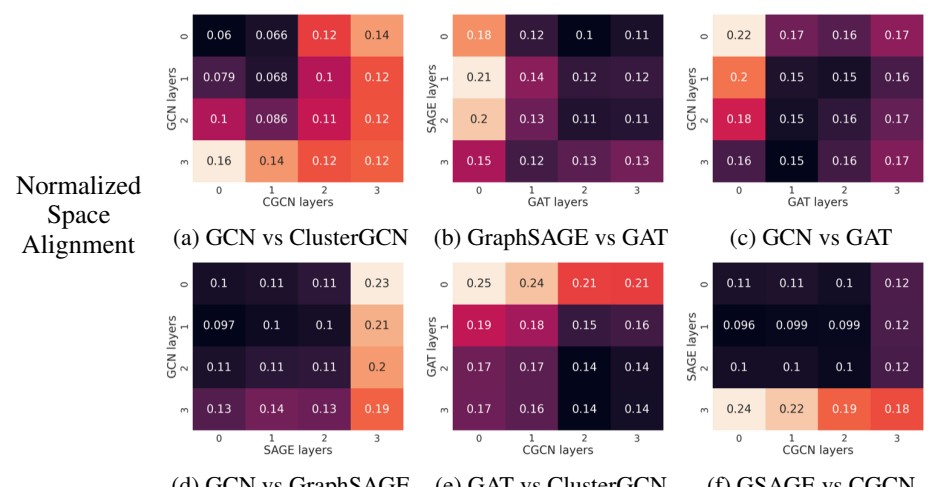

Normalized Space Alignment

(a) GCN vs ClusterGCN    (b) GraphSAGE vs GAT    (c) GCN vs GAT

(d) GCN vs GraphSAGE    (e) GAT vs ClusterGCN    (f) GSAGE vs CGCN

Figure 28: Cross Architecture Tests using NSA for Link Prediction on the Pubmed Dataset

## V.4   CROSS DOWNSTREAM TASK TESTS

We perform cross downstream task tests where we compare the layerwise intermediate representations between two models of the same architecture type but trained on different tasks. The two tasks we use are Node Classification and Link Prediction. The size of the intermediate embeddings is the same between the two models (Node Classification model vs Link Prediction model) except for the last layer. Since NSA is agnostic to the dimension of the embedding, we can compare all the layers regardless of their dimensionality. Just like the previous experiments, we test on the Amazon Computers Dataset and the three Planetoid Datasets. We observe that there is no strong correlation as we go further down the layers between the two models. The highest relative similarity is in the early layers and as we get to the final layers the similarity is almost non existent.

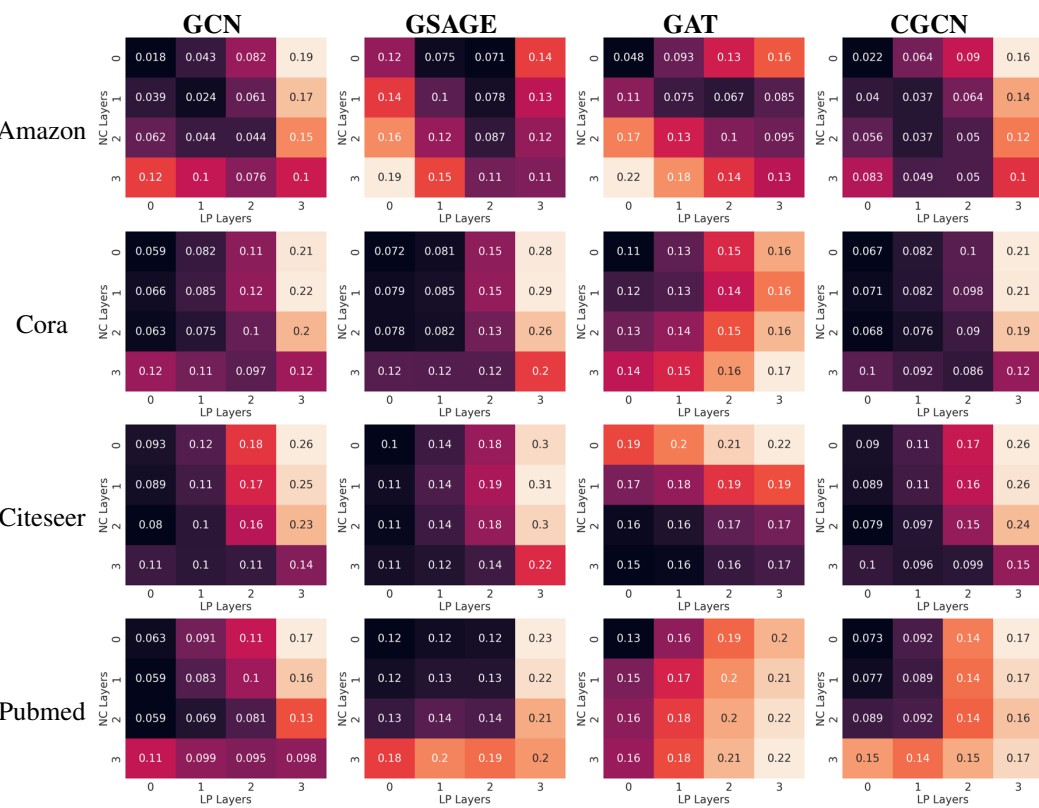

Figure 29: Cross Downstream Task Tests with NSA on the Amazon, Cora, Citeseer and Pubmed Datasets

# W    CONVERGENCE TESTS

As a neural network starts converging, the representation structure of the data stops changing significantly and the loss stops dropping drastically. Ideally, a similarity index should be capable of capturing this change. We examine epoch-wise convergence by comparing representations between the current and final epochs. We observe that NSA starts dropping to low values as the test accuracy stops increasing significantly. This shows that NSA can be used to help approximate convergence and even potentially be used as an early stopping measure during neural network training, stopping the weight updates once the structure of your intermediate representations stabilize. We show results for all 4 architectures on node classification and link prediction. Link Prediction models were trained for 200 epochs and show similar patterns to the ones observed with node classification. All convergence tests are performed with NSA only. We show the convergence tests on all 4 architectures on Node Classification in Figure 31 and on Link Prediction in Figure 30 .

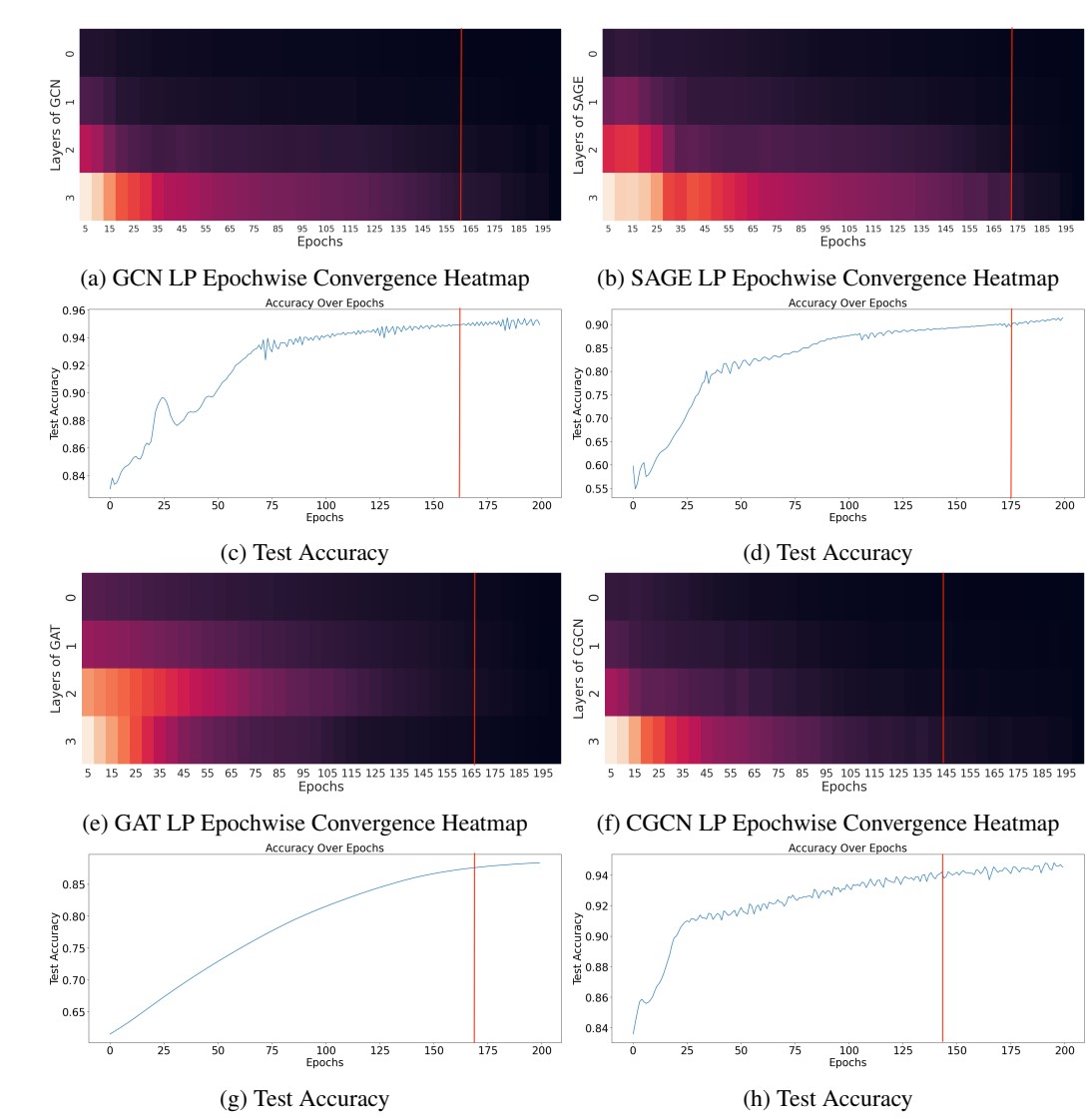

Figure 30: Convergence Tests on Link Prediction

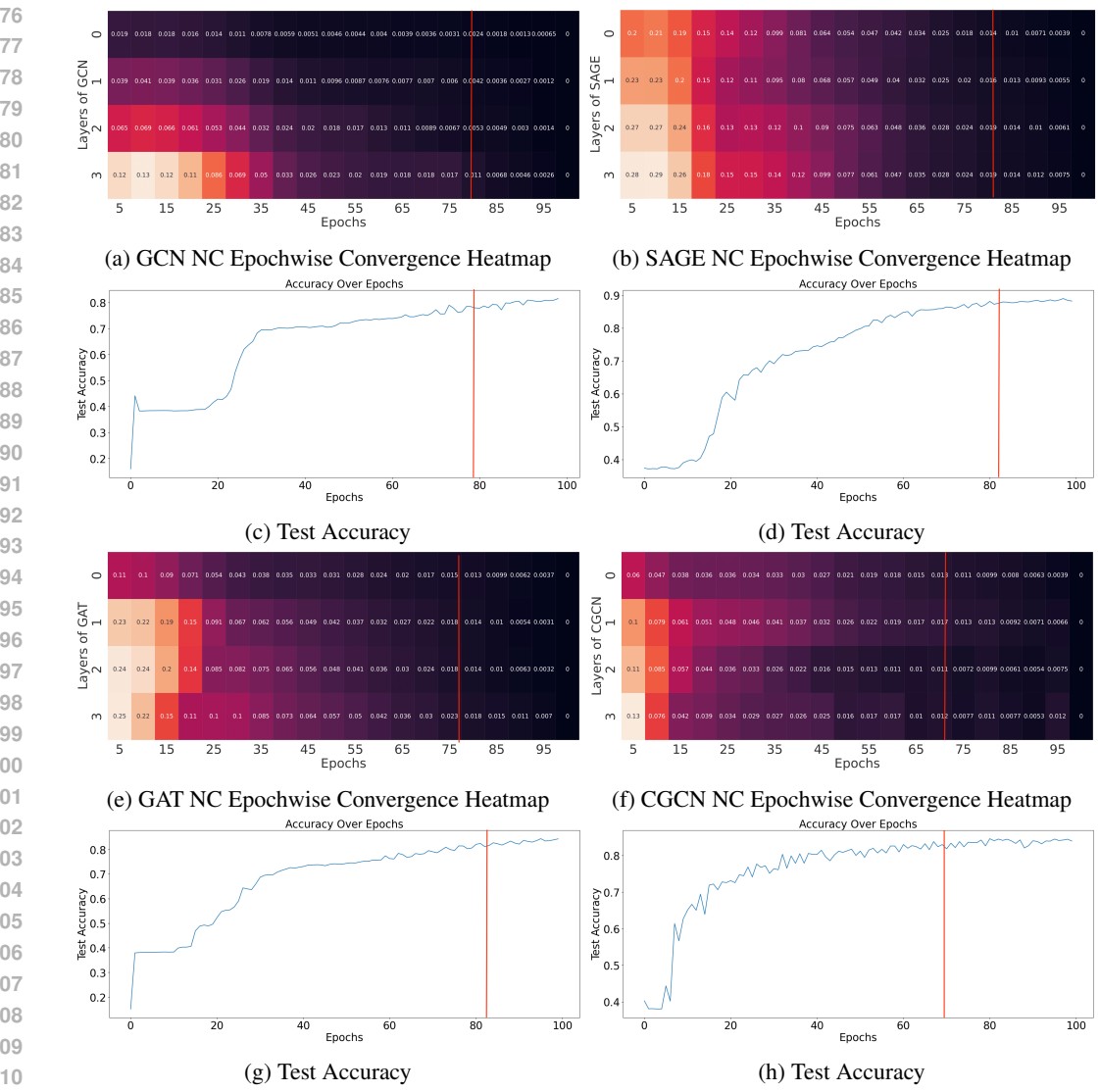

Figure 31: Convergence Tests on Node Classification

# X    RESIZED MAIN TEXT FIGURES

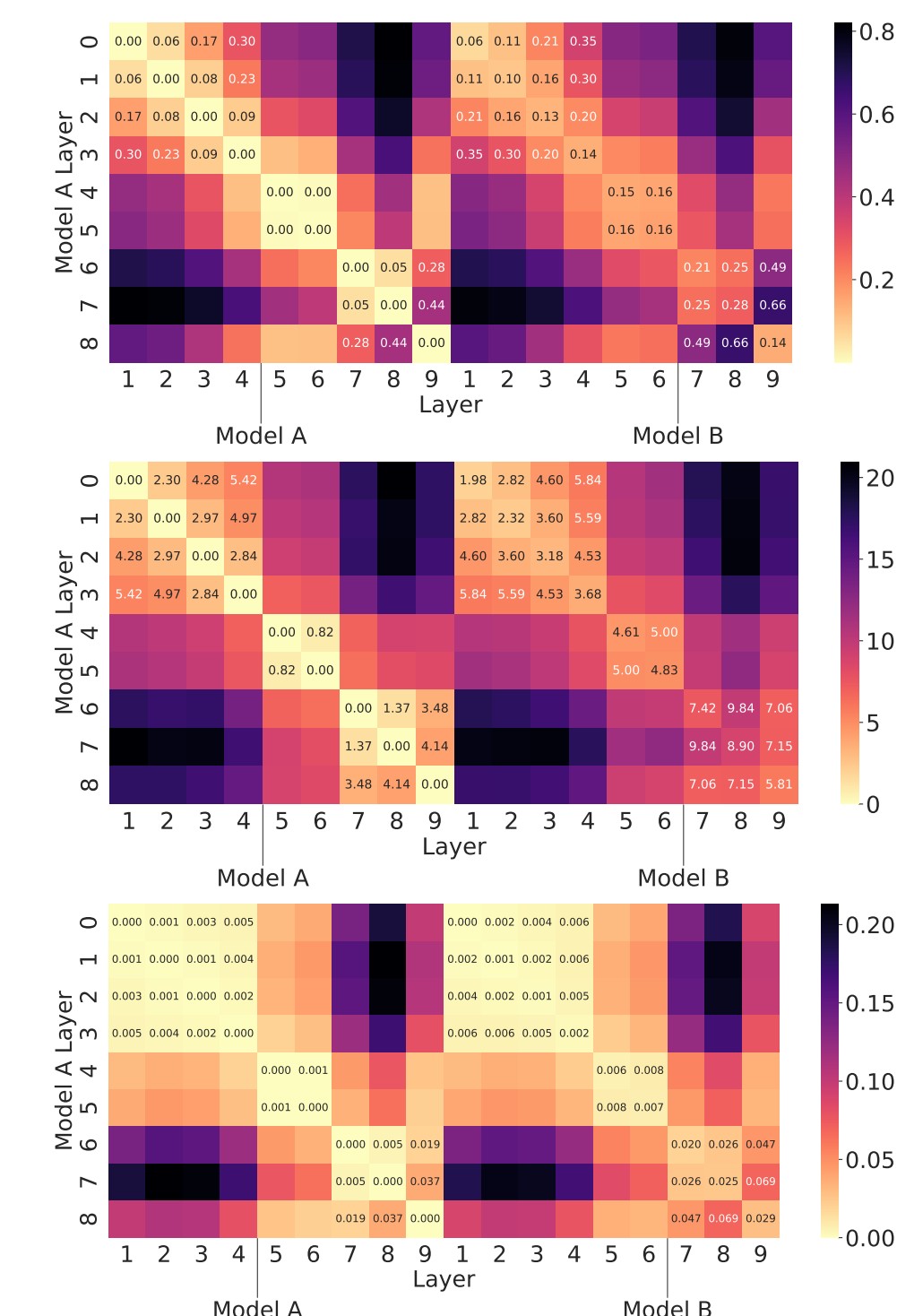

Figure 32: Specificity Tests Resized. From top to bottom: CKA', RTD and NSA pairwise distance between each layer of two differently initialized networks

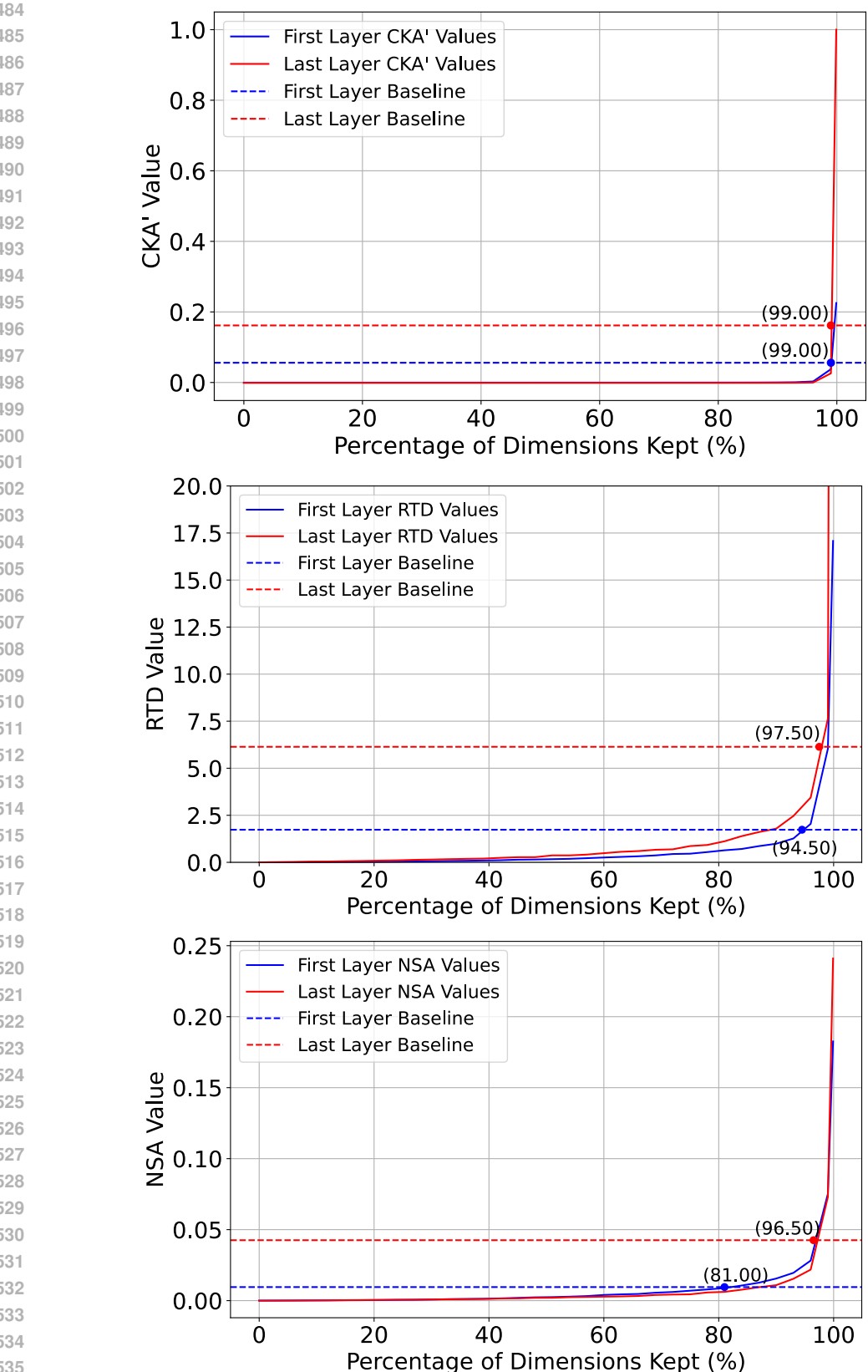

Figure 33: Sensitivity Tests Resized. From top to bottom: CKA', RTD and NSA sensitivity to removal of prinicipal components.

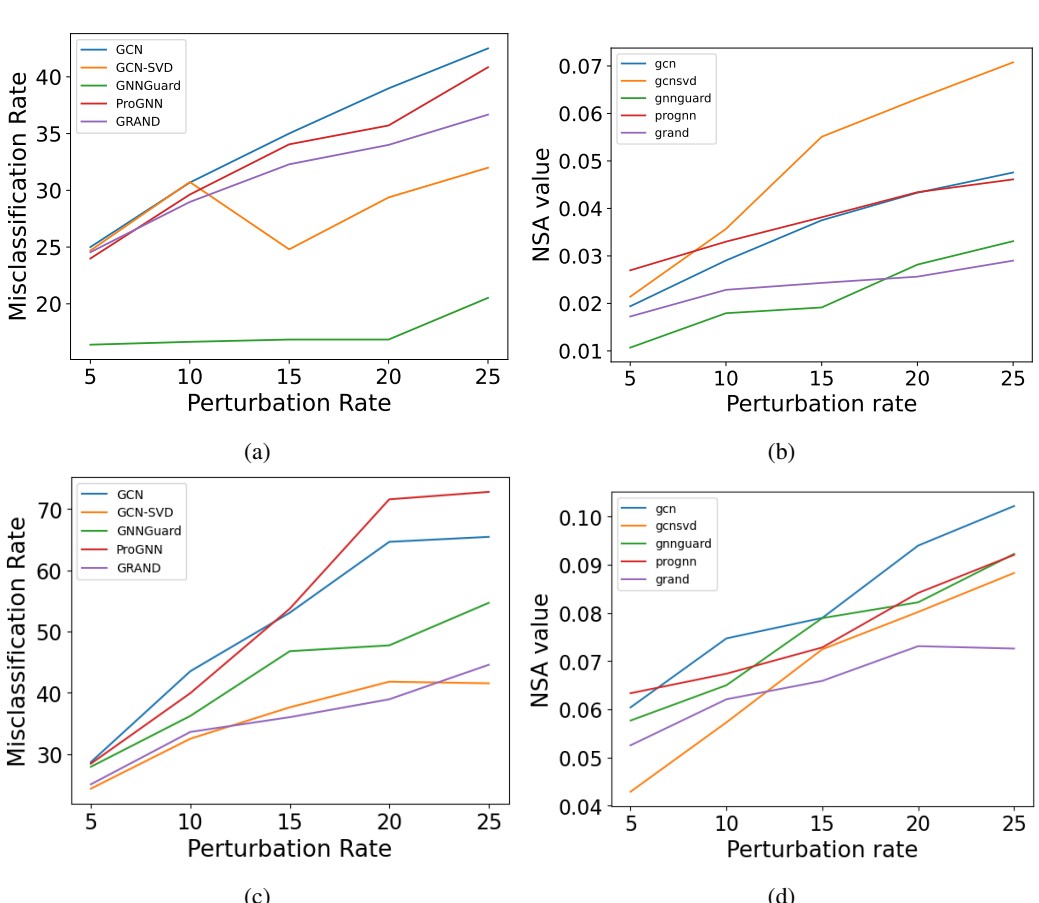

Figure 34: Robustness tests on GNN architectures with NSA resized. (a) Misclassification Rate against Data Perturbation Rate under global evasion attack. (b) NSA against perturbation rate under global evasion attack. (c) Misclassification Rate against Data Perturbation Rate under global poisoning attack. (d) NSA against perturbation rate under global poisoning attack

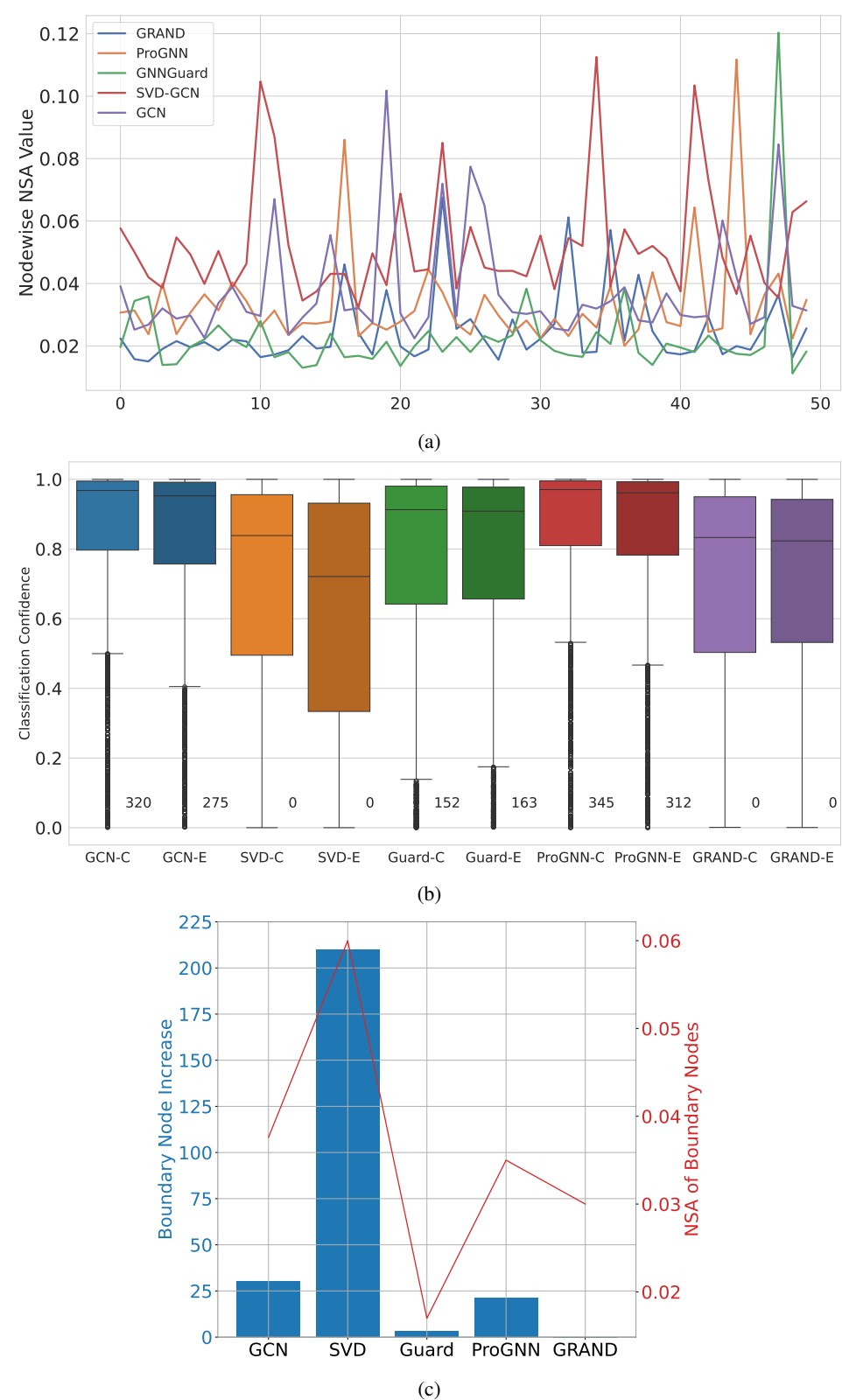

Figure 35: Analyzing node vulnerability with NSA resized. (a) Nodewise NSA of the 50 nodes with the greatest decline in Classification Confidence. SVD-GCN has the highest nodewise NSA variations. (b) Distribution of Classification Confidence before and after an evasion attack on various Graph Neural Network architectures. A suffix of 'C' after the architecture name refers to original dataset results and a suffix of 'E' refers to confidence on the poisoned dataset (c) Increase in number of boundary nodes for each model post attack and its correlation with the NSA of the boundary nodes.

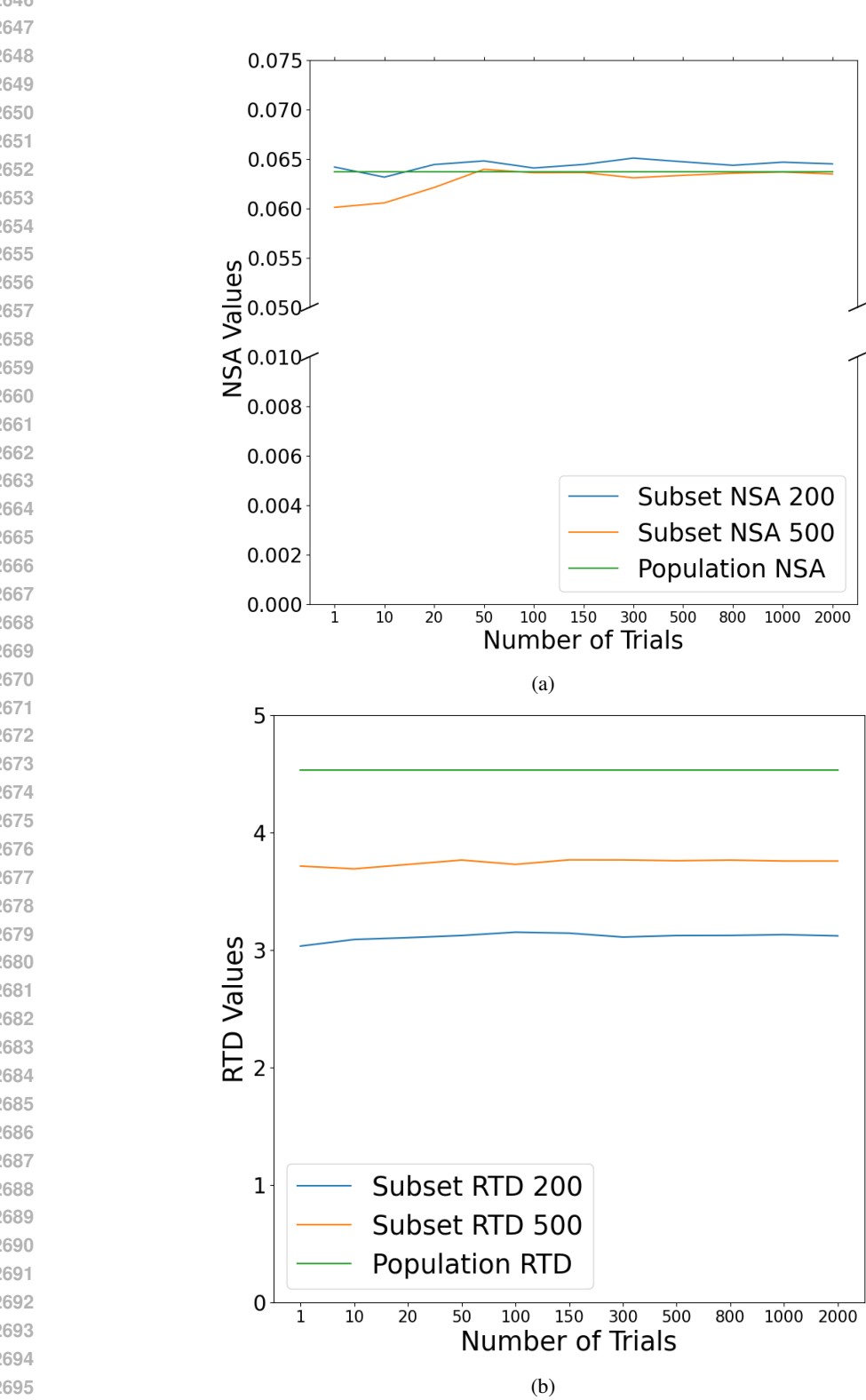

(a)

(b)

Figure 36: Expectation of subset metrics over a large number of trials resized. (a) Mean subset GlobalNSA variation over increasing trials. (b) Mean subset RTD variation over increasing trials.

## Y    INSTABILITY WITH THE ORIGINAL LID FORMULATION OF LNSA

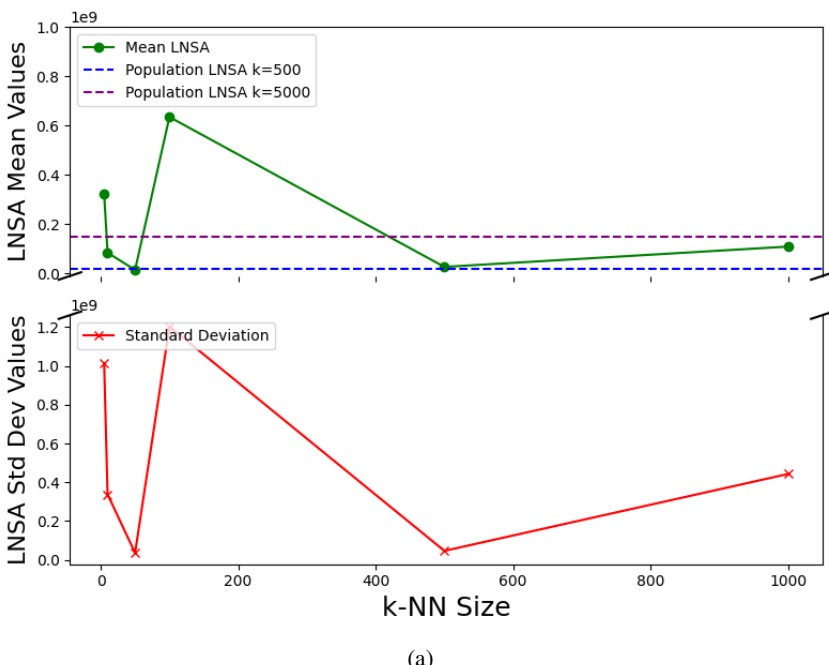

(a)

Figure 37: Variation of Mean and Standard Deviation over subsets for LNSA if we were to use the formulation of LNSA proposed by MacKay & Ghahramani (2005). Values are computed between the outputs of a ResNet-18 and a ResNet-34 on 50,000 ImageNet images. The X-axis varies the k value of the minibatch. The minibatch size is set to 5000. Not only does this variation of LNSA not show a clean pattern, the values are extremely high ( 1e8) and show erratic standard deviation.

# Z ADDITIONAL VISUALIZATIONS WITH 1D SPIRAL

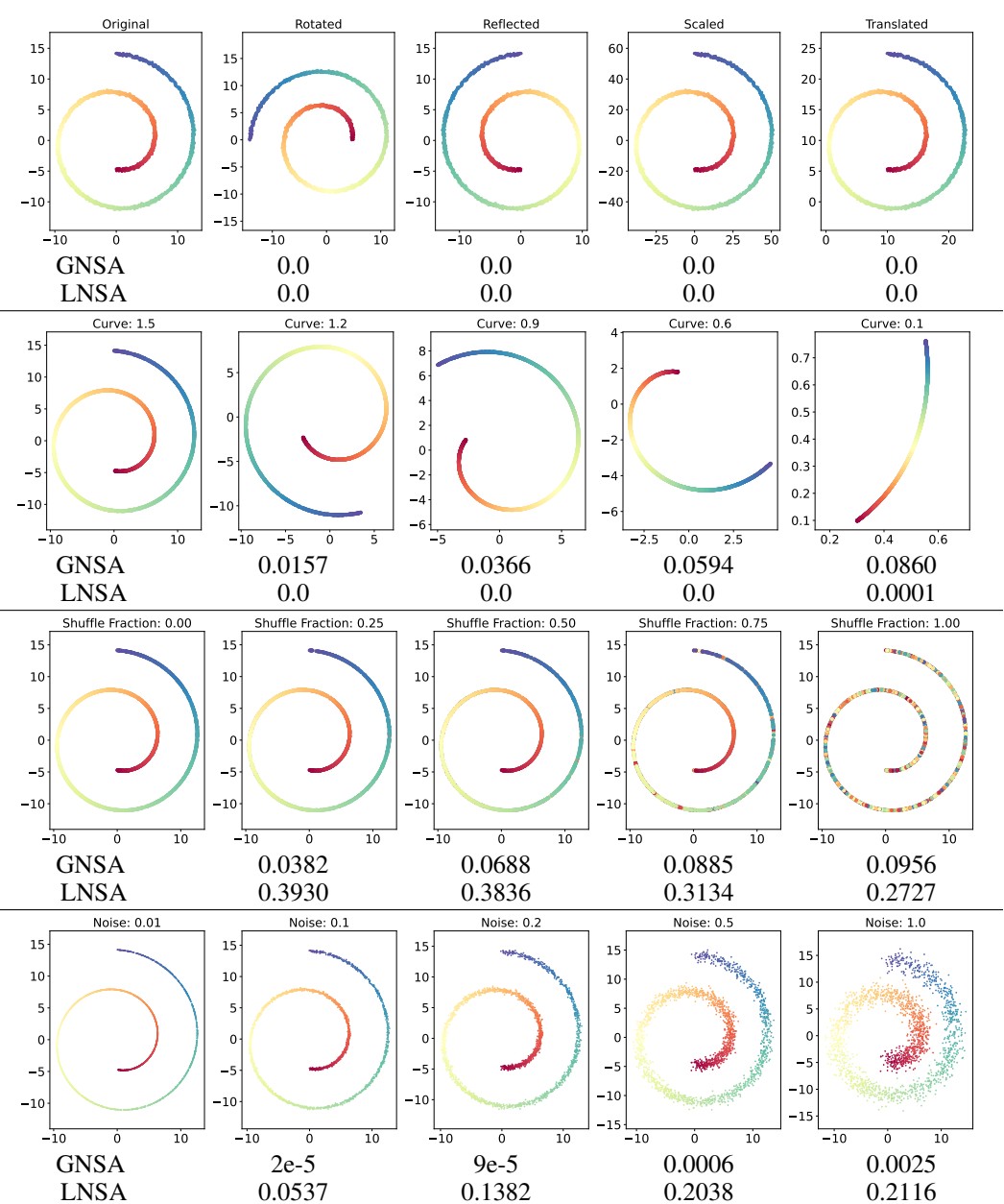

Figure 38: Comparison of LNSA and GNSA values after applications of various transformations on a 1D Spiral in 2D space. **Row 1 : Global Transformations**. We apply various global transformations to the spiral and compare the original spiral with the transformed spiral. As expected, both LNSA and GNSA are invariant to structure preserving global transformations. **Row 2: Manifold Preserving Transforms**. We progressively unravel the spiral and compare the original spiral to the unravelled spiral. As expected, LNSA remains largely invariant to this process while GNSA progressively increases. **Row 3: Structure altering transforms**. We progressively shuffle the data points in the spiral and compare the shuffled spiral against the original spiral. Both LNSA and GNSA return high dissimilarity values. **Row 4: Manifold altering transforms.** We progressively increase the noise in the spiral and compare against the original spiral. GNSA does not show a significant increase as global structure stays relatively similar but LNSA increases rapidly.

