# OpenReview forum: "Normalized Space Alignment: A Versatile Metric for Representation Analysis"
_ICLR.cc/2025/Conference — Submitted to ICLR 2025_

### Official Review · Reviewer_rMH7 · 2024-10-25

**Soundness:** 3
**Presentation:** 3
**Contribution:** 2
**Rating:** 5
**Confidence:** 4

**Summary:**

The authors introduce the Normalized Space Alignment (NSA) method for comparing two point clouds, which is based on comparing pairwise distances. The NSA consists of the local NSA, defined through the Local Intrinsic Dimensionality, and the global NSA, defined through Representational Similarity Matrices. The final NSA is defined as the weighted sum of global and local NSA. The experimental section includes experiments where NSA is used to analyze representations, as a loss in AE and for detection of adversarial attacks.

**Strengths:**

* Good background section on LID.
* Good applicability of the method.

**Weaknesses:**

General concerns:
* Despite a wide range of applications presented in the experimental section, the paper lacks comparison to relevant existing methods to really showcase the efficiency. For example, in the link prediction and adversarial attacks experiments, the method should be compared to the relevant baselines from the respective fields to be able to fairly judge the efficiency of the method.
* Datasets used in the experiments are small and basic, and the generalization of the method is questionable. How does the method behave for large sets and more complicated cases?
* No ablation studies are provided. For example, the method relies on the k nearest neighbors selection and I believe that the choice of k does influence the results. No experiments are provided on the robustness of k, neither is mentioned what k is actually used in the experiments. There is also no info on the balancing parameters l and g, and no ablation studies on the influence of these.
* The definition of GNSA depends on the choice of the origin. For example, given two point clouds X and Y, the translated point clouds will have the same structure but not the same GSNA score which is problematic. Of course one could resolve this with selecting a different origin but that is not feasible in practice.
* Figures are not well readable.

**Questions:**

Specific comments:
* Doesn’t the computation of GNSA depend on the specific order of the point clouds? For example, comparing a_i and b_i only make sense if these below to the same datapoint, otherwise you’re comparing random elements.
* In Sec 4.1 you claim that “a good structural similarity index should show high similarity between architecturally identical neural networks with different weight initializations”.  However, different initializations produce different models and there is no reason to assume that these should have the same structures. Also, in Figure 1 all the plots on the left are exactly the same. If this is not a typo, then I also don’t believe that the experiment shows what it is claimed. Additionally, the results here should be compared to the classical methods for comparing representations like Alaa et al, How Faithful is your Synthetic Data? Sample-level Metrics for Evaluating and Auditing Generative Models, Kynkäänniemi et al,  Improved precision and recall metric for assessing generative models, NeurIPS 2019, Poklukar et al, Delaunay Component Analysis, ICLR 2022, Khrulkov et al, Geometry score: A method for comparing generative adversarial networks, ICML 2018, etc, which are also missing from the related work.
* Please add details in 4.2.1. on how GSNA is even calculated. What is X and what is Y?
* In Sec 4.3., I do not understand why an AE is used on top of the produced embeddings. In my view, a baseline should be the classification accuracy on the embeddings of the GCN or alternatively of a NSA-GCN trained model but not of a frozen GCN model with an AE attached to it. Also, as mentioned above, this experiment lacks comparison to any SOTA graph based methods which makes the applicability questionable.

---

> ### Author Response · Authors · 2024-11-21
>
> We thank the reviewer for their feedback and are glad that they found the applicability of the method and background on LID satisfactory. We address their concerns below:
>
> **1. Lack of comparison to relevant existing methods and relevant baselines**
>
> Thank you for your feedback regarding comparisons to existing methods in link prediction and adversarial attack experiments. We appreciate the opportunity to clarify the intent and scope of these experiments.
>
> **General Scope of NSA:** NSA is primarily a metric for comparing representations across two spaces. It can explain the performance of machine learning tasks and serve as a differentiable loss function that can be efficiently estimated in minibatches. However, NSA itself does not prescribe how it should be applied to specific machine learning problems. Its usage, like that of other similarity metrics such as RTD and CKA, depends on the method or task in question.
>
> **Link Prediction Experiments:** The link prediction experiment was not designed to showcase NSA as a supplementary loss function for improving link prediction but rather to demonstrate NSA’s ability to retain the structural integrity of representation spaces after dimensionality reduction. For smaller models, retraining a GCN in the reduced dimension is feasible. However, for larger models where pretraining is computationally expensive, NSA-AE provides a more practical alternative by ensuring that the reduced-dimensional space preserves relative distances and contextual information. As shown in Table 2, the performance of NSA-AE on reduced-dimensional data is comparable to that of a newly trained model.
>
> In response to your suggestion, we have added results in Table 2 showing how a base GCN performs on link prediction as a reference. However, it is important to emphasize that this reference is not a baseline NSA is competing against but rather a contextual benchmark to understand NSA’s behavior. For more details, please refer to **Global Response 2**, where we discuss this experiment in depth.
>
> **Adversarial Analysis Experiments:** In the adversarial analysis experiments, NSA is not used as a defense mechanism but as a post-hoc analysis tool to evaluate the structural impact of attack and defense methods. While NSA could potentially serve as a structure-preserving loss function to facilitate robust neural network training, it is outside the scope of this manuscript which presents a breadth focused introduction to NSA. We leave the idea of using NSA to facilitate robust training of neural networks to future work. Instead, the goal is to introduce NSA’s ability to correlate with disagreements in predictions and to perform fine-grained, nodewise analysis.
>
> It is worth noting that NSA has a unique capability in this regard. Unlike the next best similarity metric, RTD, which analyzes topological features of spaces, NSA can examine individual data points and quantify their local discrepancies. This makes NSA particularly suited for tasks requiring high-resolution analysis of adversarial perturbations.
>
> **2. Datasets used in the experiments are small and basic, and the generalization of the method is questionable.**
>
> Thank you for your concern regarding the datasets used in our experiments. We address the generalizability and scalability of NSA by presenting results on datasets of varying scale and domain:
>
> 1. Dataset Variety: Our experiments span visualization-focused datasets (e.g., COIL-20, Spheres, Swiss Roll, Mammoth), benchmark datasets (e.g., MNIST, F-MNIST, CIFAR-10, Cora, Citeseer, Pubmed), and large-scale datasets (e.g., ImageNet, WordNet, Amazon Computers with 13K nodes and 500K edges, Flickr with 90K nodes and 900K edges). These datasets cover diverse domains, including images, natural language processing (NLP), and graphs. We provide detailed statistics for each dataset in Appendix P.
> 2. Scalability and Generalizability: The variety in our dataset choices demonstrates NSA’s scalability and generalizability across domains and tasks. Generalizability is evidenced by NSA’s effectiveness across different dataset types and scales, which is a key focus of this paper.
> 3. Theoretical Guarantees: In Section 4.2.1, we provide theoretical guarantees showing that GNSA converges to the global dataset value irrespective of dataset size when using mini-batching. This ensures NSA remains effective even for large-scale datasets.
> Additional Experiments:
> 4. Empirical results: In Appendix I, we present layerwise experiments on ResNet models evaluated on 100K ImageNet images.
> In Appendix Q, we provide ablation studies for GNSA and LNSA on ImageNet, showing that with mini-batching (using multiple trials on subsets), NSA can accurately approximate global values even for large datasets.
>
>
> We believe these results comprehensively demonstrate NSA’s scalability, generalizability, and robustness across a wide range of datasets and tasks.

---

> > ### Author Response · Authors · 2024-11-21
> >
> > **3. No ablation studies are provided**
> >
> > We would like to clarify that detailed ablation studies are provided in Appendix Q (Appendix P before revision), focusing on the effect of $l$ and $g$ on NSA. Our quantitative analysis demonstrates that while GNSA alone performs well for downstream tasks, combining it with LNSA yields even better results. Additionally, we include visual ablations using the Spheres dataset to show how different combinations of l and g influence the structure of complex representation spaces. As part of the revision, we have also added ablations visualizing the impact of different values of $k$ on NSA’s performance at various batch sizes, providing further insights into the method's parameter behavior.
> >
> > We kindly direct the reviewer to **Global Response 1** where we discuss the parameter tuning requirements for NSA. The presented ablations for $l$,$g$ and $k$ in Appendix Q show that NSA is mostly effective on a large range of hyperparameter values. We also added visual ablations in Appendix Q on the swiss roll dataset to demonstrate how the value of $k$ could potentially affect convergence.
> >
> > **4. Definition of GNSA depends on the choice of the origin.**
> >
> > Thank you for your comment. It is true that GNSA depends on the choice of origin, but this concern is easily manageable in practice and does not pose a significant limitation for NSA’s intended applications:
> > 1. **Normalization is Standard Practice:** In most neural network workflows, normalizing data to center it around the origin is a routine step, either as part of preprocessing or within the training pipeline. For example, when comparing two spaces, we can normalize both to share a common origin (e.g., centering them at (0,0)) without affecting the structural properties being measured. This ensures NSA scores remain invariant to translations.
> > 2. **Practical Scenarios:**
> > - In applications like dimensionality reduction, knowledge distillation, or training a lower-dimensional space using a higher-dimensional reference, the origin of the reference space is constant and can be explicitly accounted for. In these tasks, the exact position of the representation space is typically secondary to its structure, and alignment via normalization is sufficient to address origin dependency.
> > - When performing static analysis, NSA only cares about the structure of the space and not its absolute position. Since we always have access to the origin of the space, it is easy to normalize the space. NSA’s formula also accounts for the origin of the space not being 0,0 (as mentioned in footnote 2 on Page 4)
> >
> > **5. Figures are not readable.**
> >
> > Thank you for pointing this out. We have replaced some of the figures in the manuscript with higher quality figures. Could you please let us know which figures specifically have this issue so we can improve it?
> >
> > **6. Doesn't GNSA depend on specific order of point clouds?**
> >
> > The computation of GNSA indeed relies on a consistent ordering of the point clouds, as
> > a_i and b_i are assumed to correspond to the same data point in the two spaces being compared. This requirement is inherent to all pairwise structural similarity measures[1,2,3,4,5,6]. In practice, ensuring consistent ordering is straightforward in scenarios where the two spaces are derived from the same dataset (e.g., embeddings from two neural networks trained on the same data), the ordering is naturally preserved during computation.
> >
> > **7. Different initializations produce different models and there is no reason to assume these should have the same structures.**
> >
> > Thank you for your feedback. We direct the reviewer to **Global Response 3** where we address this query in detail.
> >
> > **8. In Figure 1 all the plots on the left are exactly the same.**
> >
> > Thank you for pointing this out. This is a typo. We have fixed this plot in the revised version of the manuscript.

---

> > > ### Author Response · Authors · 2024-11-21
> > >
> > > **9. The results here should be compared to the additional works referenced:**
> > >
> > > Thank you for suggesting these works.
> > >
> > > We reviewed [7, 8, 9] and decided they fall outside the scope of our current manuscript. While they provide valuable evaluation methods for generative models, they primarily focus on assessing the quality of a generated space relative to a ground truth space using metrics like precision, recall, and authenticity. These approaches differ fundamentally from NSA for the following reasons:
> > > 1. Not Structural Similarity Metrics: These methods evaluate the fidelity and diversity of generative outputs rather than measuring structural similarity between two spaces. As such, they do not fall strictly into the category of structural similarity indices (e.g., CKA[1], RTD[2], GBS[4], MTD[12], GULP[4]).
> > > 2. Non-Differentiability: The metrics rely on non-differentiable operations, such as graph-based manifold approximations and discrete counting, making them unsuitable for use as loss functions in gradient-based optimization pipelines. In contrast, NSA is fully differentiable and can be integrated seamlessly into training objectives.
> > > 3. Dependency on Ground Truth: All three methods rely on a ground truth representation space to compute precision, recall, and authenticity metrics. NSA, by design, measures structural similarity between two representation spaces, irrespective of whether one is "generated" or "ground truth," providing a symmetric quantification of discrepancies between the two spaces.
> > >
> > >
> > >
> > > Geometry Score [10]  is a topologically driven measure to compute differences between representation spaces, but it has several shortcomings [11,12].
> > > - Lack of Sensitivity: GScore fails to capture meaningful differences between distributions under simple transformations like shifts, scaling, or reflections. It also struggles to detect critical issues such as mode dropping or mode invention.
> > > - Stochasticity: Its reliance on approximate barcodes introduces significant variability, requiring thousands of repetitions for reliable results, which is computationally prohibitive.
> > > - Scalability: GScore is inefficient and impractical for high-dimensional datasets or modern large-scale applications.
> > >
> > > These shortcomings were addressed by Manifold Topology Divergence (MTD)[12], which improved upon GScore by providing a more robust and scalable topological analysis. Furthermore, Representation Topology Divergence (RTD)[2]—a direct successor to MTD from the same authors—refined and extended these improvements, making it the most robust and comprehensive method in this lineage.
> > >
> > > Since NSA is already compared extensively to RTD in our work, a comparison with GScore would be redundant and less informative. We focus on RTD because it not only improves upon GScore and MTD but also serves as a state-of-the-art method for structural analysis.
> > >
> > > We understand the importance of a comprehensive review of related work. However, we also believe it is critical to maintain a focused scope to ensure clarity and coherence in presenting our contributions. Including an extensive comparison with every evaluation metric that aims to enhance representation learning, especially those with a narrow focus on generative modeling, might divert attention from the novelty and broad applicability of NSA.
> > >
> > >
> > > **10. Details in 4.2.1. on how GSNA is calculated.**
> > >
> > > In this experiment, we extract output embeddings from two neural networks with identical architectures but different random initializations. These embeddings represent the model outputs for a given dataset. To evaluate the convergence of NSA, we compute GNSA on randomly sampled subsets of these embeddings and compare the subset-derived values to the global GNSA value computed on the entire dataset.
> > >
> > > We repeat this process multiple times for different subsets (of size 200 and 500) and observe how the subset-based GNSA approximates the global value as the number of trials increases. The results demonstrate that GNSA reliably converges to the global value with sufficient sampling, highlighting its robustness in mini-batch settings. In contrast, RTD fails to converge effectively, emphasizing the advantage of NSA in capturing global information of large datasets, even when working with subsets of data. We have added some clarification to the experimental setup in Section 4.2.1 of the revised manuscript.

---

> > > > ### Author Response · Authors · 2024-11-21
> > > >
> > > > **11.Clarification on Section 4.3**
> > > >
> > > > Thank you for raising this concern. As clarified in **Global Response 2**, the primary purpose of the link prediction experiment in Section 4.3 is to evaluate NSA’s ability to preserve structural integrity during dimensionality reduction, not to benchmark NSA as a state-of-the-art (SOTA) method for link prediction.
> > > >
> > > > 1. Why Use an AE: The autoencoder (AE) is used to test NSA’s effectiveness as a structure-preserving loss function during dimensionality reduction. It enables us to reduce the dimensionality of the embeddings produced by a frozen GCN model while preserving relative distances and structural integrity. The AE is not intended as a competitive link prediction framework but rather as a mechanism to facilitate the use of NSALoss.
> > > >
> > > > 2. Baselines: While a baseline of classification accuracy or link prediction using the raw GCN embeddings could be included, this would not serve the purpose of this experiment, which is to demonstrate how NSA ensures that reduced-dimensional embeddings retain their structural integrity. For reference, the revised manuscript's Table 2 includes the performance of a base GCN model directly trained on link prediction to provide context. However, this is not presented as a baseline that NSA competes against, as NSA’s goal in this experiment is not to improve link prediction but to demonstrate structure preservation during dimensionality reduction.
> > > >
> > > > 3. Comparison to SOTA Methods: Comparing NSA to SOTA graph-based methods would be outside the scope of this experiment, as NSA is not a graph-specific model but a general structural similarity metric. As discussed in **Global Response 2**, NSA can serve as a supplementary loss function or metric for a variety of tasks, but it is not designed to replace task-specific SOTA methods.
> > > >
> > > > We hope this clarifies the intent of Section 4.3 and its focus on dimensionality reduction and structural preservation. For a more detailed discussion, we encourage you to review Global Response 2.
> > > >
> > > > We hope we have clarified all the queries the reviewer had. Please let us know if you have any more concerns.
> > > >
> > > >
> > > >
> > > > [1] Simon Kornblith, Mohammad Norouzi, Honglak Lee, and Geoffrey E. Hinton. Similarity of neural
> > > > network representations revisited. IN ICML 2019
> > > >
> > > > [2] Serguei Barannikov, Ilya Trofimov, Nikita Balabin, and Evgeny Burnaev. Representation topology
> > > > divergence: A method for comparing neural network representations. In ICML 2022
> > > >
> > > > [3] Enric Boix-Adsera, Hannah Lawrence, George Stepaniants, and Philippe Rigollet. 2022. GULP: a prediction based metric between representations. In NeurIPS.
> > > >
> > > > [4] Zuohui Chen, Yao Lu, Wen Yang, Qi Xuan, and Xiaoniu Yang. 2021. Graph-Based Similarity of Neural Network Representations. ArXiv preprint (2021)
> > > >
> > > > [5] Trofimov, I., Cherniavskii, D., Tulchinskii, E., Balabin, N.,Burnaev, E., and Barannikov, S. Learning topology preserving data representations. In The Eleventh International Conference on Learning Representations, 2023.
> > > >
> > > > [6] Klabunde, M., Schumacher, T., Strohmaier, M., and Lemmerich, F. Similarity of neural network models: A survey of functional and representational measures.
> > > >
> > > >
> > > > [7] Alaa et al, How Faithful is your Synthetic Data? Sample-level Metrics for Evaluating and Auditing Generative Models,
> > > >
> > > > [8] Kynkäänniemi et al, Improved precision and recall metric for assessing generative models,
> > > >
> > > > [9] Poklukar et al, Delaunay Component Analysis, ICLR 2022,
> > > >
> > > > [10] Khrulkov et al, Geometry score: A method for comparing generative adversarial networks, ICML 2018
> > > >
> > > > [11] Ali Burji, Pros and Cons of GAN Evaluation Measures: New Developments
> > > >
> > > > [12] Serguei Barannikov et al. Manifold Topology Divergence: a Framework for Comparing Data Manifolds Neurips 2021

---

> > > > ### Comment · Reviewer_rMH7 · 2024-11-25
> > > >
> > > > 9. I understand that these methods are not Structural Similarity Metrics but in the experiment in 4.1. you are directly analysing the representation space, thus I do not see a reason why these methods couldn't be used. It would be a great way to highlight the strengths of your method.
> > > >
> > > > 10. Thank you for clarification and adding those details.

---

> > > ### Comment · Reviewer_rMH7 · 2024-11-25
> > > **Thank you for the excessive rebuttal**
> > >
> > > First of all, I appreciate the effort the authors put into this extensive rebuttal. I have a few follow-up comments and questions:
> > >
> > > 3.
> > > - I do not understand what Figure 11 is showing, could you please provide context?
> > > - To me, in Figure 13, the influence of the choice of k is big. I can image this is even more problematic in higher dimensions. I also do not understand you guidance on selecting k (in line 1695). According to this formula, the optimal choice for k in Figure 13 should be 128/100 which is not the case.
> > >
> > > 5. In my opinion, all figures have a too small font size (Fig 1 numbers aren't readable, Fig 2 the y axis numbers, Fig 3 the legend isn't readable, same for Fig 4 and others)

---

> ### Author Response · Authors · 2024-11-25
>
> We sincerely thank the reviewer for taking the time to review our rebuttal in detail. We present responses to their queries below:
>
> **1. I do not understand what Figure 11 is showing, could you please provide context?**
>
> Thank you for your feedback. Figure 11 illustrates the effect of varying the parameters $l$ and $g$ in the NSA formulation. The experiment involves reducing the Spheres dataset (comprising 10 spheres within an 11th sphere) from its original 100 dimensions to 2 dimensions. Figure 15 (c) plots the first 3 dimensions of the original dataset for reference.
>
> Using an NSA-AE trained solely with the NSALoss, we systematically vary $l$ and $g$ to observe how these parameters influence the dimensionality reduction visually. The figure demonstrates how different combinations of $l$ and $g$ affect the structural preservation of the dataset during this reduction.
>
> We have added additional explanatory text to the relevant subsection and caption to clarify this further. We hope this resolves any confusion regarding the purpose and interpretation of Figure 11.
>
> **2. The influence of the choice of k and clarifications on selecting k.**
>
> Figure 13 demonstrates how LocalNSA performs when varying both the $k$ value and the width of the Swiss Roll. As the width increases, we observe that a slightly larger $k$ value is required to preserve the manifold effectively. This occurs because increasing the width spreads the data points further apart, reducing the density of local neighborhoods. When $k$ is too small, the immediate neighbors of a point may fail to fully capture the local structure, leading to distortions in the reconstructed manifold. By slightly increasing $k$, we include more neighbors, compensating for this sparsity and ensuring that the manifold remains well-preserved.
>
> LocalNSA’s primary objective is to preserve local neighborhoods, and despite the changing width in Figure 13, it successfully achieves this for a wide range of $k$ values, failing slightly only when $k$ is extremely small and width is very high. The minimum requirement for preserving the manifold of the Swiss Roll is maintaining its 2D plane structure and local point neighborhood consistency. The variations in global structure observed in Figure 13 arise because LocalNSA operates without access to global information. Consequently, it converges as soon as it reconstructs the manifold perfectly. This reconstruction does not necessarily have to resemble the Swiss Roll’s original 3D geometry, as long as the manifold’s local properties are preserved.  We demonstrate this using a gradient-based coloring scheme: Each point on the Swiss Roll is labeled with a color gradient that changes minimally along the manifold. In the reconstructed figure, this gradient remains consistent, confirming that while the global geometry may be distorted, the local manifold structure is effectively preserved.
>
> Almost all the variability across the $k$ values in Figure 13 can be explained by the fact that LNSA does not preserve global structure and that the swiss roll does not align the same way in different reconstructions (since LNSA is invariant to scaling and transformations it does not preserve these when reconstructing the structure) with some minor variance explained due to the increasing width of the swiss roll.
>
> ***On selecting k***
>
> The guidance provided in line 1695 is not for determining an optimal absolute $k$, but rather for maintaining an optimal relative $k$ when working with mini-batches. Specifically:
> - If the dataset size is $N$ and you wish to preserve local neighborhoods up to $k_{global}$ nearest neighbors in the full dataset, the value of $k$ in mini batches ($k_{mini}$) should be chosen to maintain the following proportion:
> $$\[
> \frac{N}{k_{\text{global}}} = \frac{\text{mini-batch size}}{k_{\text{mini}}}
> \]$$
> This ensures that the mini-batch LNSA value approximates the global LNSA value
>
> - There is no specific value of $k$ that works "best" for the swiss roll experiment in Figure 13 as we do not place any constraints on approximating the mini batch LocalNSA
> - For example, if we were working with the Swiss Roll dataset from Figure 13 and had specific local neighborhood consistency constraints, for a full dataset size of 20480 and a mini batch size of 128, we would set the values using the following proportionalities:
>   - Set $k_{mini} = 2$ if we want $k_{global} = 320$ (dataset size: $k_{global}$ = 64)
>   - Set $k_{mini} = 5$ if we want $k_{global} = 800$ (dataset size: $k_{global}$ = 25.6)
>   - Set $k_{mini} = 10$ if we want $k_{global} = 1600$ (dataset size: $k_{global}$ = 12.8)
>
> Thus if you had a specific degree of local neighborhood preservation you wanted to guarantee on your dataset, you could do so by appropriately adjusting the $k$ value to the mini-batch size. But for most datasets we observed that a wide range of $k$ values work very similarly in practice.

---

> > ### Author Response · Authors · 2024-11-25
> >
> > **3. Readability of figures**
> >
> > Thank you for your feedback regarding the font sizes in the figures. We acknowledge this concern and are actively working on improving the figures for better readability. Once we are aware of all the necessary changes for the revised version, we will update the main body of the manuscript to reflect these improvements.
> >
> > As a temporary measure, we have included larger versions of all the figures from the main text in Appendix X of the latest revision to the manuscript. We hope this helps address your concerns in the interim.
> >
> > **4. Additional metrics for Section 4.1**
> >
> > Thank you for your suggestion. While we understand the motivation behind applying sensitivity and specificity tests to Generative model evaluation metrics, we believe these metrics are not well-suited for such analyses. This is because these methods produce multiple evaluation metrics (e.g., precision, recall, authenticity, consistency, quality etc) rather than a single global measure of structural similarity, which makes them less directly comparable to NSA.
> >
> > However, we are currently working on adapting the sensitivity and specificity tests from Section 4.1 to Delaunay Component Analysis (DCA)[1]. DCA is the most recent of these metrics and is designed to address some of the limitations in earlier methods like IPR[2] and Geometry Score[3]. As highlighted in Section 2 of the DCA paper, both IPR and Geometry Score suffer from inaccuracies and limitations, which informed our decision to prioritize DCA for this analysis.
> >
> > We are unsure if these experiments will be completed within the rebuttal period but will update the manuscript with the results as soon as possible.
> >
> > We hope our response clarifies the queries of the reviewer. Please feel free to ask any additional questions you might have.
> >
> > [1] Poklukar et al, Delaunay Component Analysis, ICLR 2022
> >
> > [2] Kynkäänniemi et al, Improved precision and recall metric for assessing generative models, in Neurips 2019
> >
> > [3] Khrulkov et al, Geometry score: A method for comparing generative adversarial networks, ICML 2018

---

> ### Author Response · Authors · 2024-11-26
> **Result of experiments on Generative Model Evaluation Metrics**
>
> While experimenting with DCA to evaluate specificity and sensitivity tests, we encountered several limitations in its current definition.
> - DCA and similar generative model evaluation metrics are unable to compute $\text{metric}(A, B)$ when $\text{dim}(A) \neq \text{dim}(B)$. This restriction prevents their use in generating specificity heatmaps, as they can only compute values across corresponding layers (i.e., along the diagonal). The inability to evaluate intermediate embeddings for mismatched dimensionalities severely limits our ability to evaluate their performance against structural similarity metrics.
> - Furthermore, during our experiments, we observed that the precision and recall values from DCA dropped drastically as the dimensionality of the embeddings increased. For example, while the final layer embeddings (10 dimensions) yielded precision and values around 0.98, the first layer embeddings (4096 dimensions) return a recall and precision of 0. This drastic disparity makes it nearly impossible to generate meaningful sensitivity plots across layers.
> - Additionally, given the same experimental setup, we found DCA to be significantly inefficient even when using GPUs. Computing the metrics once on a subset of 3000 points in 4096 dimensions takes ~7 minutes. This is magnitudes higher than the compute cost of NSA, which takes only a few seconds for larger subsets and much larger dimensionalities (we present empirical runtimes on 200,000 dimensional data in Appendix H).
> - The reliance on multiple separate metrics (precision, recall, authenticity etc) rather than a single global similarity measure further complicates the process of comparison and their utility in structural similarity tasks.
>
> These limitations make us believe DCA and other generative model evaluation metrics are best suited to compare output embeddings only and are ill-suited for comparison with structural similarity metrics like NSA, CKA and RTD, which are designed to evaluate and quantify discrepancies across spaces irrespective of dimensionality differences or the specific layer being analyzed.
>
> We hope this satisfies the reviewer's concerns regarding comparison of NSA to these metrics. Please let us know if you have any additional questions.

---

### Official Review · Reviewer_YeRH · 2024-11-04

**Soundness:** 2
**Presentation:** 2
**Contribution:** 2
**Rating:** 3
**Confidence:** 4

**Summary:**

The paper introduces a method (NSA) for comparing two data representations of the same dataset. NSA is a weighted sum with some tuned weights of GNSA which essentially compares the pairwise euclidian distances in the two representations of the same points, and of LNSA which is a local dissimilarity measure, based on k-NN graph. Experiments are described in order to empirically validate the expected properties of the method, although no access to the source code is provided.

**Strengths:**

-The paper addresses an interesting problem of constructing a reasonable measure of dissimilarity of two data representations of the same dataset.

-Different experiments are described in order to empirically validate the method, although no source code is provided making reproducibility check difficult.

**Weaknesses:**

1.Dependence on Euclidean distance in high-dimensional spaces. NSA uses essentially the comparison of Euclidean distances as its measure of structural similarity. This choice can be suboptimal in high-dimensional spaces due to the "curse of dimensionality," which makes Euclidean distances less informative and can lead to unreliable similarity measurements.

2.An access to the source code is not provided making the paper results reproducibility check difficult.

3.Lack of universality without parameter tuning. NSA's performance across different tasks relies heavily on parameter tuning and specific integration with other loss functions.  The choice of k in the construction of k-nn graph is essential in the definition of LNSA. The weights in front of the local and global parts of NSA clearly lead to drastically different results depending on their values.

4.No thorough guidance is provided for the choices and tuning of these hyperparameters. For example how to do 'appropriately adjusting the number of nearest neighbors considered in each mini-batch' on line 366 remains unspecified.

5.High computational complexity for large datasets. Despite claims of efficiency, NSA has a quadratic computational complexity concerning the number of data points, \( O(N^2 D + kND) \). This can become prohibitively expensive as the dataset size grows.

6.The method's focus on structural preservation might make it less effective in scenarios where functional similarity is more relevant, limiting its applicability.

7.Absence of interpretability mechanisms for practical applications. Although NSA provides a structural similarity measure, it lacks interpretability features that could make its outputs more useful in real-world applications. For instance, it does not offer insights into which specific features or dimensions contribute most to the observed structural discrepancies.

**Questions:**

Why an access to the source code was not provided for reproducibility check purposes?

---

> ### Author Response · Authors · 2024-11-21
>
> We thank the reviewer for their feedback. We address their concerns below:
>
> **1 Dependence on Euclidean distance in high dimensional spaces might not be effective in high dimensional spaces**
>
> We appreciate the reviewer’s observation regarding the potential limitations of Euclidean distance in high-dimensional spaces due to the curse of dimensionality. While it’s true that Euclidean distance can face challenges in high-dimensional settings, we chose it deliberately for a few key reasons:
>
> 1. **Prevalent Use Across Neural Network Tasks:** Euclidean distance remains a widely adopted metric in representation learning, particularly in tasks involving contrastive and triplet losses, where it has shown strong empirical performance. Many state-of-the-art neural network architectures still rely on Euclidean distance for computing similarity between embeddings, even in high-dimensional spaces.
> 2. **Empirical Robustness:** In our experiments, NSA demonstrated robust performance across multiple datasets and tasks, even when applied to high-dimensional embeddings (we test up to 200,000 dimensions). This suggests that, while Euclidean distance may not be theoretically optimal under the curse of dimensionality, it continues to yield practical benefits in diverse applications, including representation analysis and alignment tasks.
> 3. **Potential for Adaptability:** The NSA framework is flexible enough to accommodate alternative distance metrics if future work finds this necessary. In this study, we prioritized Euclidean distance for its simplicity, efficiency, and empirical success in maintaining local and global structural alignment across different model architectures.
>
> Additionally we include a local component for NSA for this specific purpose. LocalNSA only looks at the k-nearest neighborhood of a point and can alleviate some of the discrepancies that can arise from GlobalNSA running at high dimensionalities.
>
> **2. No Access to source code**
>
> Thank you for your feedback. We do provide access to our source code in Appendix T (Appendix R in the un-revised version), where detailed instructions for reproducing our results are included. We will update the codebase with the experiments performed for the revision and update the anonymous repository soon.
>
> **3. Lack of universality without parameter tuning. NSA's performance across different tasks relies heavily on parameter tuning and specific integration with other loss functions. The choice of k in the construction of k-nn graph is essential in the definition of LNSA. The weights in front of the local and global parts of NSA clearly lead to drastically different results depending on their values.**
>
> Thank you for your feedback. We kindly direct the reviewer to **Global Response 1** where we discuss the parameter tuning requirements for NSA. We present extensive ablations for $l$,$g$ and $k$ in Appendix Q where we show that NSA is mostly effective on a large range of values. We also added visual ablations in Appendix Q on the swiss roll dataset to demonstrate how the value of $k$ could potentially affect convergence.
>
> **4. No thorough guidance is provided for the choices and tuning of these hyperparameters. For example how to do 'appropriately adjusting the number of nearest neighbors considered in each mini-batch' on line 366 remains unspecified.**
>
> Thank you for your feedback. We have included an experiment in Appendix Q demonstrating empirical convergence of LNSA with the right value of $k$ and best practices for selecting $k$. As stated in Section 4.2.1 LNSA does not formally converge with mini-batching but has several favorable properties that help it perform well. For example, consider the scenario without mini-batching, where we examine a k-sized neighborhood of each point. When using mini-batches of size N/10, in expectation, we have k/10 points from the original neighborhood present in each mini-batch. Therefore, using LNSA with the number of nearest neighbors set to k/10 results in comparable outcomes to the non-mini batched case. By extension this applies to any fraction. We demonstrate this in Figure 12.b in Appendix Q.2

---

> > ### Author Response · Authors · 2024-11-21
> >
> > **5. High computational complexity for large datasets. Despite claims of efficiency, NSA has a quadratic computational complexity concerning the number of data points, ( O(N^2 D + kND) ). This can become prohibitively expensive as the dataset size grows.**
> >
> > While NSA has quadratic computational complexity with respect to the number of data points, it remains feasible and efficient for large datasets due to its compatibility with mini-batch training. As discussed in Section 4.2.1, NSA’s global measure can be effectively approximated using mini-batches, similar to stochastic gradient descent (SGD). Lemma G.1 further proves that the expectation of GNSA over mini-batches equals the GNSA of the entire dataset, ensuring that structural discrepancies are accurately captured even in mini-batch settings. With additional experiments in the revised manuscript we demonstrate that this applies for LNSA too. We would like to emphasize that this property of converging under mini batching is unique to NSA and unlike other measures. We show in 4.2.1 that RTD can never converge to its global value as it will always miss the ideal global topological feature when working with mini batches. Additionally, most similarity metrics are not even differentiable and therefore cannot be used as a loss function at all.
> >
> > This property keeps NSA’s complexity manageable regardless of dataset size, as long as the mini-batch sizes are not exceedingly large. Practitioners can work with arbitrarily large datasets by iterating over smaller, feasible chunks. As shown in the runtime analysis in Appendix H, NSA computation remains efficient: even for large batch sizes (up to 5000) and very high-dimensional data (~200,000), NSA takes only a few seconds per batch.
> >
> > **6. The method's focus on structural preservation might make it less effective in scenarios where functional similarity is more relevant, limiting its applicability.**
> >
> > Thank you for pointing this out. The reviewer is correct in that NSA's focus on structural preservation makes it less effective in functional similarity scenarios. NSA is designed as a structural similarity metric, with the goal of preserving and analyzing the geometric and topological properties of representation spaces. Focusing on structural preservation inherently limits the ability to directly measure functional similarity, as the two concepts often emphasize different aspects of representation. This tradeoff is intrinsic to similarity metrics: prioritizing structural fidelity can make functional relationships harder to capture, and vice versa. Both approaches have distinct strengths and applications.
> >
> > We also present experiments on layerwise analysis across architectures in Appendix V, where two GNN architectures are trained on the same dataset and task, making them functionally similar but NSA does not identify layerwise correspondence perfectly. Despite this, we demonstrate in our work that structural similarity is highly relevant and viable for a wide range of applications, including dimensionality reduction, adversarial robustness analysis, neural network similarity analysis and representation alignment. In scenarios where functional similarity is more critical, other metrics designed specifically for that purpose may be more appropriate.
> >
> > **7. Absence of interpretability mechanisms for practical applications.**
> >
> > Thank you for your feedback. We would like to clarify that NSA does include interpretability mechanisms. Both components of NSA, LNSA and GNSA, are aggregates of nodewise discrepancies. Specifically, GNSA is an aggregate of nodewise global discrepancies, while LNSA aggregates local nodewise discrepancies. By examining the nodewise formulations of NSA (e.g., Equation 8 presents the nodewise equation for GNSA, and LNSA can be similarly expanded), it is possible to pinpoint exactly which nodes contribute most to the overall structural discrepancy.
> >
> > We demonstrate this interpretability feature in Section 4.4, where we perform a node wise analysis of perturbations. This analysis highlights why SVD-GCN exhibits poor structural similarity and heightened vulnerability, despite showing only a small drop in accuracy when perturbed with adversarial edges. We also demonstrate this for CNNs using ResNets in Appendix I. NSA can identify which images have been perturbed and quantify their contribution to the overall discrepancy. Such pointwise insights make NSA particularly useful for interpreting structural changes in practical applications.
> >
> >
> > We hope our response clarifies the queries of the reviewer. Please feel free to ask any questions you might have about our work.

---

> > > ### Author Response · Authors · 2024-11-26
> > > **Friendly Reminder**
> > >
> > > Dear Reviewer,
> > >
> > > The revision period for the rebuttal will be ending soon. We would be extremely grateful if you could take the time to review our rebuttal and let us know if it has resolved all of your concerns. If you have any further questions, we would be happy to answer them.
> > >
> > > Sincerely,
> > >
> > > The Authors

---

### Official Review · Reviewer_1hcq · 2024-11-04

**Soundness:** 2
**Presentation:** 3
**Contribution:** 3
**Rating:** 3
**Confidence:** 4

**Summary:**

The paper introduces Normalized Space Alignment (NSA), a novel metric to analyze neural network representations. NSA compares two point clouds (representing data structures within neural networks) by preserving both global and local structures, regardless of differing dimensionalities. It can be used to preserve representation structures in tasks such as suitable for diverse tasks such as dimensionality reduction, adversarial robustness assessment, and cross-layer representation analysis. NSA’s main advantage is its ability to efficiently preserve global and local structure across different dimensional spaces. The authors showcase NSA’s versatility by applying it across various tasks, demonstrating its computational efficiency and robustness, particularly in mini-batch processing.

**Strengths:**

- NSA introduces a new approach for representation alignment with applications in dimensionality reduction, structure-preserving autoencoders, and robustness analysis, highlighting its adaptability to multiple tasks.

- Its quadratic computational complexity improves on the cubic complexity of alternative metrics like RTD, making it suitable for large datasets and mini-batch processing in training.

- NSA is evaluated across multiple tasks and datasets, and compared with established metrics (CKA and RTD).

**Weaknesses:**

__Figure 1__: all 3 plots in the left column are the same.

__Specificity assumptions:__ In section 4.1.1, the authors expect that the same layers of two networks trained on the same data and differing only in the initial weights should have high structural similarity. However, the actual layer in which similar features are learned may vary, particularly in ResNets (due to their residual connections). This is a well-known phenomenon: residual connections allow networks to adapt flexibly, enabling the model to skip certain layers or distribute features across them depending on initial weights and learning dynamics. See:

[1] Veit, A., Wilber, M. J., & Belongie, S. (2016). Residual networks behave like ensembles of relatively shallow networks. Advances in neural information processing systems, 29.

Thus, instead of showing a single example result in Figure 1, the authors would make a stronger case if they (i) reported the average across multiple instances of the same networks; and (ii) used multiple architectures and datasets.

__Equation 6__: for the units of LNSA to make sense, you should take the inverse again, after computing the mean of the inverses. That's what MacKay and Ghahramani actually do -- notice the -1 power in the formulas for their estimators ($\hat{m}^{-1}$). You can also check this on the source code they provided. In their Fig. 2, the best curves are: "the inverse of the average of the inverse m-hats (orange), and our preferred maximum likelihood estimator (which is just equation (7) again."

Having said that, I don't think you should compute the individual residuals using the Lid inverses. The residuals should keep their units of "dimension". How do the authors justify this?

__GNSA:__ I see a problem with this dissimilarity in that it can produce large values if the geometry of the manifold changes but the topology stays the same. A classic example where this would happen is for the "swiss roll" dataset (https://scikit-learn.org/1.5/auto_examples/manifold/plot_swissroll.html): the GNSA value comparing the original roll and its unrolled counterpart would be very large since, although the first several nearest neighbors of a point $i$ would not change their distances much, points that are far away (following along the spiral) would become considerably farther after flattening the roll. I believe this would lead to large GNSA even though the two manifolds are topologically identical. Have the authors considered this? If they agree, I suggest a more thorough discussion on strengths and weaknesses of GNSA.

__Lack of ground truth__: I believe this study would greatly benefit from using toy datasets that provide some ground truth to verify the efficacy of the method proposed. E.g., Gaussian clusters of various dimensionalities, the 1-D spiral, the 2-D S-curve, a plane with a hole; these have been classically used in the manifold learning literature. Here are a couple examples of recent papers that use interesting toy datasets as ground truth for comparing low-dimensional embeddings and dimensionality:

[2] Wang, Yingfan, et al. (2021) "Understanding how dimension reduction tools work: an empirical approach to deciphering t-SNE, UMAP, TriMAP, and PaCMAP for data visualization." Journal of Machine Learning Research 22.201: 1-73.

[3] Dyballa, L., & Zucker, S. W. (2023). IAN: Iterated Adaptive Neighborhoods for manifold learning and dimensionality estimation. Neural Computation, 35(3), 453-524.

It would be informative to have some simple, intuitive examples that could be directly visualized in 2 or 3 dimensions. Such datasets could be perturbed in ways that _did_ change their topology and structural relationships vs. others that _did not_, the goal being to check whether the values produced by LNSA and GNSA would reflect the truth.

**Questions:**

__Figure 1:__ assuming the values being plotted are means, how many tests were performed? What are their standard deviations? This is especially important since the data is being subsampled for the computations, and training seeds will produce different networks.

__Figure 3:__ same questions here with regards to whether the curves represent means. Stating how many repetitions and the standard deviations is important to understand the significance of these curves.

__Lines 324--328__: I had trouble understanding the sensitivity test. Although the notion of testing robustness to the removal of principal components makes perfect sense to me, it was not clear how the plots in Fig. 1 demonstrated, e.g., that "NSA is more sensitive to the removal of high variance PCs compared to RTD and CKA". Moreover, I'm not sure how to interpret the values for "detection threshold", especially since the values in the main text are different than those in the figure. What are the "baselines" mentioned in the plots' legends?

__Line 435:__ "the latent embeddings are then tested on their ability to predict the existence of links between nodes". How exactly are they tested on this? Are they used as inputs in another GCN? This wasn't clear to me.

In lines 197, 199, 272, surely the authors mean dissimilarity, not similarity (since they compute distances)? There are more instances throughout the paper where these metrics are called "similarities".

__Line 497:__ "We used a GCN along with four __robust__ GNN variants...". Why robust? Robust to what exactly?

__Line 500:__ "by introducing perturbations ranging from 5% to 25%". These percentages are w.r.t. what exactly? And what is the nature of these perturbations? Removing/changing links, nodes, or both?

__Minor points:__

- I found no pointer to Figure 3 in the main text.

- Line 493: "we applied NSA in the context of GNNs, but the method __can__ be equally effective in analyzing the robustness of other architectures". I recommend changing __can__ to "might", or "could", unless the authors have actually tested this empirically.

- Line 503: I recommend saying "the __original__ graph" instead of "the _clean_ graph".

---

> ### Author Response · Authors · 2024-11-21
>
> We thank the reviewer for taking the time to evaluate our work in detail and for their comprehensive review. We have added several experiments to the revised manuscript as responses to the raised concerns and present our responses to the queries below:
>
> **1. Figures on the left in Figure 1 are wrong.**
>
> We apologize for the error. We have fixed this in the revised manuscript
>
> **2. Specificity assumptions, especially in ResNets**
>
> Thank you for the constructive feedback. Numerous studies have shown that similar layers across neural networks trained on the same data, differing only in initial weights, often exhibit high structural similarity upon convergence. This has been observed across various architectures, including ConvNeXt, VGG, ResNets, and Transformers, as demonstrated by prior structural similarity metrics. We present a more detailed response to this in **Global Response 3**. We also provide additional results in Appendix V showing consistent layerwise similarity across multiple GNN architectures and datasets, further supporting this phenomenon across diverse network types. For additional evidence across architectures and datasets, we refer to [1,2,3], which provide a broader set of results on various architectures. Since NSA performed on par or better than these metrics in our tests, we expect it to extend effectively to other architectures not explicitly covered in this paper.
>
> **Explanation of ResNet Analysis and Layer Selection:** Our ResNet similarity heatmaps take outputs from the end of each residual block rather than from intermediate layers within blocks. This choice aligns with findings from Kornblith et al.[1] with CKA, where ResNet layerwise comparisons show distinct patterns only when examining post-residual outputs. When every layer (including intermediate layers) is compared, as in Figure 4 of [1], the similarity matrix forms a grid-like pattern due to mismatches within block layers. By focusing on outputs at the end of each block (8 blocks in ResNet-18, plus the final fully connected layer), we observe a clearer similarity pattern across instances of the same architecture. This approach also aligns with findings by Veit et al. (2016)[4], which demonstrated that in ResNets, most significant gradient flow occurs along the short paths (skip connections and post-residual outputs), as longer paths contribute minimal gradients.
>
> Experimental Setup: For thoroughness, we compute average similarity over 10 trials for each metric when using it in subsets. For Figure 1, we average the performance across 3 separate runs (6 models trained with different seeds forming 3 pairs of comparisons). The standard deviation of most metrics on subset computation is a few orders of magnitude below the mean value and does not affect the results, so we do not include it in Figure 1. For reference we provide the standard deviation on both RTD and NSA in Table 1 and provide standard deviation of NSA's individual components in Appendix Q. Additionally we also present layerwise specificity tests in Appendix U where we show the standard deviation of the metrics across runs for Figure 1's specificity tests. We also report layerwise similarity results of ResNet-18 and ResNet-34 trained on ImageNet in Appendix I for further validation on image datasets. We hope these additional experiments will alleviate the reviewer's concerns.
>
> **3. Query on Equation 6**
>
> Thank you for your question. While LocalNSA is inspired by the Local Intrinsic Dimensionality (LID) measure proposed by MacKay and Ghahramani, we intentionally deviate from their formulation for the following reasons:
>
> - LocalNSA's Objective: While LID computes the dimensionality of the entire space, LocalNSA does not aim to measure the LID itself. Instead, it is designed to quantify pointwise discrepancies between two spaces, ensuring that its theoretical properties remain valid.
> - Pointwise Discrepancy Requirement: LocalNSA needs to operate at the pointwise level to preserve its role as a structural similarity metric. This requirement distinguishes it from global estimators like the one in MacKay and Ghahramani's formulation.
> - Normalization and Range Stability: The use of the inverse in LocalNSA is a deliberate design choice. This ensures that:
> (a) LocalNSA values remain consistent across different mini-batch sizes, maintaining stability during training.
> (b) The values stay within a reasonable numerical range, avoiding issues such as skewed loss functions. As demonstrated in Figure 3b of Mackay and Ghahramani’s note, directly averaging the inverses (as in their estimator) results in a much wider range of values, which could lead to instability when used as a loss function.
> - Theoretical Properties: The choice to use the inverse does not affect LocalNSA's theoretical properties. For instance, the convergence of x−y is equivalent to the convergence of 1/x - 1/y. Thus, the design choice to not inverse again is primarily for practical and numerical convenience.

---

> ### Author Response · Authors · 2024-11-21
>
> **4. GNSA can have problems with Swiss Roll and similar datasets**
>
> Thank you for your valuable feedback. We agree that GNSA, as currently formulated with  Euclidean distances, preserves the exact geometric structure rather than the topology of the manifold. This design choice means that NSA is sensitive to changes in global distances, as observed in cases like the Swiss Roll dataset when flattened. Appendix R specifically addresses this situation, highlighting that NSA’s use of Euclidean distance is intended to preserve geometric fidelity rather than topological structure.
>
> To extend NSA’s applicability to manifold topology, one can substitute Euclidean distance with geodesic distance. This modification enables NSA to capture the underlying manifold by approximating geodesic distances between points, preserving topological similarity while maintaining all NSA properties. Appendix R compares NSA-AE’s performance on the swiss roll dataset when it aims to minimize euclidean distances vs geodesic distances.
>
> **5. Addition of toy and visualization focused datasets**
>
> Thank you for your feedback. In our study, we incorporate several well-established visualization focused datasets to validate NSA’s efficacy and to demonstrate its flexibility across different use cases. We also added results from the Mammoth dataset taken from [5] in the revision. Please find below a list of visualization focused datasets in the manuscript:
>
> - In Section 4.2, we utilize the COIL-20 dataset to evaluate NSA’s ability to preserve structural similarity during dimensionality reduction.
> - The Swiss Roll dataset is used to illustrate the difference between geometric and topological preservation. By comparing NSA with Euclidean distances to NSA with geodesic distances, we show how the choice of distance metric influences NSA’s ability to capture manifold structure. The Swiss Roll dataset is also used to perform ablation studies on the parameter $k$ of LNSA.
> - In Appendix R, we analyze NSA’s performance on toy datasets like Spheres and Mammoth to evaluate its capacity to maintain both global and local structure. Additionally, we also use the Spheres dataset to perform visual ablation studies in Appendix Q that demonstrate how different parameter choices affect NSA’s performance.
> - In Section 4.4 we demonstrate how NSA can pinpoint sources of discrepancy by switching to a node wise variation of NSA (refer to the formula here). This showcases NSA’s ability to not only identify structural discrepancies but also pinpoint the source of these deviations.
>
> **6. Mean and Standard Deviation values for Figure 1 and Figure 3**
>
> Thank you for your feedback. As previously stated, we compute average similarity over 10 trials for each metric when using it in subsets. For Figure 1, we average the performance across 3 separate runs (6 models trained with different seeds forming 3 pairs of comparisons). The results on mean and standard deviation values across runs and the experimental setup is presented in Appendix U. In Figure 3 we run our experiments on the Cora dataset which has 2708 nodes and hence we do not need to use subsets to compute NSA. We have clarified in the main text
>
> **7. Clarifications on the sensitivity test**
>
> Thank you for the question. In principle, the idea proposed by Ding et al.[6] states that as principal components are sequentially removed, starting with the lowest-variance components, the dissimilarity score should increase. A good metric should be sensitive to the removal of these high-variance components. Figure 1 demonstrates that NSA’s dissimilarity score rises significantly when high-variance PCs are removed, indicating greater sensitivity to impactful structural changes. In contrast, RTD and CKA show a flatter response, reflecting reduced sensitivity until the most significant components are removed. In line with Ding et al.'s[6] setup, we perform this analysis on the representations extracted from the first and last layers of a neural network.
>
> Since different metrics operate within different ranges, the absolute values of the curves may not be directly comparable. To address this, [6] proposed a detection threshold, which defines the dissimilarity score above which changes become “detectable.” This baseline is the dissimilarity score between representations from differently initialized networks, serving as a reference for determining when structural changes due to PC removal become significant.
>
> The main text refers to the average percentage of principal components that can be removed before crossing the detection threshold. The confusion arose because we referred to this average percentage as the detection threshold itself. We have updated the text to clarify this distinction.

---

> > ### Author Response · Authors · 2024-11-21
> >
> > **8. Line 435. Clarification on how the embeddings are tested for link prediction, using ROC-AUC**
> >
> > The latent embeddings are tested for link prediction by computing the dot product between the embeddings of node pairs, which was the method used during the original GCN training to minimize the link prediction loss. The computed values are then compared to the ground truth labels (consisting of an equal amount of positive and negative edge indices) to evaluate performance using metrics like ROC-AUC.
> >
> > The dot product reflects the likelihood of an edge: closer nodes in the embedding space produce higher dot products, indicating a higher edge probability. We use RTD-AE and NSA-AE to reduce the dimensionality of the original representation space and test the reduced space in the same manner. Since the reduced space maintains a one-to-one mapping with the original, a good structure-preserving method should retain the relative distances between node representations. This is quantified by measuring the ROC-AUC score in the reduced space, reflecting how well the structural integrity is preserved. We direct the reviewer to **Global Response 2** for additional clarification on Section 4.3
> >
> >
> > **9. Dissimilarity vs similarity**
> >
> > Thank you for pointing this out. We have added a footnote in the paper to denote that we use the term similarity index colloquially throughout the paper to refer to metrics that quantify relationships between representation spaces. We have also identified and changed the phrasing wherever we use the term similarity when talking about NSA computations in section 4.1. We will identify and change the phrasing throughout the manuscript in the final version.
> >
> > **10. Line 497. Why Robust GNNs and robust to what exactly?**
> >
> > Thank you for the question. By "robust," we refer to GNN variants designed to improve resilience against adversarial attacks or noisy input data. These models either have preprocessing steps or built in training mechanisms to detect and remove perturbations with varying degrees of effectiveness. We chose robust variants of GNNs for two reasons specifically.
> > 1. Gradient in Performance: Robust GNN variants exhibit varying degrees of resilience to adversarial perturbations. This allows us to evaluate NSA’s sensitivity across a gradient of robustness, providing a nuanced view of how structural discrepancies manifest in models with different levels of inherent robustness.
> > 2. Comparison with Related Work: By using a similar set of architectures to Mujkanovic et al. (2022) [7], we align our experimental setup with prior works in adversarial robustness. This enables us to correlate our findings with theirs.
> >
> > **11. What do the perturbation percentages mean?**
> >
> > The perturbation percentages represent the proportion of edges added to the graph during an adversarial attack. For example, a 10% perturbation means that edges equal to 10% of the total number of edges in the original graph were added. We have clarified in the main text that the perturbations involve only edge additions and will ensure this is explicit in the final version of the manuscript.
> >
> > **12. No pointer to Figure 3**
> >
> > Thank you for pointing this out. We have fixed it.

---

> > > ### Author Response · Authors · 2024-11-21
> > >
> > > **13. Applying NSA on CNN adversarial analysis**
> > >
> > > Thank you for your feedback. NSA looks at the original and perturbed representation space when a model has been attacked. Regardless of the architecture, if an adversarial attack attempts to modify the model by changing how it classifies certain data points, then the output representation space of the perturbed model will change and NSA will be able to detect this. We have added experiments in Appendix I where demonstrate that NSA shows a strong correlation with misclassification rate on a CNN too, using a ResNet trained on CIFAR-10. We also present experiments showing NSA's ability to perform pointwise analysis to identify the source of perturbations with various ResNet architectures.
> > >
> > > **14. Original graph instead of clean graph**
> > >
> > > Thank you for your feedback. We have made this change.
> > >
> > > We hope we have addressed all the queries raised by the reviewer. Please let us know if you have any additional concerns.
> > >
> > >
> > >
> > >
> > > [1] Simon Kornblith, Mohammad Norouzi, Honglak Lee, and Geoffrey E. Hinton. Similarity of neural
> > > network representations revisited. IN ICML 2019
> > >
> > > [2] Serguei Barannikov, Ilya Trofimov, Nikita Balabin, and Evgeny Burnaev. Representation topology
> > > divergence: A method for comparing neural network representations. In ICML 2022
> > >
> > > [3] Zuohui Chen, Yao Lu, Wen Yang, Qi Xuan, and Xiaoniu Yang. 2021. Graph-Based Similarity of Neural Network Representations. ArXiv preprint (2021)
> > >
> > > [4] Veit, A., Wilber, M. J., & Belongie, S. (2016). Residual networks behave like ensembles of relatively shallow networks. Advances in neural information processing systems, 29.
> > >
> > > [5] Wang, Yingfan, et al. (2021) "Understanding how dimension reduction tools work: an empirical approach to deciphering t-SNE, UMAP, TriMAP, and PaCMAP for data visualization." Journal of Machine Learning Research 22.201: 1-73.
> > >
> > > [6] Ding.,et al. Grounding representation similarity through statistical testing. Advances in Neural Information Processing Systems, 2021
> > >
> > > [7] Felix Mujkanovic, Simon Geisler, Stephan Günnemann, and Aleksandar Bojchevski. Are defenses for graph neural networks robust? In Neural Information Processing Systems, NeurIPS, 2022

---

> > > > ### Author Response · Authors · 2024-11-26
> > > > **Additional experiments to clarify the formulation of LNSA**
> > > >
> > > > Dear Reviewer,
> > > >
> > > > In the latest revision of the manuscript we also provide empirical evidence in Appendix Y to support our design choice of not taking another inverse as recommended by Mackay and Ghahramani [1]. We show that taking another inverse not only leads to absurdly large LNSA values but also results in erratic behavior. We hope this satisfies the reviewer’s concerns regarding the choice to not invert the LID values again before computing LNSA.
> > > >
> > > > Additionally as the revision period will be ending soon, we would be extremely grateful if you could take the time to review our rebuttal and let us know if it has resolved all of your concerns. If you have any further questions, we would be happy to answer them.
> > > >
> > > > Sincerely,
> > > >
> > > > The Authors
> > > >
> > > >
> > > > [1] David J.C. MacKay and Zoubin Ghahramani. Comments on ’maximum likelihood estimation of intrinsic dimension’ by e. levina and p. bickel (2004), 2005.

---

> > > > > ### Comment · Reviewer_1hcq · 2024-11-26
> > > > >
> > > > > I thank the authors for their responses.
> > > > >
> > > > > "Appendix R compares NSA-AE’s performance on the swiss roll dataset when it aims to minimize euclidean distances vs geodesic distances."
> > > > >
> > > > > How were geodesic distances computed/estimated? I saw no information about this in the manuscript.

---

> > > > > > ### Comment · Reviewer_1hcq · 2024-11-26
> > > > > >
> > > > > > In my review I suggested including toy datasets to more concretely evaluate the performance of the method as a measure of structural dissimilarity. "Such datasets could be perturbed in ways that did change their topology and structural relationships vs. others that did not, the goal being to check whether the values produced by LNSA and GNSA would reflect the truth."
> > > > > >
> > > > > > Did the authors have the chance to perform any of these structural perturbation experiments as means of providing ground truth? I could not find them in the revised manuscript.

---

> ### Comment · Reviewer_1hcq · 2024-11-26
>
> While I appreciate the authors' efforts, in my view the changes to the main text of the manuscript were minimal, despite important concerns raised by me and other reviewers. The material added to the appendices does not convincingly demonstrate how NSA represents a significant advance as a structure representation metric other than in the particular examples chosen. Thus I will maintain my original score.

---

> ### Author Response · Authors · 2024-11-27
>
> We sincerely thank the reviewer for engaging in a discussion and their feedback.
>
> **1. Clarification on Geodesic Distance computation**
>
> We use the Floyd-Warshall algorithm to approximate geodesic distances, ensuring that the pairwise shortest path distances on the manifold are accurately represented. We first create a k-nearest neighbors graph from the dataset then compute the all pairs shortest path using the FW algorithm. Given that the reference space remains fixed, this is a one time preprocessing cost. And since this matrix inherently reflects Euclidean distances when the representation space is reduced to its manifold dimension, we can use NSA with no modifications to the metric to operate as a loss function. We have added a line in Appendix R mentioning the algorithm used. We hope this clarifies the process used to perform the swiss roll manifold approximation experiment. Please let us know if you require additional details.
>
> **2. Addition of toy dataset experiments**
>
> We did perform experiments with a few toy datasets to visually observe the ability of NSA to preserve the structure of the representation space.
>
> - Figure 14,15 and 16 showcase NSA’s ability to preserve local and global structure with the 3D Swiss Roll, Spheres and Mammoth Dataset
> - In the latest revision we also added visualization experiments with a 1D spiral in 2D space in Appendix Z, where we introduce targeted transformations to the original space (we perform global structure preserving transforms, manifold preserving transforms, structure altering transforms and manifold altering transforms) and present the GNSA and LNSA values to demonstrate their invariances and properties. Both GNSA and LNSA reflect the changes made to the ground truth based on their definitions i.e GNSA changes when the global structure is modified, LNSA changes when local structure is changed.
>
> We hope this satisfies the reviewer’s concerns on using Toy Datasets to visualize the effects of NSA.

---

> > ### Author Response · Authors · 2024-11-27
> > **Clarifications on manuscript structure**
> >
> > The main text of our manuscript is structured following the standard framework for introducing and validating structural similarity metrics. In Section 3, we introduce NSA and provide its formal definition. Section 4. proves its validity through specificity and sensitivity tests, while Sections 4.2 to 4.4 demonstrate its application across multiple use cases.
> >
> > The appendices present theoretical guarantees, detailed ablation studies, additional experimental results, and practical implementation details. If the reviewer believes that specific content from the appendices should be included in the main manuscript or if additional results are necessary to address their concerns, we welcome such suggestions and are happy to restructure accordingly.

---

> > > ### Author Response · Authors · 2024-11-27
> > > **Clarification on novelty**
> > >
> > > We respectfully disagree with the statement that NSA does not convincingly demonstrate a significant advance as a structural representation metric. Besides outperforming previous works in our application benchmarks, NSA offers the following advancements over previous state-of-the-art (SoTA) methods:
> > >
> > > - **Higher Sensitivity and Specificity:** NSA demonstrates superior sensitivity to high-variance components and specificity to structural changes across layers of neural networks compared to methods like RTD and CKA. These properties are validated in Section 4.1 and Appendix U.
> > > - **Computational Efficiency:** NSA is significantly more computationally efficient than its best competitor, RTD
> > > - **Differentiability and Continuity:** NSA is fully differentiable and continuous, enabling its use as a loss function in optimization pipelines. This is a critical limitation of prior metrics, which are often discrete and non-differentiable.
> > > - **Global Approximation in Mini-Batching:** As far as we are aware, NSA is the first structural similarity metric that is differentiable and capable of approximating its global value when applied in mini-batches. Previous metrics, even if differentiable, lack this property, making NSA uniquely suited for scalable applications
> > > - **Local and Global Structure Preservation:** NSA provides both global and local focus on structure preservation unlike previous works who only look at one of the two perspectives
> > > - **Explainability:** NSA’s reliance on distance-based measures makes it inherently explainable, as discrepancies can be directly tied to measurable distances between points in the representation space.
> > > - **Pointwise Discrepancy Analysis:** Unlike prior metrics, NSA provides pointwise granularity, enabling the identification of specific sources of structural discrepancies in the representation space. Topologically focused works like RTD look at topological features instead of points and thus do not have the ability to perform pointwise analysis
> > > - **Flexibility:** NSA’s formulation can be modified to suit alternative distance measures without fundamentally altering the metric
> > >
> > > We would be grateful if the reviewer could clarify which **important concerns** remain unaddressed and why they believe NSA does not represent a significant advancement over existing metrics. This would allow us to address their concerns more directly and improve the clarity and impact of our manuscript.

---

### Official Review · Reviewer_rENM · 2024-11-04

**Soundness:** 3
**Presentation:** 3
**Contribution:** 3
**Rating:** 8
**Confidence:** 3

**Summary:**

The paper introduces Normalized Space Alignment(NSA), a new manifold analysis technique designed to compare neural network representations; NSA compares pairwise distances between point clouds from the same data source but with different dimensionalities. NSA is proposed as both a differentiable loss function and a similarity metric, and it is computationally efficient. The paper demonstrated the NSA's versatility in representation analysis, structure-preserving tasks, and robustness testing against adversarial attacks.

**Strengths:**

1) NSA can be used both as a loss function and a similarity metric across different applications
2) NSA is designed to work efficiently in large-scale applications with a quadratic complexity that is better than some existing methods
3) It is also effective in preserving structural characteristics and identifying vulnerabilities in neural networks, even under adversarial attacks
4) the paper provides a thorough analysis with multiple experiments and comparisons to other methods like RTD, CKA validating NSA's effectiveness

**Weaknesses:**

1) The reliance on Euclidean distance as a primary metric may limit performance in high dimensional spaces due to curse of dimensionality
2) NSA is versatile but may not require careful tuning and modifications to work effectively in specific scenarios
3) The limitations of NSA are not explored beyond high-dimensionality issue

**Questions:**

1) How does NSA perform in extremely high-dimensional spaces where Euclidean distance is known to be problematic? Are there alternative distance metrics that could be integrated into NSA?
2) How sensitive is NSA to parameter settings, and what are the best practices for tuning it in different applications (e.g., adversarial robustness vs. dimensionality reduction)?
3) Given the versatility of NSA, do you envision any specific areas where its application would be limited or challenging?

---

> ### Author Response · Authors · 2024-11-21
>
> We thank the reviewer for their feedback and for appreciating our work. We address the reviewer's concerns below:
>
> **1. The reliance on Euclidean distance as a primary metric may limit performance in high dimensional spaces due to curse of dimensionality**
>
> We appreciate the reviewer’s observation regarding the potential limitations of Euclidean distance in high-dimensional spaces due to the curse of dimensionality. While it’s true that Euclidean distance can face challenges in high-dimensional settings, we chose it deliberately for a few key reasons:
>
> 1. **Prevalent Use Across Neural Network Tasks:** Euclidean distance remains a widely adopted metric in representation learning, particularly in tasks involving contrastive and triplet losses, where it has shown strong empirical performance. Many state-of-the-art neural network architectures still rely on Euclidean distance for computing similarity between embeddings, even in high-dimensional spaces.
> 2. **Empirical Robustness:** In our experiments, NSA demonstrated robust performance across multiple datasets and tasks, even when applied to high-dimensional embeddings (we test up to 200,000 dimensions). This suggests that, while Euclidean distance may not be theoretically optimal under the curse of dimensionality, it continues to yield practical benefits in diverse applications, including representation analysis and alignment tasks.
> 3. **Potential for Adaptability:** The NSA framework is flexible enough to accommodate alternative distance metrics if future work finds this necessary. In this study, we prioritized Euclidean distance for its simplicity, efficiency, and empirical success in maintaining local and global structural alignment across different model architectures.
>
> Additionally we include a local component for NSA for this specific purpose. LocalNSA only looks at the k-nearest neighborhood of a point and can alleviate some of the discrepancies that can arise from GlobalNSA running at high dimensionalities.
>
>
> **2. NSA is versatile but may require careful tuning and modifications to work effectively in specific scenarios**
>
> Certain applications may benefit from tuning NSA parameters to optimize performance. However, our findings suggest that NSA requires minimal parameter adjustments compared to traditional manifold learning techniques such as ISOMAP, where parameter selection can heavily impact results. NSA’s tuning process is straightforward and produces only incremental gains, indicating that the method performs robustly without extensive parameter optimization.
>
> To provide further transparency, we included ablation studies in Appendix Q to identify the optimal parameter configurations for NSA and present a more detailed response to the need for parameter tuning in **Global Response 1**.
>
> **3. The limitations of NSA are not explored beyond high-dimensionality issue**
>
> Thank you for raising this point. We acknowledge that while NSA has proven effective across a variety of tasks, it has a few limitations worth noting:
>
> 1. **Minor Parameter Tuning:** NSA requires minimal parameter tuning, though, as shown in our ablation studies (Appendix Q), these adjustments result in only small gains compared to manifold learning methods like ISOMAP, which are more sensitive to parameter selection.
>
> 2. **High-Dimensional Euclidean Distance and Geometric vs. Topological Similarity:** As noted, NSA leverages Euclidean distance, which can be affected by high-dimensionality. However, the versatility and widespread success of Euclidean distance in neural network applications make it a pragmatic choice for NSA. To address this, we present studies in Appendix R where we evaluate the reconstruction of a Swiss Roll using NSA-AE and demonstrate how geodesic distance can be used as replacement for euclidean distance in high dimensional spaces or in spaces where the manifold lies in a lower dimension than the original data. NSA-AE with geodesic distance as the distance measure can unravel complex manifolds.
>
> 3. **Inability to measure functional similarity:** NSA is primarily a structural similarity index and is therefore incapable of measuring functional similarity accurately. We illustrate this with cross architecture experiments in Appendix V. While NSA shows promise, it is incapable of perfectly capturing layerwise similarity even though both GNN architectures are trained on the same task and on the same dataset making them functionally similar.
>
> These considerations are relatively minor and do not affect NSA's definition and superiority in its domain, and our experimental results demonstrate NSA’s robustness across datasets and tasks.

---

> > ### Author Response · Authors · 2024-11-21
> >
> > **4. How does NSA perform in extremely high-dimensional spaces where Euclidean distance is known to be problematic? Are there alternative distance metrics that could be integrated into NSA?**
> >
> > We have indeed explored NSA’s performance in high-dimensional spaces through our ResNet-18 experiments (Appendix I), where the hidden layer embeddings, once flattened, reach very high dimensionalities (up to approximately 200,000). Despite these high dimensionalities, NSA has consistently demonstrated strong alignment and similarity retention between representation spaces, indicating robust performance even in challenging settings.
> >
> > **Alternative Distance Metrics:** We acknowledge that alternative distance metrics may offer benefits for certain tasks or extremely high-dimensional spaces. Potential alternatives include:
> > - Cosine Similarity: Often used in high-dimensional embeddings, cosine similarity could serve as an alternative for comparing directional similarity, though it lacks the same geometric interpretation as Euclidean distance.
> > - Mahalanobis Distance: This metric could adapt to the covariance structure of the data, capturing differences in distributions more effectively than Euclidean distance, particularly in cases with correlated features.
> > - Geodesic Distance: For complex manifolds, geodesic distances (as applied in ISOMAP) could better capture topological structure (as explored in Appendix R).
> >
> > While Euclidean distance remains the default in NSA for its simplicity and efficiency, NSA’s modular design allows for flexibility in adopting alternative metrics, depending on the data’s structure and specific application requirements.
> >
> > **5. How sensitive is NSA to parameter settings, and what are the best practices for tuning it in different applications (e.g., adversarial robustness vs. dimensionality reduction)?**
> >
> > NSA is robust to a wide range of parameter settings, as demonstrated in our ablation studies in Appendix Q. The key parameters include the weights for local (LNSA) and global (GNSA) components and the k-value in the k-NN graph. For most applications:
> >
> > - **Adversarial Robustness:** Prioritize GNSA (higher weight on global structure) to capture overall structural discrepancies caused by perturbations.
> >
> > - **Dimensionality Reduction:** Balance LNSA and GNSA to preserve both local and global structures. The choice of k is less critical, with a moderate range working well across datasets.
> > Our studies show that NSA achieves competitive performance with default parameter settings, and fine-tuning provides incremental improvements tailored to specific tasks. We recommend starting with default values and adjusting based on the task's emphasis (local vs. global structure).
> >
> > **6. Given the versatility of NSA, do you envision any specific areas where its application would be limited or challenging?**
> >
> > While NSA is versatile, its application may be limited in scenarios where:
> >
> > - Functional Similarity: Tasks prioritizing functional rather than structural similarity (e.g., task-specific feature alignment) may require alternative metrics.
> > - Topology-Specific Analysis: Applications requiring explicit topological analysis (e.g., persistent homology) may benefit more from specialized topology-centric methods.
> >
> > These limitations are inherent trade-offs of NSA’s design, focusing on structural similarity, and can often be addressed through preprocessing or complementary metrics.
> >
> >
> > We hope our responses clarified any queries that the reviewer had. Please let us know if you have any more questions.

---

> > > ### Author Response · Authors · 2024-11-26
> > > **Friendly Reminder**
> > >
> > > Dear Reviewer,
> > >
> > > The revision period will be ending soon. We would be extremely grateful if you could take the time to review our rebuttal and let us know if it has resolved all of your concerns. If you have any further questions, we would be happy to answer them.
> > >
> > > Sincerely,
> > > The Authors

---

### Author Response · Authors · 2024-11-21
**Global Response Part 1**

We are sincerely grateful to all reviewers for their thoughtful feedback and suggestions, which we believe are very beneficial for our work. Your suggestions have helped improve our manuscript and below we present responses to a few common queries along with a description of the changes made to the manuscript in this revision.

**1. NSA requires significant fine-tuning of hyperparameters and ablation studies are missing**

We appreciate the reviewers' concerns regarding the dependency of NSA on the hyperparameters $l$, $g$ and $k$. While NSA includes three parameters, $l$ and $g$ are straightforward to tune as they simply control the balance between local manifold preservation and global geometric structure preservation. In most cases, unless one is weighted significantly more than the other, LNSA and GNSA complement each other to produce consistent results. Across all experiments where both components are used together, we set \($l$= 1\) and \($g$ = 1\). Ablation studies provided in Appendix Q demonstrate that GNSA alone performs satisfactorily for both dimensionality reduction and downstream tasks, while LNSA alone is less effective. However, combining both components produces the best results. Additionally, we include visual results using the Spheres dataset to show that the encoder successfully preserves the structure of the dataset across a wide range of $l$ and $g$ values.

The parameter $k$ requires more nuance than $l$ and $g$, as it controls the number of nearest neighbors considered for LNSA. Unless one aims to optimize local structure preservation at a specific scale, most $k$ values perform well to preserve local structure. In the revised manuscript, we provide ablation studies on $k$, including visual results on the Swiss Roll dataset, illustrating how varying $k$ affects the degree of structure preservation. Additionally, we present empirical results demonstrating LNSA’s convergence to its global value under mini-batching with the appropriate $k$ value. We also offer guidance on selecting an optimal $k$ to align with the desired level of local structure preservation.


**2. Clarifications on Section 4.3 (Downstream Task Analysis)**

In Section 4.3, we evaluate NSA’s ability to preserve critical structural information during dimensionality reduction by performing a link prediction task on the lower-dimensional embeddings. The embeddings are generated by reducing the original high-dimensional representation space using NSA-AE, and we assess their quality using ROC-AUC. Even in the reduced dimensional space, there is a one-to-one mapping between nodes, allowing us to compute ROC-AUC using the original graph's positive edge indices and an equal number of sampled negative edge indices. The dot product of the embeddings determines the probability of a link existing between two nodes.

This experiment is not intended to showcase NSA as a supplementary loss function for improving link prediction but rather to highlight its ability to retain the structural integrity of the representation space after dimensionality reduction. While retraining a GCN in the reduced dimension may be feasible for smaller models, this approach becomes impractical for larger models where pretraining is computationally expensive. Instead, NSA-AE offers a computationally efficient alternative by ensuring that the relative distances and contextual information in the reduced-dimensional space remain preserved, yielding performance comparable to a newly trained model as shown in Table 2.

To clarify, link prediction is not the ideal task for NSA, as it requires a reference space to align to. For link prediction, directly training a GCN is simpler and more efficient than introducing NSA-AE. NSA is a metric for comparing representations across two spaces. It can be used to explain the performance of machine learning tasks. It can also be used as a loss function that can be differentiated and estimated quickly in mini batches and will be much more effective in tasks like dimensionality reduction, adversarial training, and knowledge distillation, where preserving representation structure or aligning to a reference space is critical. But it is up to a method to decide how to use it for solving a specific machine learning problem (in the same spirit as other similarity metrics such as RTD and CKA).

---

> ### Author Response · Authors · 2024-11-21
> **Global Response Part 2**
>
> **3. Different initializations produce different models and there is no reason to assume these should have the same structures**
>
> The idea that corresponding layers of architecturally similar models show high relative structural similarity is not novel to NSA but is a well-established benchmark for evaluating similarity metrics. This approach was first introduced by Kornblith et al.[1] in their seminal work on CKA, and it has since been validated and adopted by several subsequent studies [2,3,4,5,6,7]. The idea of similar generalization and performance between networks trained with different seeds has been extensively examined by Thomas et al.[8]. Their experiments with 100 BERT models confirm that, while functional differences may lead to significant variability in out-of-distribution (OOD) performance, networks trained to convergence on the same dataset exhibit highly similar structural representations, particularly for in-distribution data. This idea is key to the validity of not only the layerwise analysis presented in similarity metric papers but also to the grounding tests proposed by Ding et al [7]. This structural alignment is a key factor contributing to the consistent performance of such networks on in-distribution tasks.
>
> All our specificity tests are conducted exclusively on in-distribution data, ensuring that our analysis aligns with the well-validated findings from prior literature. We have fixed the error with the figures in Section 4.1 and also improved the writing on Specificity tests. We also include an in-depth layerwise breakdown of the specificity test in Appendix U.
>
> In our initial presentation, we claimed that a good structural similarity index should show high similarity between architecturally identical networks trained with different weight initializations. We have revised this claim to a more nuanced one: a good similarity index should exhibit the highest relative similarity for corresponding layers in architecturally identical networks with different initializations, compared to non-corresponding layers.
>
> We present a detailed changelog of all the improvements made to the manuscript in this revision:
>
> - Fixed the error in Figure 1 (left) which had the same figure for all 3 metrics.
> - Improved the readability of some figures by replacing them with higher quality versions
> - Added reference to Figure 3 that was missing
> - Rewrote Section 4.1.1 and improved the clarity of Section 4.1.2
> - Added additional Specificity Tests in Appendix U
> - Added ablation studies visualizing the effect of k on the performance of LocalNSA in Appendix Q.
> - Added explanations for the choice of k and figures demonstrating empirical convergence of LocalNSA in mini batching scenarios to -Appendix Q.
> - Added reconstruction results of NSA-AE on the Mammoth Dataset to Appendix R
> -Added GCN performance on Link Prediction at different dimensionalities in Section 4.3
> - Added heatmaps showing cross layer similarity performance on a subset of the ImageNet dataset (100K images) on ResNet-18 and ResNet-34 to show NSA’s performance on large datasets in Appendix I
> - Added a plot to show the variation of mean NSA and standard deviation of Global NSA and Local NSA to show the standard deviation of both metrics is significantly lower when working with subsets in Appendix Q
> - Added results on NSA’s correlation to misclassification rate in CNNs in Appendix I
> - Added visualizations on the 1D spiral in 2D Space dataset to provide simple visual explanations on how LNSA and GNSA reflect the ground truth when different types of transforms are applied to the data in Appendix Z.
>
>
> [1] Simon Kornblith, Mohammad Norouzi, Honglak Lee, and Geoffrey E. Hinton. Similarity of neural network representations revisited. IN ICML 2019
>
> [2] Serguei Barannikov, Ilya Trofimov, Nikita Balabin, and Evgeny Burnaev. Representation topology divergence: A method for comparing neural network representations. In ICML 2022
>
> [3] Enric Boix-Adsera, Hannah Lawrence, George Stepaniants, and Philippe Rigollet. 2022. GULP: a prediction based metric between representations. In NeurIPS.
>
> [4] Zuohui Chen, Yao Lu, Wen Yang, Qi Xuan, and Xiaoniu Yang. 2021. Graph-Based Similarity of Neural Network Representations. ArXiv preprint (2021)
>
> [5] Trofimov, I., Cherniavskii, D., Tulchinskii, E., Balabin, N.,Burnaev, E., and Barannikov, S. Learning topology preserving data representations. In The Eleventh International Conference on Learning Representations, 2023.
>
> [6] Klabunde, M., Schumacher, T., Strohmaier, M., and Lemmerich, F. Similarity of neural network models: A survey of functional and representational measures.
>
> [7] Ding.,et al. Grounding representation similarity through statistical testing. Advances in Neural Information Processing Systems, 2021
>
> [8] R. Thomas McCoy, Junghyun Min, Tal Linzen. BERTs of a feather do not generalize together: Large variability in generalization across models with similar test set performance. 2020 BlackboxNLP workshop

---

### Meta-Review · Area_Chair_5A2V · 2024-12-18

**Metareview:**

The authors propose normalized space alignment (NSA) as an analysis technique for neural network representations. NSA combines the global NSA (GNSA) which compares the pairwise Euclidean distances in the two representations, and local NSA (LNSA) which measures dissimilarity from k-NN graph. The proposed NSA can be applied as an analytical tool and a loss function. The authors evaluate the proposed method on various datasets for these aforementioned tasks.

Although the Reviewers think the proposed approach is interesting, the Reviewers raised concerns on the choice of Euclidean distance in GNSA (for such manifold analysis technique), sensitivity of the k-NN graph in LNSA, scalability for large datasets. The Reviewers also raised concerns on the evaluation without ground truth for manifold analysis task. Additionally, the Reviewers also question about empirical evidences on the advantages of the proposed approach in applications, it is better to compare the proposed approach with recent baselines for the corresponding tasks. Overall, we think that the submission is not ready for publication yet. The authors may consider the Reviewers' comments to improve the submission

**Additional Comments On Reviewer Discussion:**

The Reviewers raised several concerns about the proposed method, especially on the choice of Euclidean metric for GNSA for manifold analysis, and sensitivity of k-NN graph in LNSA. Additionally, the empirical evidences do not convince the Reviewers yet, several concerns are raised as listed above in the meta-review.

---

### Decision · Program_Chairs · 2025-01-22

Reject